



Ocean Science

# Measuring ocean total surface current velocity with the KuROS and KaRADOC airborne near-nadir Doppler radars: a multi-scale analysis in preparation for the SKIM mission

Louis Marié[1], Fabrice Collard[2], Frédéric Nouguier[1], Lucia Pineau-Guillou[1], Danièle Hauser[3], François Boy[4], Stéphane Méric[5], Peter Sutherland[1], Charles Peureux[1], Goulven Monnier[6], Bertrand Chapron[1], Adrien Martin[7], Pierre Dubois[8], Craig Donlon[9], Tania Casal[9], and Fabrice Ardhuin[1]

[1]Laboratoire d'Océanographie Physique et Spatiale (LOPS), UMR 6523, Univ. Brest, CNRS, Ifremer, IRD, Brest, France
[2]OceanDataLab, Locmaria Plouzané, TS1 France
[3]CNRS, Univ. Versailles St Quentin, Sorbonne Université, LATMOS, TS2 France
[4]TS3CNES, Toulouse, France
[5]Institut d'Électronique et de Télécommunication de Rennes (IETR), UMR CNRS 6164, Rennes, France
[6]TS4SCALIAN, Rennes, France
[7]TS5NOC, Southampton, UK
[8]TS6CLS, Ramonville St Agne, France
[9]TS7ESA, Nordwijk, the Netherlands

**Correspondence:** Louis Marié (louis.marie@ifremer.fr)

**Abstract.** TS8 CE1 CE2 Surface currents are poorly known over most of the world's oceans. Satellite-borne Doppler wave and current scatterometers (DWaCSs) are among the proposed techniques to fill this observation gap. The Sea surface KInematics Multiscale (SKIM) proposal is the first satellite concept built on a DWaCS design at near-nadir angles and was demonstrated to be technically feasible as part of the European Space Agency Earth Explorer program. This article describes preliminary results from a field experiment performed in November 2018 off the French Atlantic coast, with sea states representative of the open ocean and a well-known tide-dominated current regime, as part of the detailed design and feasibility studies for SKIM. This experiment comprised airborne measurements performed using Ku-band and Ka-band Doppler radars looking at the sea surface at near-nadir incidence in a real-aperture mode, i.e., in a geometry and mode similar to that of SKIM, as well as an extensive set of in situ instruments. The Ku-Band Radar for Observation of Surfaces (KuROS) airborne radar provided simultaneous measurements of the radar backscatter and Doppler velocity in a side-looking configuration, with a horizontal resolution of about 5 to 10 m along the line of sight and integrated in the perpendicular direction over the real-aperture 3 dB footprint diameter (about 580 m). The Ka-band RADar for Ocean Current (KaRADOC) system, also operating in the side-looking configuration, had a much narrower beam, with a circular footprint only 45 m in diameter. Results are reported for two days with contrasting conditions, a strong breeze on 22 November 2018 (wind speed 11.5 m s$^{-1}$, Hs 2.6 m) and gentle breeze on 24 November 2018 (wind speed 5.5 m s$^{-1}$, Hs 1.7 m). The measured line-of-sight velocity signal is analyzed to separate a non-geophysical contribution linked to the aircraft velocity, a geophysical contribution due to the intrinsic motion of surface waves and the desired surface current contribution. The surface wave contribution is found to be well predicted by Kirchhoff scattering theory using as input parameters in situ measurements of the directional spectrum of long waves, complemented by the short-wave spectrum of Elfouhaily et al. (1997). It is found to be closely aligned with the wind direction, with small correc-

tions due to the presence of swell. Its norm is found to be weakly variable with wind speed and sea state, quite stable and close to $C_0 = 2.0\,\mathrm{m\,s^{-1}}$ at the Ka band, and more variable and close to $C_0 = 2.4\,\mathrm{m\,s^{-1}}$ at the Ku band. These values are 10 %–20 % smaller than previous theoretical estimates. The directional spread of the short gravity waves is found to have a marked influence on this surface wave contribution. Overall, the results of this study support the feasibility of near-nadir radar Doppler remote sensing of the ocean total surface current velocity (TSCV).

## 1 Introduction

The ocean total surface current velocity (TSCV) is defined as the Lagrangian mean velocity at the instantaneous sea surface, corresponding to an effective mass transport velocity at the surface. The TSCV is currently only reliably measured by high-frequency (HF) radars deployed in some coastal regions. Elsewhere, available estimates depend on numerical model outputs, sea level and wind measurements, and on assumptions such as the balance between the surface pressure gradient and the Coriolis force. The situation is similar regarding directional wave statistics, which are currently mainly estimated through numerical modeling.

These estimates of the TSCV are not reliable at small scales, particularly so in the tropical ocean (e.g., Sudre et al., 2013; Stopa et al., 2016), and these limitations hamper current efforts to observe and understand the fluxes of heat, fresh water, carbon, plastics and the coastal impacts of sea states.

Whereas new data on ocean waves are becoming available with the Surface Waves Investigation and Monitoring (SWIM) instrument carried by the China–France Ocean SATellite (CFOSAT) (Hauser et al., 2017, 2020), direct spaceborne measurements of surface current have been limited to a few regions and single projections of the current vector (Chapron et al., 2005; Rouault et al., 2010; Hansen et al., 2011). Several concepts based on SAR CE3 interferometry (Romeiser et al., 2003; Buck, 2005) or Doppler scatterometry (Rodriguez, 2018; Chelton et al., 2019) have been proposed for satellite missions aimed at mapping the ocean surface current vector (see review by Ardhuin et al., 2019). Airborne demonstrators have also been developed in that context (Martin et al., 2018; Rodríguez et al., 2018) and are now becoming operational tools for oceanographic research.

The Doppler frequency shift (DFS) signal provided by these phase-resolving radar instruments is complex: it contains a geophysical contribution due to waves and currents, as well as a large non-geophysical contribution due to the platform motion. The platform velocity in space being of the order of $7\,\mathrm{km\,s^{-1}}$ for low Earth orbit, it is obviously critical to have accurate knowledge of the measurement geometry to correctly estimate the non-geophysical component. The contribution due to ocean waves is, however, also an or-

der of magnitude larger than the expected TSCV contribution (Nouguier et al., 2018) and must also be precisely estimated using an accurate sea state description.

The Sea surface KInematics Multiscale monitoring (SKIM) satellite mission has been designed to address all these requirements and provide direct global-coverage measurements of TSCV. It is based on the combination of two instruments, the SKIM Ka-band Radar (SKaR), a phase-resolved SWIM-like conically scanning radar providing simultaneous Ka-band observations of sea state and DFS, and a state-of-the-art nadir altimeter providing the sea surface elevation observations necessary to control the SKaR acquisition geometry with sufficient accuracy but also significant wave height and wind speed observations.

SKIM was preselected as one of the two candidate missions for the European Space Agency (ESA) 9th Earth Explorer. As part of the detailed design and feasibility (phase A) studies, ESA funded a dedicated measurement campaign, DRIFT4SKIM, which was organized from 21 to 27 November 2018 off the French Atlantic coast in an area with sea states characteristic of the open ocean and a well-known tide-dominated current regime, monitored by a two-site 12 MHz high-frequency radar system (Ardhuin et al., 2009; Sentchev et al., 2013). A range of in situ instruments (surface current drifters, drifting and moored wave-measuring buoys), as well as two airborne Doppler radars operating in the Ku (KuROS – Ku-Band Radar for Observation of Surfaces) and Ka (KaRADOC – Ka-band RADar for Ocean Current) bands, were deployed. The campaign goals were to do the following:

- demonstrate how the non-geophysical contribution $V_{\mathrm{NG}}$ to the DFS can be estimated from the motion of the platform carrying the radar, the antenna diagram properties, and the azimuth and incidence angle dependencies of the radar cross section;

- explore the geophysical component $V_{\mathrm{GD}}$ and its decomposition as a sum of contributions due to currents and waves, $V_{\mathrm{CD}}$ and $V_{\mathrm{WD}}$ (Nouguier et al., 2018); and

- validate the Radar Sensing Satellite Simulator (Nouguier, 2019) and its capability to simulate airborne configurations.

As highlighted in Fig. 1, the viewing geometry of an airborne system is vastly different from that of a satellite system, with a much smaller footprint and incidence angle variations at scales comparable to the wavelength of the dominant ocean waves. Another obvious difference is the stability of the platform and its velocity, $7\,\mathrm{km\,s^{-1}}$ for low Earth orbit and around $120\,\mathrm{m\,s^{-1}}$ for the ATR-42 aircraft used here. As a result, transposing the performance of an airborne system to a satellite system requires a thorough analysis, supplemented by carefully designed and validated simulation tools. Performing this analysis is, however, worthwhile, as it leads

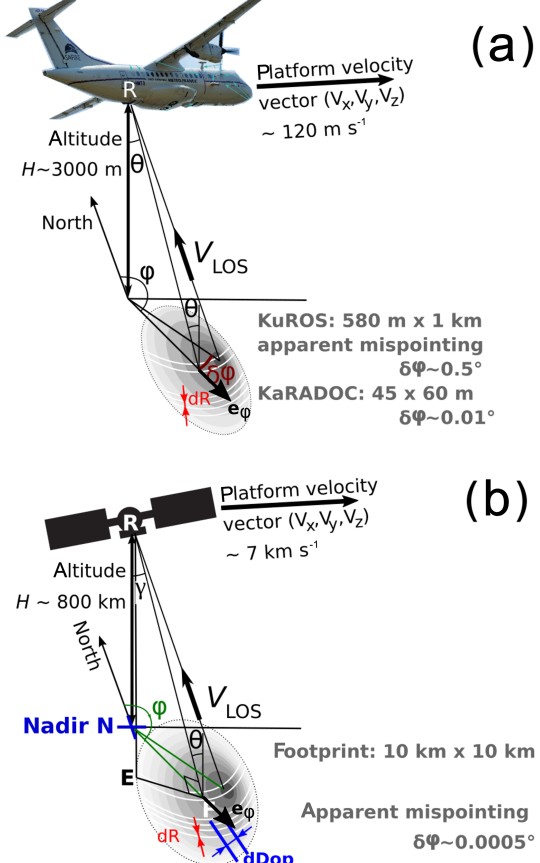

**Figure 1. (a)** Schematic of the ATR-42 and KuROS instrument with a definition of viewing angles, azimuth $\varphi$ and incidence angle $\theta$. **(b)** A comparison with the SKIM viewing geometry. The unit vector $e_\varphi$ TS9 is the projection on the horizontal of the line-of-sight direction vector. The variation of surface backscatter across the footprint and as a function of azimuth $\varphi$, which causes the effective mispointing $\delta\varphi$, is represented as grey shading. In the KuROS data, each measurement is integrated in azimuth across the antenna lobe. In the case of SKIM, the use of unfocused SAR processing allows for the separation of echoes in the azimuth direction with a resolution of dDop $\simeq 300$ m.

one to develop valuable insight into the instrument imaging principle and design trade-offs.

This article is intended to provide an overview of the DRIFT4SKIM campaign data and a first discussion of their implications for the emerging field of near-nadir Doppler radar observations of TSCV. It is structured as follows: the principle of the pulse-pair measurements and the different contributions to the observed DFS are detailed in Sect. 2 and Appendix A. Section 3 gives a brief account of the fieldwork performed and conditions encountered during the campaign. The results of the airborne measurements are presented in Sect. 4. Results and implications for SKIM are then discussed in Sect. 5. Conclusions and perspectives follow in Sect. 6.

## 2 Near-nadir radar Doppler measurements of ocean velocities: theory

Shipborne Doppler measurements of ocean currents are routinely performed using so-called vessel-mounted acoustic Doppler current profilers (VMADCPs; see, for instance, Rossby et al., 2019). Some of the data processing concepts transpose directly to the spaceborne context: the raw DFS signal contains a large non-geophysical contribution due to the platform motion, which must be estimated from ancillary sensors and compensated for. The accuracy of the final geophysical product is practically set by the accuracy of the non-geophysical velocity estimation and correction procedure. In the VMADCP context, however, the backscattering elements responsible for the production of the acoustic return signal (particulate suspended matter, zooplanktonic organisms) are passive and accurately follow the water mass. This does not carry over in the electromagnetic case: here, the return signal is produced by the interaction of the transmitted signal with the roughness elements of the sea surface, which move with respect to the water mass with an intrinsic phase velocity that is an order of magnitude larger than typical ocean currents. This effect is, for instance, well known in the ground-based HF radar current measurement context (Stewart and Joy, 1974) and must also be compensated for.

In our case, the measurement geometry is represented in Fig. 1, and the line-of-sight Doppler velocity $V_{\text{LOS}}$ looking towards incidence angle $\theta$ and azimuth $\varphi$ (in this paper, line-of-sight DV contributions are denoted by $V$, and the corresponding horizontal velocity contributions are denoted by $U$) is the sum of the projection of a horizontal current contribution $U_{\text{CD}}(\varphi)$, a wave-induced contribution $V_{\text{WD}}(\theta, \varphi)$ and a non-geophysical contribution $V_{\text{NG}}(\theta, \varphi)$. The equation that permits the retrieval of the TSCV contribution $U_{\text{CD}}(\varphi)$ from the raw measured $V_{\text{LOS}}$ can be written as

$$U_{\text{CD}}(\varphi) = \frac{[V_{\text{LOS}}(\theta, \varphi) - V_{\text{NG}}(\theta, \varphi) - V_{\text{WD}}(\theta, \varphi)]}{\sin\theta}. \quad (1)$$

The aim of this section is to provide a detailed analysis of the different terms of this expression. The non-geophysical contribution $V_{\text{NG}}$ is discussed in Sect. 2.1 and Appendix A. The wave Doppler contribution is discussed in Sect. 2.2. A brief summary of the measurement error budget is finally provided in Sect. 2.3.

### 2.1 Non-geophysical velocity $V_{\text{NG}}$

As mentioned above, the accuracy of shipborne acoustic Doppler current measurements is affected in a dominant way by the platform motion compensation process. In the spaceborne context, the platform velocity is almost 3 orders of magnitude larger (7000 m s$^{-1}$ vs. 10 m s$^{-1}$ for shipborne measurements). The accuracy requirements are thus tremendously exacerbated, and particular attention must be paid to the detailed effects of the antenna radiation diagram and the

**Table 1.** KuROS and KaRADOC antenna radiation diagram characteristics. All angles are in degrees. See Appendix B for the definitions of $\alpha$ and $\beta$.

| Instrument | KuROS | KaRADOC |
|---|---|---|
| Polarization | HH | HH |
| Azimuth one-way beamwidth ($\alpha_{-3\,\mathrm{dB}}$) | 15.0 | 1.85 |
| Elevation one-way beamwidth ($\beta_{-3\,\mathrm{dB}}$) | 22.6 | 1.20 |
| Boresight elevation ($\beta^0$) | 11.8 | 12.1 |
| Boresight azimuth (°) | $\sim 0$ | $-0.05$ |

sea surface normalized radar cross section (NRCS) variations with space and observation azimuth. A detailed discussion of these effects is given in Appendix A.

In summary, in the case of a sufficiently narrow radiation diagram, $V_{\mathrm{NG}}$ can be approximated as the radar carrier velocity projected on an effective look direction. This effective look direction differs from the geometric boresight direction by an effective azimuthal mispointing $\delta\varphi$ due to the finite antenna beamwidth combined with the variations of NRCS within the radar footprint, as well as by an effective incidence angle mispointing $\delta\theta$ due to radar timing or surface-tracking errors.

The beamwidth at the working incidence angle is thus a very important parameter of a radar intended for TSCV measurements. Table 1 summarizes the parameters of the KuROS and KaRADOC antennas. For KuROS they have been determined following the procedure detailed in Appendix B. For KaRADOC, they are the result of anechoic chamber measurements (Appendix C). As discussed in Appendix B, these parameters describe the antenna radiation diagrams when expressed as functions of variables, $\alpha$ and $\beta$, which do not coincide with azimuth and incidence angle. In the case of constant-altitude flight and near-nadir observations with the antenna looking towards azimuth $\varphi_{\mathrm{b}}$, one can, however, obtain a Gaussian approximation of the one-way radiation diagram as

$$G \simeq \exp\left[-\frac{(\varphi - \varphi_{\mathrm{b}})^2}{2}\left[\frac{\sin^2(\theta)}{\sigma_\alpha^2} + \frac{(\beta_0 - \tan(\theta))\tan(\theta)}{\sigma_\beta^2}\right]\right], \quad (2)$$

where $\sigma_\alpha = \alpha_{-3\,\mathrm{dB}}/\sqrt{8\log(2)}$ and $\sigma_\beta = \beta_{-3\,\mathrm{dB}}/\sqrt{8\log(2)}$. For 12° observations the second term in the exponential can safely be neglected, and the effective azimuthal beamwidth can be estimated as

$$\varphi_{-3\,\mathrm{dB}} = \frac{\alpha_{-3\,\mathrm{dB}}}{\sin(\theta)}. \quad (3)$$

When projected on the ground, $\varphi_{-3\,\mathrm{dB}}$ is thus larger than $\alpha_{-3\,\mathrm{dB}}$ by a factor $1/\sin(\theta)$, equal to 4.8 for 12° measurements. Provided that the beam is not too wide, the Gaussian approximation in Eq. (A29) of $G$ as a function of $\varphi$ can then be used with the parameter

$$\sigma_\varphi \simeq \alpha_{-3\,\mathrm{dB}}\Big/\left[\sin\theta\sqrt{8\log(2)}\right]. \quad (4)$$

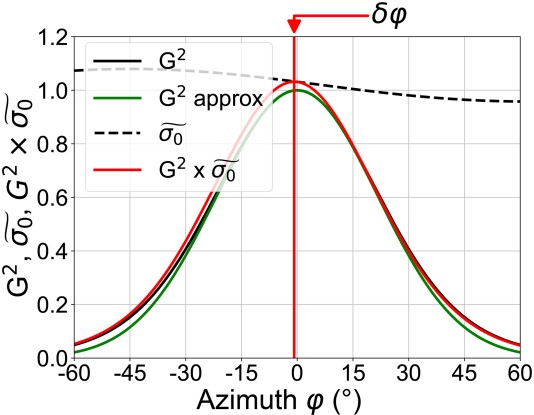

**Figure 2.** KuROS azimuth integral weight at $\theta = 12°$ for a north-facing ($\varphi_{\mathrm{b}} = 0°$) antenna (black), Gaussian approximation (Eq. A29) (green) and variation of $\widetilde{\sigma^0}$ for a typical $11\,\mathrm{m\,s^{-1}}$ TS10 wind from 140° (dashed black). The peak of the $\widetilde{\sigma^0}\,G^2$ product (red) is shifted with respect to the peak of $G^2$ by $\delta\varphi \simeq -0.81°$.

Due to the width of the azimuthal aperture, the NRCS-weighted line-of-sight azimuth $\varphi_{\mathrm{a}}$ can differ from the boresight azimuth $\varphi_{\mathrm{b}}$ by a mispointing angle $\delta\varphi$. Expressions for $\delta\varphi$ are obtained in Appendix A in the two limiting cases of slow linear and fast sinusoidal variations of the ocean surface NRCS with respect to azimuth. In the slow variation case, $\delta\varphi$ is obtained as

$$\delta\varphi = \varphi_{\mathrm{a}} - \varphi_{\mathrm{b}} = \frac{1}{2}\frac{\sigma_\alpha^2}{\sin^2\theta}\frac{1}{\sigma^0}\frac{\partial\sigma^0}{\partial\varphi}. \quad (5)$$

Denoting by $\varphi_{\mathrm{t}}$ the flight track azimuth and $V_{\mathrm{p}}$ the along-track flight velocity, the spurious azimuth gradient Doppler contribution to the DV caused by the mispointing reads

$$U_{\mathrm{AGD}} = \sin(\varphi_{\mathrm{b}} - \varphi_{\mathrm{t}})\frac{V_{\mathrm{p}}}{2}\frac{\sigma_\alpha^2}{\sin^2\theta}\frac{1}{\sigma^0}\frac{\partial\sigma^0}{\partial\varphi}. \quad (6)$$

As an example, Fig. 2 shows the variations of the two-way antenna radiation diagram $G^2$, its Gaussian approximation and the $G^2\sigma^0$ (see Eq. A11) product as a function of azimuth at a 12° incidence angle for a northward-looking KuROS antenna ($\varphi_{\mathrm{b}} = 0°$) using $\sigma^0$ data from the DRIFT4SKIM campaign on 22 November 2018. The effect of the wind-induced azimuthal gradient of $\sigma^0$ is to shift the effective radiation diagram towards the brighter upwind and downwind directions, with an apparent pointing azimuth $\varphi_{\mathrm{a}}$. The shift induced in this case is $\delta\varphi = \varphi_{\mathrm{a}} - \varphi_{\mathrm{b}} = -0.81° = -15 \times 10^{-3}\,\mathrm{rad}$. For comparison (see Sect. 2.3 and Table 2), the pointing accuracy required to achieve a $15\,\mathrm{cm\,s^{-1}}$ error on the horizontal current in the airborne configuration is 1.2 mrad.

Here, it is important to note that KuROS was not specifically designed for this experiment but primarily as a calibration and validation instrument for the CFOSAT mission, which required a broad radiation diagram. Though the analysis of the KuROS data helped uncover many interesting

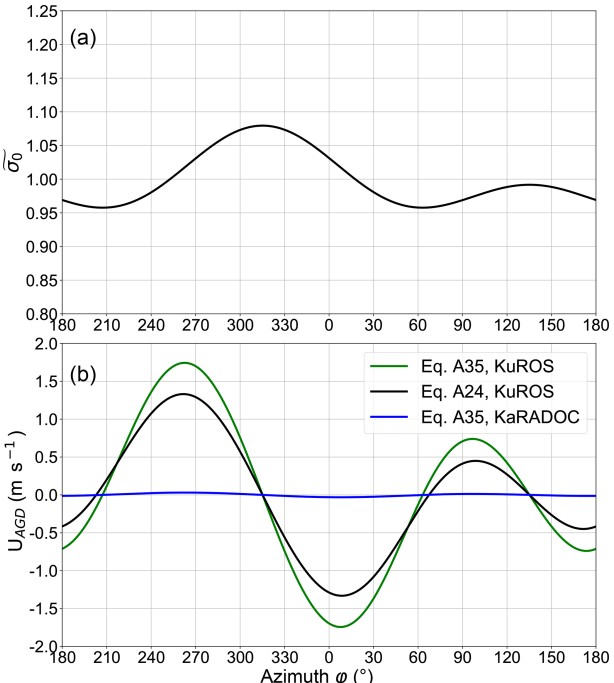

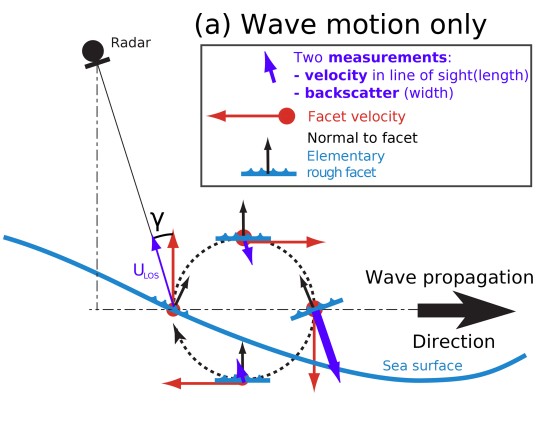

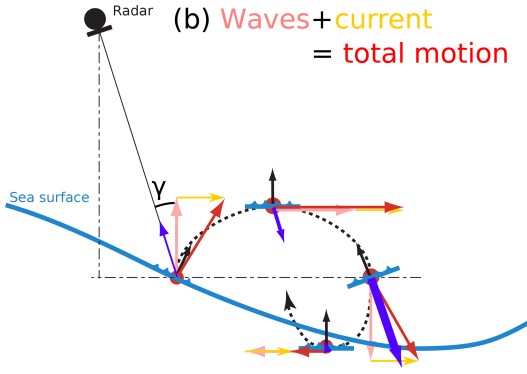

**Figure 3. (a)** Example of azimuthal variation of $\widetilde{\sigma^0}$ at a 12° incidence angle, corresponding to the 22 November case (11 m s$^{-1}$ wind from 140°) discussed in Sect. 3, and **(b)** associated spurious velocity $U_{AGD}$ as a function of look azimuth $\varphi_b$ in the case of a port-looking antenna mounted on a platform in constant-altitude flight at 120 m s$^{-1}$. For the KuROS case, the green line shows the result of the approximation in Eq. (A35), and the black line shows the result of the full azimuthal integration in Eq. (A24). The blue line represents the result of Eq. (A35) for KaRADOC using the same $\widetilde{\sigma^0}$ as in the Ku-band case.

**Figure 4.** Schematic of **(a)** wave and **(b)** wave and current contributions to Doppler velocities at the scale of elementary facets. These small-scale processes are averaged over the radar field of view, and a mean velocity signal emerges due to the correlation of surface brightness and velocities in the wave field.

effects relevant to Doppler observations of the sea surface, its design was not fully appropriate to validate the inversion of the geophysical velocities, for which the pencil-beam antenna diagram of KaRADOC was better suited.

Figure 3a shows a typical example of the azimuthal variation of $\sigma^0$ at a 12° incidence angle for the Ku band. As expected for near-nadir measurements (Chapron et al., 2002; Munk, 2008; Chu et al., 2012), the NRCS is largest in the downwind look direction ($\varphi = 320°$), has a secondary peak

in the upwind direction and is weakest in the crosswind look directions. Figure 3b shows the corresponding $U_{AGD}$ contribution for the KuROS and KaRADOC cases using an aircraft velocity $V_p = 120$ m s$^{-1}$ and the Ku-band NRCS fit for both instruments (this is a reasonable assumption for order-of-magnitude estimates). As detailed in Appendix A, Eqs. (5)

and (6) only apply for a narrow beam when projected on the ground, which is not a very good approximation for the KuROS case, even at a 12° incidence angle. As shown in Fig. 2, the Gaussian approximation for the antenna diagram as a function of $\varphi$ gives a distribution that is too narrow and

does not properly take into account the azimuthal integra-

tion, leading to an overestimation of $U_{AGD}$. It is clear, however, that even the more exact Eq. (A24) gives very large correction magnitudes, in excess of 1.2 m s$^{-1}$ in some azimuth ranges.

Because the azimuth gradient $U_{AGD}$ contribution to the observed DV is proportional to $V_p \sigma_\varphi^2$, this effect is much larger (and correcting it is correspondingly more demanding in terms of antenna characterization) for KuROS than for KaRADOC or DopplerScatt (Rodríguez et al., 2018) thanks

to their narrow azimuthal beam aperture. Another remark is that the approximate expression in Eq. (A35), though it gives the appropriate dependency of $U_{AGD}$ with respect to look azimuth, tends to overpredict its magnitude as the widening associated with the ground projection saturates for broad

beams.

Although the relative variations $\partial_\varphi \sigma^0 / \sigma^0$ are larger for larger incidence angles, this is more than compensated for by the $1/\sin^2\theta$ reduction in azimuthal diversity across the footprint. This effect can thus be neglected for much higher

incidence angles (Rodríguez et al., 2018).

## 2.2   Geophysical velocity $U_{GD}$: waves and current Doppler velocities

The geophysical part of the DFS measured by a microwave radar over the ocean, using both along-track interferometry and Doppler centroid techniques, emerges from the average over the instrument field of view (FOV) of the backscatter-weighted line-of-sight projection of the surface velocity, as illustrated in Fig. 4.

In the well-understood case of decametric electromagnetic waves interacting with the sea surface at grazing incidence, the interaction is dominated by the Bragg coherent backscattering mechanism (Crombie, 1955), in which the backscattered field reflects the properties (amplitude, phase speed) of a very finely selected component of the sea state, namely that whose wave vector is precisely equal to the so-called Ewald vector, the difference between the wave vectors of the scattered and incident electromagnetic waves. Exploiting the deviation of the phase speed of this sea state component from its theoretical value is the principle of the HF radars operationally used to measure ocean TSCV in coastal areas (Barrick et al., 1974; Stewart and Joy, 1974).

In the case of the near-nadir interaction of microwaves with the sea surface, which is the configuration considered for SKIM and used by the AirSWOT, KuROS and KaRADOC airborne instruments, this mental picture must be adapted: the Bragg scattering mechanism is not dominant, and the main contribution comes from quasi-specular reflections on those facets of the sea surface which are normal to the Ewald vector. The backscattering cross section of the sea surface and DFS in this case do not depend on the properties of a single Fourier component of the sea state but on the probability density function of the sea surface slope, which is a complex functional of its entire directional spectrum.

As discussed by Nouguier et al. (2018), who applied it to the analysis of AirSWOT NRCS and DFS data collected during the Gulf of Mexico LASER experiment in 2016, the theoretical framework appropriate for this configuration is the Kirchhoff approximation (Beckmann and Spizzichino, 1987). In this approximation, the geophysical DFS $\omega_{GD}$ can be expressed as

$$\omega_{GD} = -i \left. \frac{\partial_\tau C}{C(\tau)} \right|_{\tau=0}, \tag{7}$$

where $C(\tau)$ is the temporal covariance function of the ensemble-averaged electromagnetic field backscattered in the direction of the radar.

Assuming Gaussian statistics for the sea surface, introducing $\rho(\boldsymbol{\xi}, \tau) = \langle \eta(\boldsymbol{x} + \boldsymbol{\xi}, t + \tau)\eta(\boldsymbol{x}, t)\rangle$, the space–time covariance function of the sea surface elevation, and $\boxed{\text{TS1}}\, Q_H = -\frac{4\pi}{\lambda} \sin(\theta) \boldsymbol{e}_\varphi$ and $Q_z = \frac{4\pi}{\lambda} \cos(\theta)$, the horizontal and vertical components of the Ewald vector (with $\lambda$ the radar wave-

length), one obtains $C(\tau)$ and $\partial_\tau C$ as

$$C(\tau) = \int e^{i Q_H \cdot \boldsymbol{\xi}} \left[ e^{Q_z^2(\rho(\boldsymbol{\xi}, \tau) - \rho(0,0))} - e^{-Q_z^2 \rho(0,0)} \right] \mathrm{d}\boldsymbol{\xi}, \tag{8}$$

$$\partial_\tau C = Q_z^2 \int \partial_\tau \rho(\boldsymbol{\xi}, \tau) e^{i Q_H \cdot \boldsymbol{\xi}} e^{Q_z^2(\rho(\boldsymbol{\xi}, \tau) - \rho(0,0))} \mathrm{d}\boldsymbol{\xi}. \tag{9}$$

The clear upwind–downwind asymmetry of $\sigma^0$ observed in the DRIFT4SKIM radar observations (see Fig. 10) shows that the Gaussian assumption, which is unable to describe such skewness-related effects, is clearly questionable. It is, however, the only practical option, as going further would require prescriptions for the higher-order statistics of the sea surface, which are at present not available.

The occurrence of $\rho$ as the argument of an exponential in these integrals renders further analytical progress difficult (see, however, Nouguier et al., 2011). Approximate expressions can, however, be obtained by performing a Taylor expansion of $\rho$ in the neighborhood of the origin. This results in a Gaussian approximation of the integrand. The integrals can be readily evaluated, yielding (denoting by "·" the usual matrix product)

$$\omega_{GD} \simeq -\boldsymbol{Q_H}^T \cdot \left[\nabla_{\xi\xi}\rho\right]^{-1} \cdot \partial_\tau \nabla_\xi \rho. \tag{10}$$

The derivatives of $\rho$ are taken at $\boldsymbol{\xi} = 0$ and $\tau = 0$, and they can be expressed as moments of the directional sea state spectrum $S_d(\boldsymbol{k})$ as

$$\partial_\tau \nabla_\xi \rho = \text{msv}, \ \nabla_{\xi\xi}\rho = -\mathbf{Mss}, \tag{11}$$

where, in the notation of Nouguier et al. (2018), msv stands for the mean slope velocity and $\mathbf{Mss}$ for the mean square slope matrix,

$$\text{msv} = \begin{bmatrix} \text{mss}_{xt} \\ \text{mss}_{yt} \end{bmatrix}, \ \mathbf{Mss} = \begin{bmatrix} \text{mss}_{xx} & \text{mss}_{xy} \\ \text{mss}_{yx} & \text{mss}_{yy} \end{bmatrix}, \tag{12}$$

with

$$\text{mss}_{x^\alpha y^\beta t^\gamma} = 2 \int_{\mathbb{R}^2} k_x^\alpha k_y^\beta \omega^\gamma S_d(\boldsymbol{k}) \mathrm{d}\boldsymbol{k}. \tag{13}$$

The surface current enters through its effect on the dispersion relation of surface waves $\omega(\boldsymbol{k})$. In the presence of a vertically homogeneous current $\boldsymbol{U}$ (a detailed discussion of the effect of shear can be found in Kirby and Chen, 1989),

$$\omega(\boldsymbol{k}) = \boldsymbol{k} \cdot \boldsymbol{U} + \omega_0(|\boldsymbol{k}|), \tag{14}$$

where

$$\omega_0(|\boldsymbol{k}|) = \sqrt{g|\boldsymbol{k}| \left(1 + |\boldsymbol{k}|^2/\kappa_M^2\right)} \tag{15}$$

is the dispersion relation of gravity–capillary waves in deep water, with $\boldsymbol{k}$ the wave vector and $\kappa_M = 363.2\,\text{rad}\,\text{m}^{-1}$ the wavenumber corresponding to the gravity–capillary regime

Please note the remarks at the end of the manuscript.

transition. Introducing this expression in Eq. (10), and defining $\text{msv}_0$ as the spectral moment obtained using the dispersion relation of Eq. (15) in Eq. (13), one obtains the approximate DFS as

$$\omega_{\text{GD}} = Q_{\text{H}}^{T} \cdot \left[ \mathbf{Mss}^{-1} \cdot \text{msv}_0 + U \right] \tag{16}$$

and the corresponding $V_{\text{GD}}$ as

$$V_{\text{GD}} = -\sin(\theta)\, \boldsymbol{e}_{\varphi} \cdot \left[ \mathbf{Mss}^{-1} \cdot \text{msv}_0 + U \right]. \tag{17}$$

While clearly oversimplified (it is, for instance, independent of the electromagnetic wavelength, which is known to have a significant influence on $\sigma^0$), this expression has a definite pedagogical interest, as it allows one to distinguish a number of interesting features.

- The raw velocity projection $V_{\text{GD}}$ accessible to Doppler radar instruments is composed of a "genuine" current component $V_{\text{CD}}$, equal to the projection of the TSCV along the radar line of sight, plus a wave Doppler component $V_{\text{WD}}$ induced by the natural motion of the sea surface.

- This $V_{\text{WD}}$ component involves sea surface statistics of two different natures: the mean slope velocity vector $\text{msv}_0$ and the mean squared slope matrix $\mathbf{Mss}$. To this order of approximation it can be seen as the projection along the radar line of sight of the constant vector $\mathbf{Mss}^{-1} \cdot \text{msv}_0$. In the rest of this article, $M_{\text{WD}}$ denotes the norm of this vector.

- As noted in Nouguier et al. (2018), $\text{msv}_0$ is equal to one-half the surface Stokes drift velocity of deepwater waves $U_S^{\infty}$. As noted in Nouguier et al. (2016), the effective mean squared slope matrix $\mathbf{Mss}$ components ($\text{mss}_{\text{shape}}$), accounting for the electromagnetic filtering effect and part of the non-Gaussianity of the sea surface statistics, can be obtained from the derivatives of $\sigma^0$ as a function of incidence angle for different azimuths.

- In simple cases represented by parametric spectral forms such as the Elfouhaily et al. (1997) spectrum used in this work, $\text{msv}_0$ and the eigenvectors of $\mathbf{Mss}$ are aligned with the downwind direction, and the $V_{\text{WD}} = -G_{\text{D}} \sin(\theta)\, \boldsymbol{e}_{\varphi} \cdot U_S^{\infty}$ relation proposed in Chapron et al. (2005) is recovered, with $G_{\text{D}} = \frac{1}{\text{mss}_{\text{shape}}}$.

- Both these statistics are, however, known to be influenced by waves at all scales. The asymptotic behaviors of the weighting factors as functions of the surface wave wavenumber in the gravity wave range are $k^{3/2}$ and $k^2$ for the $\text{msv}_0$ and $\mathbf{Mss}$ terms, respectively, while the parametric spectrum of Elfouhaily et al. (1997), used in this work, decays as $k^{-3}$, leading to a logarithmic divergence for the $\mathbf{Mss}$ components and a slow convergence of the $\text{msv}_0$ components at high wavenumbers.

- The $\mathbf{Mss}$ components are sensitive to the detailed shape of the spectrum up until the short capillary wave roll-off or to the electromagnetic cutoff, whichever is reached first.

- Estimating these terms requires knowledge of all the components of the sea state: the long gravity wave range can be measured (either in situ, as during the DRIFT4SKIM campaign, or using the radar measurements themselves as intended in the SKIM context), but the high-wavenumber range cannot be neglected, and its effect must be accounted for, for instance, through the use of a parametric spectral form.

- The $\text{msv}_0$ vector appears as a multiplicative factor, to which the inverse of the $\mathbf{Mss}$ matrix is applied. These terms thus have opposing influences on the final result: modifications of the sea spectrum, which tend to increase the weight of small-scale components, increase the mean slope velocity but also, and rather more, the mean squared slope by which it is divided. A certain degree of stability of the end result is thus likely.

- On a similarly reassuring note, whereas the low-wavenumber part of the spectrum is affected by swell systems of remote origin that have arbitrary orientations, the short waves represented by the parametric tail of the spectrum are known to be aligned with the wind direction and to depend on local variables only (wind strength and direction, fetch).

Figure 5 gives orders of magnitude for the natural range of variability of the different factors thus isolated. Figure 5a shows the variability of the Stokes drift velocity estimated following Kenyon (1969) and Ardhuin et al. (2009) using wind and directional wave measurements collected from 2010 to 2017 at Ocean Station Papa. Even though $U_S$ is highly correlated with the wind speed with a Pearson's linear correlation coefficient of 0.85, a strong dependence on the long-wavelength part of the spectrum, for which Hs is a proxy, definitely has to be accounted for.

Figure 5b, taken from Nouguier et al. (2016), instead shows the dependence on wind speed and Hs of Ku- and Ka-band effective mean squared slope $\text{mss}_{\text{shape}}$ retrieved from the GPM CE4 satellite measurements. The variability is even more strongly dominated by the dependence on wind speed, the variability due to the long-wavelength part of the spectrum being much smaller. These measurements very clearly show the filtering effect of the electromagnetic wavelength and are a clear warning that Eq. (17), suggestive though it is, should be considered with caution.

Finally, Fig. 5c and d show the magnitude of the horizontal $U_{\text{WD}}$ component as a function of wind speed and Hs, estimated by numerically evaluating the integrals of Eqs. (8) and (9) for $C(\tau)$ and $\partial_{\tau} C$ using the numerical tools of Nouguier et al. (2011) on the basis of long-wavelength spectra extracted from global runs of the WAVEWATCH III model

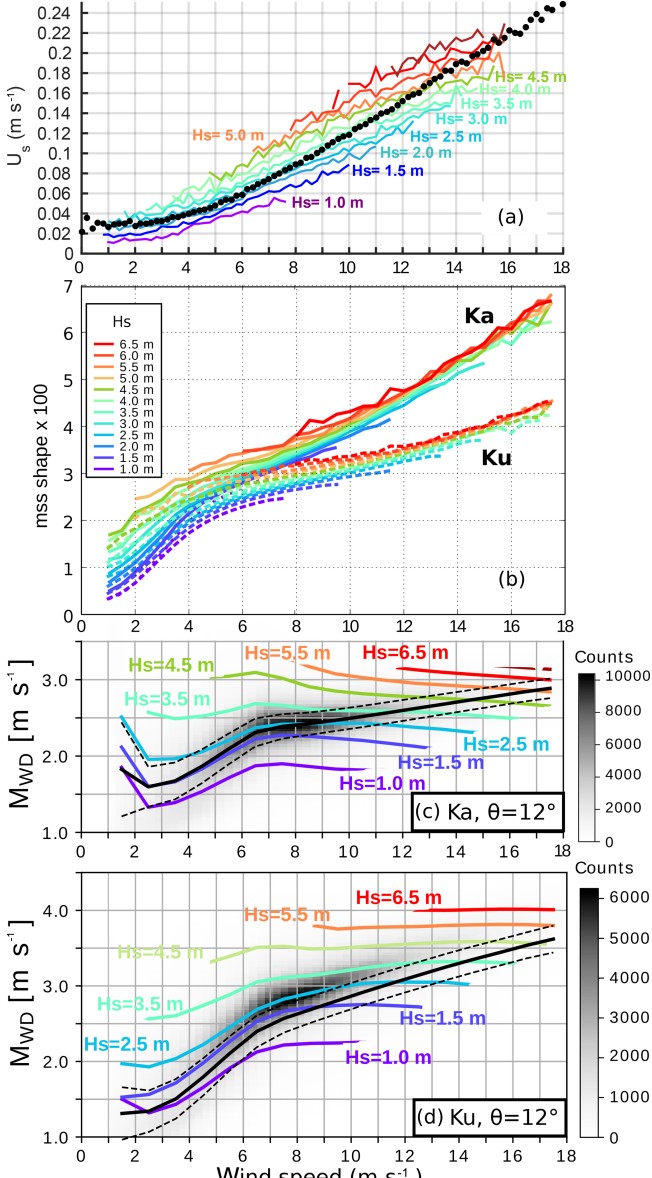

**Figure 5.** Computed variability of the Stokes drift velocity, the diffraction-effective mean square slope $\text{mss}_{\text{shape}}$ and the wave Doppler velocity magnitude $M_{\text{WD}}$. **(a)** 2010–2017 statistics of Stokes drift magnitude at Ocean Station Papa, computed using the buoy wind speed data and wave data from the nearby WMO buoy 46 246, maintained by the University of Washington (Thomson et al., 2013). **(b)** $\text{mss}_{\text{shape}}$ estimated from GPM satellite backscatter using modeled co-located wind speed and wave height; reproduced from Nouguier et al. (2018). **(c, d)** Statistics of the Ka- and Ku-band $M_{\text{WD}}$, computed using the theoretical model of Nouguier et al. (2011) for ocean wave spectra modeled over the global ocean using the WAVEWATCH III model (Stopa et al., 2016) and plotted as a function of the wind speed. The colored curves show the median value for different classes of wave height for a given wind speed; each curve is separated by 0.5 m in panels **(a)** and **(b)** and by 1 m in panels **(c)** and **(d)**. In **(c)** and **(d)**, the grey shading represents the histogram of the computed $M_{\text{WD}}$ values in the global simulation.

(Stopa et al., 2016), completed in the high-wavenumber range by Elfouhaily et al. (1997) spectral tails. The shading in the background represents the histogram of the different (wind speed, $M_{\text{WD}}$) pairs. As could be hoped for, the opposing influences of the wind speed on $\text{msv}_0$ and **Mss** tend to counteract each other, greatly reducing the range of variability of $M_{\text{WD}}$ with wind. This effect appears stronger in the Ka rather than the Ku band, possibly due to the saturation of the Ku-band $\text{mss}_{\text{shape}}$ at high winds. These figures show a strong remaining impact of the long-wavelength waves, which clearly must be accounted for. As wind speed and significant height are highly correlated variables, the frequently encountered situations fall in a quite narrow interval $M_{\text{WD}} \simeq C_0$, with $C_0 \simeq 2.6$ and $C_0 \simeq 2.2\,\text{m}\,\text{s}^{-1}$ in the Ku and Ka band, respectively. In other words, most of the variability of $U_{\text{WD}}$ is controlled by the directionality effect, and the magnitude $M_{\text{WD}}$ is a weakly varying function of the wind, the wave age and the presence of swell (see also Yurovsky et al., 2019; LOPS, 2019).

A final remark is that, though these general patterns can probably be assumed to be robust, the precise numerical values depend on the parametric spectral shapes which have been used to fill the high-wavenumber range of the spectra. Changing, for instance, the high-wavenumber azimuthal spreading functions, which are for the moment not very well constrained observationally, has different impacts on the $\text{msv}_0$ and **Mss** terms and can thus be expected to marginally change the numbers.

### 2.3 Error budget

Considering the errors on the different terms to be independent, developing Eq. (1) allows one to derive the error variance of Doppler radar measurements of the TSCV as

$$\text{Var}\left(\delta U_{\text{CD}}\right) = \left(\frac{U_{\text{CD}}}{\tan\theta}\right)^2 \text{Var}(\delta\theta) + \frac{\text{Var}\left(\delta V_{\text{LOS}}\right)}{\sin^2\theta}$$
$$+ \frac{\text{Var}\left(\delta V_{\text{NG}}\right)}{\sin^2\theta} + \text{Var}\left(\delta U_{\text{WD}}\right). \qquad (18)$$

As a first step, four contributions to the uncertainty on $U_{\text{CD}}$ can thus be isolated, with different origins.

- A first part corresponds merely to the error caused by imperfect knowledge of the projection angle between the TSCV and the line of sight. Its order of magnitude is controlled by the TSCV, and it is thus negligible with respect to similar terms that involve the platform velocity.

- The second term corresponds to the random error in the DFS measurements and subsumes the dependence on the signal-to-noise ratio, antenna beamwidth, orientation of the boresight with respect to the platform velocity vector and algorithmic choices. A very thorough analysis of this term can be found in Rodriguez (2018).

The standard deviation of the raw DV signal carries over to the end result, multiplied by a $1/\sin\theta$ factor of the order of 5 for $\theta = 12°$.

– The third term corresponds to the error caused by mismatches between the actual platform motion contribution to $V_{LOS}$ and the estimate computed from the ancillary sensors. The order of magnitude of this term is set by the (very large) platform velocity. It is by far the largest.

– The fourth and final term corresponds to the uncertainty on the wave Doppler removal stage. Errors in the $U_{WD}$ model carry directly over to the $U_{CD}$ estimates.

The third term dominates the overall error budget and must be further analyzed. It is convenient for that purpose to start from Eq. (A23), which gives the expression of the beam direction vector, and use the platform velocity components in the local (north–east–down) frame at the observation point. Neglecting terms involving the vertical velocity of the platform and introducing the difference between the boresight and flight track azimuths $\psi = \varphi_b - \varphi_t$, one obtains the consolidated error budget as

$$
\begin{aligned}
\mathrm{Var}(\delta U_{CD}) = {} & \frac{\mathrm{Var}(\delta V_{LOS})}{\sin^2\theta} + \mathrm{Var}(\delta U_{WD}) + \frac{\mathrm{Var}(\delta V_D)}{\tan^2(\theta)} \\
& + \mathrm{Var}(\delta V_N)\cos^2(\phi_b) + \mathrm{Var}(\delta V_E)\sin^2(\phi_b) \\
& + V_P^2 \left[ \frac{\cos^2(\psi)}{\tan^2(\theta)}\mathrm{Var}(\delta\theta) + \sin^2(\psi)\mathrm{Var}(\delta\varphi) \right].
\end{aligned} \tag{19}
$$

This equation summarizes the dependence of the overall $U_{CD}$ error on the errors introduced by the Doppler measurements, the $U_{WD}$ model, the individual platform velocity components, and the incidence angle and azimuth mispointing errors.

As an illustration, Table 2 summarizes the requirements that have to be met to keep the standard deviation of each of the seven terms below $0.15\,\mathrm{m\,s^{-1}}$, ensuring a $0.4\,\mathrm{m\,s^{-1}}$ standard deviation for $U_{CD}$. The requirement for $\theta$ is translated to the corresponding altitude-tracking accuracy requirement for the KuROS and SKIM configurations. The requirements for linear velocity components are stringent but can be reached using current-day technology. The requirement for altitude accuracy is easily within the specifications of the SKIM nadir altimeter payload but definitely out of reach of KuROS. The KuROS data could, however, be analyzed in the cross-track-looking configurations for which this requirement does not apply. The requirement for azimuthal pointing accuracy is by far the most stringent. In the airborne case, it is met for the antenna boresight by the CE5 plane IMU, allowing a straightforward analysis of the KaRADOC data. In the KuROS case, however, it is exceeded by a factor of 10 by the mispointing induced by the azimuthal gradients of sea surface, which required the development of a specific data correction procedure. Finally, in the spaceborne case, it seems only achievable using a combination of high-end inertial measurements and data-driven analysis techniques.

## 3 Campaign overview

This section provides a general overview of the campaign. The location, timing and overall organization are described in Sect. 3.1, the environmental conditions encountered during the campaign are described in Sect. 3.2, and the two main instruments, the KuROS and KaRADOC airborne radars, are described in Sect. 3.3 and 3.4, respectively.

### 3.1 Campaign organization

The DRIFT4SKIM experiment differs from previous airborne Doppler radar campaigns (Martin et al., 2016; Rodríguez et al., 2018) in two important respects: in order to observe the effect of wave development on the geophysical Doppler velocity $U_{GD}$, it was performed in a midlatitude, eastern basin oceanic environment open to offshore swells. Also, given the campaign objectives of demonstrating the sensitivity of airborne radar Doppler measurements to the geophysical contributions of currents and waves, it comprised an extensive in situ component designed to have commonly accepted reference measurements for these parameters.

Fieldwork was performed in two areas (denoted by square boxes in Fig. 6) named the "offshore" area, centered on the Trèfle buoy (see below), and the "Keller Race" area to the north of the island of Ushant. Both locations are in the range of coverage of a two-site WERA (Gurgel et al., 1999) high-frequency radar system, operated by Service Hydrographique et Oceanographique de la Marine (Shom) and already used for several studies, in particular related to wave–current interactions (Ardhuin et al., 2009, 2012; Guimaraes et al., 2018).

Keller Race is an area with very strong horizontal gradients of the current (Sentchev et al., 2013). Although it is easy to show a strong effect of the current on the measured DFS, the spatial variability of the sea state is difficult to measure in situ, introducing uncertainties when combining $U_{CD} + U_{WD}$ in a forward model or using $U_{WD}$ estimates when retrieving $U_{CD}$ from the measured $U_{GD}$. The offshore area, on the other hand, was chosen for its spatial uniformity, being located far enough from the islands and with a near-uniform depth of 110 m. Only airborne data acquired over the offshore area are presented in this paper.

The week around spring tides in November 2018 was selected in order to allow for a wide range of current speeds (Fig. 7a).

The KuROS and KaRADOC radars were installed on an ATR-42 plane operated by the French institutional scientific flight facility, SAFIRE, which is equipped with an AIRINS™ GNSS-FOG INS providing position, pitch, roll and head-

**https://doi.org/10.5194/os-16-1-2020**

**Ocean Sci., 16, 1–32, 2020**

**Table 2.** Standard deviations of the different error terms in Eq. (19) necessary to achieve a $0.40\,\mathrm{m\,s^{-1}}$ standard deviation for $U_{\mathrm{CD}}$.

|  | KuROS 12° | KaRADOC 12° | SKIM 12° | SKIM 6° |
|---|---|---|---|---|
| $\delta V_{\mathrm{LOS}}$ $(\mathrm{m\,s^{-1}})$ | $3.1 \times 10^{-2}$ | $3.1 \times 10^{-2}$ | $3.1 \times 10^{-2}$ | $1.6 \times 10^{-2}$ |
| $\delta U_{\mathrm{WD}}$ $(\mathrm{m\,s^{-1}})$ | $15 \times 10^{-2}$ | $15 \times 10^{-2}$ | $15 \times 10^{-2}$ | $15 \times 10^{-2}$ |
| $\delta V_{\mathrm{N,E}}$ $(\mathrm{m\,s^{-1}})$ | $15 \times 10^{-2}$ | $15 \times 10^{-2}$ | $15 \times 10^{-2}$ | $15 \times 10^{-3}$ |
| $\delta V_{\mathrm{D}}$ $(\mathrm{m\,s^{-1}})$ | $3.2 \times 10^{-2}$ | $3.2 \times 10^{-2}$ | $3.2 \times 10^{-2}$ | $1.6 \times 10^{-2}$ |
| $\delta\theta$, up-track and down-track (rad) | $0.26 \times 10^{-3}$ | – | $4.5 \times 10^{-6}$ | $2.3 \times 10^{-6}$ |
| $\delta h$, up-track and down-track (m) | $17 \times 10^{-2}$ | – | $80 \times 10^{-2}$ | $2 \times 10^{-2}$ |
| $\delta\phi$, cross-track (rad) | $1.2 \times 10^{-3}$ | $1.2 \times 10^{-3}$ | $21 \times 10^{-6}$ | $21 \times 10^{-6}$ |

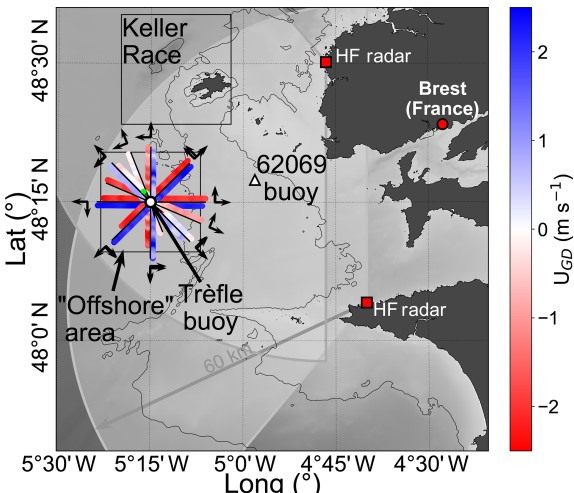

**Figure 6.** Location of the measurement campaign and in situ assets, including a map of the KaRADOC measurements of the geophysical Doppler velocity $U_{\mathrm{GD}}$ acquired on 22 November 2018.

ing information with stated tolerances of a few centimeters, 0.005, 0.005 and 0.01°, respectively.

Ground truth measurements comprised two permanent operational systems: the HF radar system mentioned previously, with an expected depth of measurement around 1 m (Stewart and Joy, 1974), and the Pierres Noires (WMO no. 62069) wave-measuring buoy. Dedicated instrumentation was also deployed for the campaign.

– The Trèfle buoy was moored at 5°15′ W, 48°15′ N at the center of the offshore area. This buoy monitored the surface current (Sutherland et al., 2016) and provided directional wave spectra (Fig. 8).

– Several types of drifting buoys, including CARTHE drifters (Novelli et al., 2017) drogued around 40 cm, SVP drifters (Niiler and Paduan, 1995) drogued at 15 m and Spotter wave-measuring buoys (Raghukumar et al., 2019), were deployed in the measurement areas.

– The R/V *Thalia* worked in the offshore area, providing continuous underway measurements of meteorolog-

ical parameters using a Météo-France BATOS operational system comprising a Vaisala WXT series sonic anemometer located approximately 10 m above the sea surface. The ship also carried a SBE21 thermosalinograph.

In the summer, the so-called Ushant tidal front has a strong influence on the surface currents, as well as hydrographic (Le Boyer et al., 2009) and atmospheric (Redelsperger et al., 2019) conditions in the offshore area. This seasonal feature typically disappears in October, and conductivity–temperature–depth (CTD) CE6 casts were performed from R/V *Thalia* to confirm that it had indeed vanished when the campaign took place. The water column was found to be very well mixed, with surface-to-bottom potential density anomalies being smaller than $0.002\,\mathrm{kg\,m^{-3}}$. The spatial homogeneity was also checked using the ship thermosalinograph and an infrared camera mounted on a Piper PA-23 plane, which surveyed the offshore area in a "lawn-mowing" pattern, flying under the clouds at an altitude of 500 to 1000 m. While small-scale surface features were observed on calm days, it is clear that no density-associated mesoscale structures were present.

The airborne radar measurements geometry over the offshore area consisted of relatively long (12 km) and straight tracks with different aircraft headings, forming a star pattern, as for the 22 November 2018 flight shown in Fig. 6. Tracks were flown every 12, 22.5 or 45° in azimuth, depending on flight duration constraints. The KaRADOC antenna was fixed relative to the aircraft and looking to port, while the KuROS antenna could either be fixed in the up-track or port cross-track directions, or it could rotate in the clockwise sense relative to the flight track. The KuROS Doppler data presented in this paper were acquired in the port-looking configuration.

## 3.2 Geophysical conditions

A wide range of geophysical conditions were encountered during the 1-week-long campaign. Four flights were performed over the offshore area on 21 November from 13:50 to 15:50, on 22 November from 12:15 to 15:00, on 24 November from 11:20 to 13:20, and finally on 26 November from

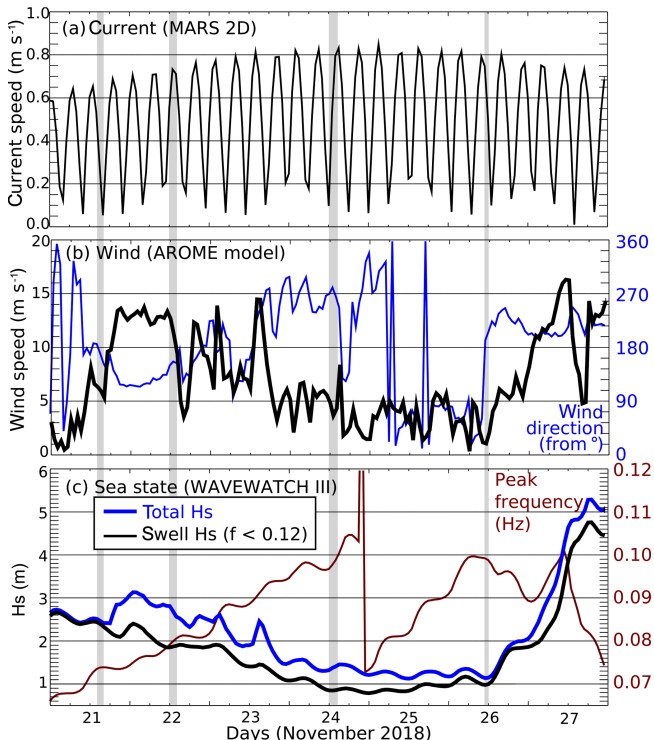

**Figure 7.** Time series at the location of the Trèfle buoy (5°15′ W, 48°15′ N) in the offshore zone of **(a)** ocean surface current speed from the MARS2D numerical model run at LOPS (Lazure and Dumas, 2008). **(b)** Wind speed (black) and direction (blue) from the AROME regional operational model run by Météo-France. **(c)** Total (blue) and swell (black) significant wave height and wave peak frequency (red) from the WAVEWATCH III numerical wave model run at LOPS (Roland and Ardhuin, 2014). The four time periods shaded in grey correspond to the times of fixed-antenna KuROS measurements. The corresponding observed environmental parameters are detailed in Table 3.

09:40 to 11:00. In this paper, we focus on data acquired on 22 and 24 November as the geophysical conditions were interesting and complementary (see below), and data were acquired with the largest azimuth diversity on these two days.

The 22 November flight took place at the end of a steady southeasterly wind episode ($13\,\mathrm{m\,s^{-1}}$ from 140°). The 24 November flight, in contrast, took place during a steady weak southwesterly wind period ($5\,\mathrm{m\,s^{-1}}$ from 225°) (Fig. 7b).

The wave height during the campaign was dominated by the presence of two swell systems from North Atlantic remote storms. The swell height decreased from 2.5 m on 21 November to 0.9 m on 24 November, with a peak frequency increasing from 0.07 to 0.1 Hz and a mean direction gradually veering from northwest to west. This swell has a small contribution to the Stokes drift of the order of 10 % of the wind-sea contribution on 22 November.

The main environmental conditions at the time of these star-pattern flights are summarized in Table 3.

### 3.3 KuROS instrument

KuROS is a Ku-Band (13.5 GHz) pulse-pair Doppler radar with a dual antennae system and azimuthal scanning possibility, which was developed in the framework of the CFOSAT prelaunch studies. Of the two antennas, the low-incidence (LI) antenna is nominally centered on a 14° incidence angle, while the medium-incidence antenna is nominally centered on a 40° incidence angle. Only the LI antenna, which was the more relevant for SKIM, was used during the campaign. This antenna uses an HH CE7 polarization. A comprehensive description of the system can be found in Caudal et al. (2014). A new antenna was used for the DRIFT4SKIM campaign, with characteristics given in Table 1.

The radar transmits a frequency-modulated pulse (chirp) with a 100 MHz bandwidth, achieving a 1.5 m range resolution and an effective ground-projected resolution of approximately 7 m (at 12°). The one-way 3 dB footprint in azimuth is 580 m wide at 12° and 3000 m of flight altitude. The pulse repetition frequency (PRF = 1 / PRI) depends on the altitude and is 23 kHz when the aircraft flies at 3000 m. The ambiguity of the Doppler velocity measurement (see Sect. A1.4 in the Appendix) is about $126\,\mathrm{m\,s^{-1}}$, which is much larger than expected from the measurements (below aircraft speed of $120\,\mathrm{m\,s^{-1}}$). In order to reduce the thermal noise contribution, the range-resolved pulse-pair signal is coherently averaged in the instrument over 1 ms, corresponding to 22 pulse pairs per instrument sample. For the purpose of this article, this was further coherently averaged per blocks of 15 samples.

As discussed in Appendix A, accuracy requirements for observation geometry are much less stringent for cross-track than for up-track and down-track Doppler velocity observations. The Doppler velocity data discussed in this article were all collected with the KuROS antenna in the port-looking orientation. This configuration also ensures an overlap with the KaRADOC footprint.

### 3.4 KaRADOC instrument

The Ka-band RADar for Ocean Current (KaRADOC) monitoring airborne radar sensor was developed for the DRIFT4SKIM campaign. KaRADOC is derived from the Still WAter Low Incidence Scattering (SWALIS) instrument, developed for the measurement of the NRCS of inland water surfaces in the Ka band. Further details on the system are given in Appendix C.

KaRADOC was mounted under the ATR-42 aircraft in a port-looking configuration. The two-way 3 dB footprint from 3000 m of altitude over a flat sea surface is an ellipse with diameters 45 and 60 m in the cross-track and along-track directions, respectively. The antenna is of the slotted-waveguide type and allows steering of the beam in elevation (incidence angle) by varying the working frequency. Data were acquired at different incidence angles from 6 to 14°, corresponding

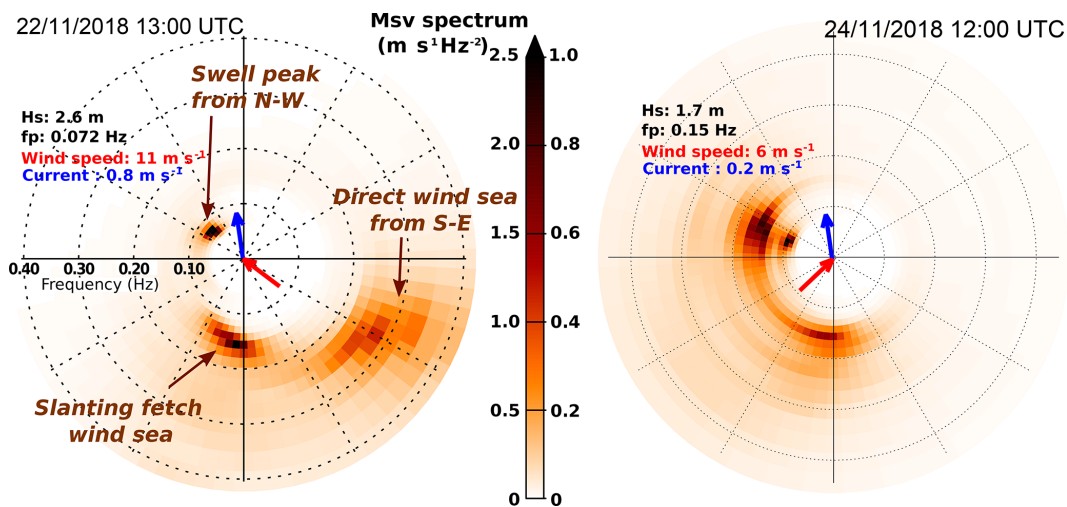

**Figure 8.** Directional wave spectra $E(f_r, \theta)$, as functions of the relative wave frequency $f_r$ and incoming wave azimuth $\theta$, estimated from the motions of the Trèfle buoy on 22 November at 13:00 UTC and Spotter buoy number 10 on 24 November at 12:00 UTC. The measured directional moments were transformed with the maximum entropy method (Lygre and Krogstad, 1986) and Doppler-shifted with $f_r = f - \mathbf{k} \cdot \mathbf{U}/(2\pi)$ for the moored Trèfle buoy. The red and blue arrows represent the AROME wind and MARS2D surface current vectors directions, respectively.

**Table 3.** Surface current velocity, Stokes drift and wind speed measured or estimated near position 48°15′ N, 5°15′ W. For each table entry, the parenthesized pair contains the (eastward, northward) components of the vector (cm s$^{-1}$) for current or Stokes drift estimates and (m s$^{-1}$) for wind speed estimates. Please note that the Stokes drift is only integrated up to 0.5 Hz. Stokes drift buoy data correspond to the Trèfle buoy for 22 November and Spotter buoy number 10 for 24 November.

| Time (mm/dd hh:mm) | CARTHE | SVP | HF radar | Buoy ($U_s$, $V_s$) | WW3 ($U_s$, $V_s$) | Wind (ship) | Wind (AROME) |
|---|---|---|---|---|---|---|---|
| 11/21 14:00 | (18, 72) | (21, 72) | (26, 69) | (0.69, 2.23) | (0.44, 2.06) | (−0.0, 7.3) | (0.5, 6.3) |
| 11/21 14:30 | (17, 58) | (19, 58) | (25, 58) | (0.88, 2.02) | | (−4.3, 6.9) | – |
| 11/21 15:00 | (15, 45) | (16, 49) | (17, 41) | (0.21, 2.54) | (0.41, 2.12) | (−4.5, 5.0) | (−1.1, 5.8) |
| 11/21 15:30 | (15, 22) | (15, 21) | (16, 26) | (0.23, 1.97) | | (−4.7, 7.8) | – |
| 11/22 12:00 | (−2, 73) | (−3, 81) | (−5, 58) | (−5.47, 8.86) | (−7.38, 11.55) | (−9.1, 7.1) | (−6.8, 10.7) |
| 11/22 12:30 | (−3, 97) | (4, 84) | (2, 71) | (−5.44, 9.19) | (−7.42, 11.37) | (−9.4, 7.2) | – |
| 11/22 13:00 | (6, 102) | (4, 94) | (7, 84) | (−4.72, 8.37) | (−7.07, 11.39) | – | (−5.2, 10.0) |
| 11/22 13:30 | (10, 85) | (12, 89) | (14, 88) | (−4.75, 8.02) | (−6.68, 11.50) | (−4.5, 9.1) | – |
| 11/22 14:00 | (9, 82) | (12, 87) | (23, 81) | (−3.28, 7.19) | (−6.35, 11.66) | (−3.9, 11.1) | (−4.4, 8.3) |
| 11/22 14:30 | (10, 78) | (11, 78) | (25, 72) | (−3.35, 6.93) | (−5.82, 11.76) | (−7.4, 7.1) | – |
| 11/24 11:30 | (−10, v2) | (−11, −6) | – | (2.47, 1.81) | – | (3.8, 2.9) | – |
| 11/24 12:00 | (−6, 19) | (−7, 16) | – | (2.49, 1.20) | (0.75, 2.92) | (4.0, 3.8) | (4.9, 0.1) |
| 11/24 12:30 | (−2, 40) | (−1, 40) | – | (2.92, 1.66) | (0.68, 2.71) | (4.8, 2.9) | – |
| 11/24 13:00 | (−1, 60) | (1, 59) | – | (3.20, 1.35) | (0.68, 2.71) | (4.5, 2.0) | (3.5, −0.7) |
| 11/24 13:30 | (−1, 77) | (2, 78) | – | (2.73, 1.29) | (0.70, 2.60) | (3.4, 2.8) | – |
| 11/26 10:00 | (−19, −83) | (−20, −87) | (−25, −62) | (0.46, −0.19) | (0.59, −0.64) | (−2.0, 0.5) | (−1.0, −0.6) |
| 11/26 10:30 | (−22, −80) | (−24, −84) | (−28, −63) | (0.32, −0.23) | (0.59, −0.64) | (−1.0, 1.4) | – |
| 11/26 11:00 | (−20, −74) | (−27, −74) | (−33, −66) | (0.30, −0.20) | (0.59, −0.64) | (0.6, 1.4) | (0.2, 1.1) |

to a range of frequencies from 32.5 to 38.2 GHz. This article focuses on the observations collected at $\theta = 12°$ and at 33.7 GHz.

KaRADOC does not implement a range-resolution scheme: the transmitted pulses last several microsec-

onds, and the whole FOV is illuminated simultaneously. The demodulated return signal is sampled at 15 MHz and archived. It is essentially constant while the electromagnetic wave is actually interacting with the sea surface. The useful signal segment is selected and its average is computed in

order to reduce the thermal noise contribution, yielding one complex amplitude for each pulse. Several hundred pulses are sent at 4 kHz PRF for each burst of measurements, with a burst repetition frequency of the order of 5 to 10 Hz, depending on the number of incidence angles in the scanning sequence. These parameters were varied during the acquisitions. Though they have a strong impact on NRCS and DFS estimate quality, we have found the low-pass-filtered DFS signal to be robust.

The pulse-pair complex signal is averaged for each burst in order to reduce the effect of coherent speckle. One complex pulse-pair sample is thus obtained per burst. Even at the lowest burst repeat frequency of 5 Hz, the plane moves by less than a third of the FOV along-track extension between bursts.

The impact of the acquisition parameters on the KaRADOC measurement normalization is not yet fully understood, and the NRCS measurements could not be exploited in the scope of this study. The noise-filtered DFS measurements are, however, not affected by these normalization changes and are valid.

## 4 Measurements

### 4.1 KuROS NRCS–DFS imagery

The KuROS NRCS–DFS imagery reveals a host of interesting features, modulations and dependencies. An in-depth analysis of all these processes is clearly out of the scope of this paper and will be the subject of forthcoming contributions from the DRIFT4SKIM team. This section thus only provides a cursory description of a few segments of $\sigma^0$ and DV data collected on 22 November 2018 when the wind speed was approximately 11 m s$^{-1}$, which are displayed in Fig. 9.

A first remark is that the NRCS is smooth, with a typical modulation depth of 1 dB after removing its mean trend as a function of incidence angle (Fig. 9a). This smoothness is in part due to the large footprint, but it also shows that the radiometric quality of the data and the coherent averaging performed are sufficient to control the thermal noise. Speckle noise is, however, still present, with different statistics depending on the radar look direction and the variable considered (not shown). The cross-track observation geometry leads to the best speckle noise reduction for the NRCS but to the worst-case speckle noise statistics for the DV.

The KuROS data clearly show a modulation in both NRCS and $U_{GD}$ associated with the northwesterly swell observed by the Trèfle buoy with a peak frequency of 0.07 Hz, corresponding to a wavelength $L = 320$ m (Fig. 8). This is particularly visible on the north–south-oriented flight tracks numbered 5 and 6 in Fig. 9b (see also Fig. 9f–g for a zoom-in on track 6). The apparent swell crest direction (dashed lines in Fig. 9b) differs from the true direction due to the scanning

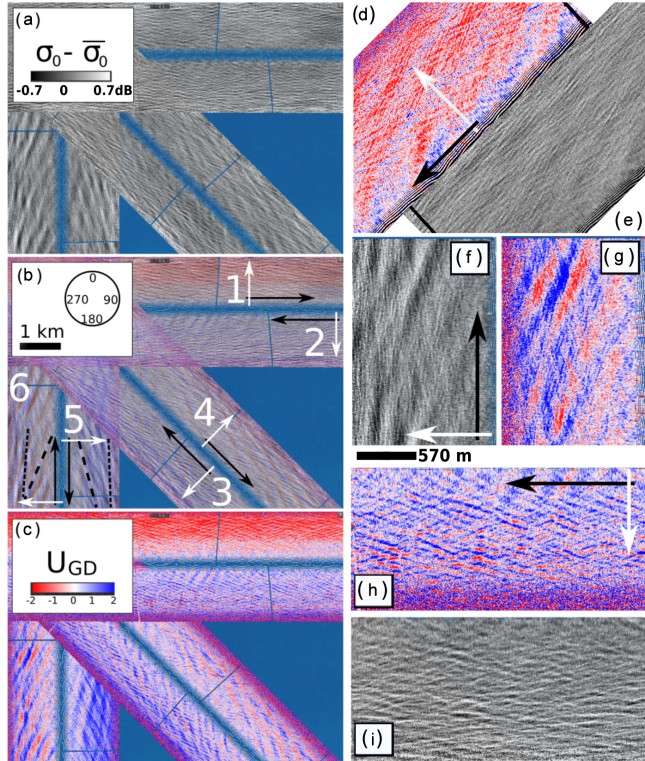

**Figure 9. (a, c)** Mosaics of KuROS backscattering intensity and Doppler velocity data acquired on 22 November with fixed port-looking antenna. **(b)** Overlay of the Doppler velocity and backscattering intensity. Flight tracks are numbered 1 to 6; black arrows indicate the flight direction, and white arrows point in the radar look direction. The long dashed lines represent the apparent direction of swell crests. Panels **(d)**–**(i)** show close-up views of selected tracks from **(a)** and **(c)**. The tracks shown in **(d)** and **(e)** are out of the frame of **(a)** and **(c)**. Panels **(f)**–**(i)** show close-ups of flight tracks 6 and 2. The 570 m scale bar applies to **(h)**–**(h)** TS14 and corresponds to the along-track 3 dB width of the radar beam at a 12° incidence angle, i.e., near the middle of the swath. The mean trend of $\sigma^0$ as a function of $\theta$ has been removed from the $\sigma^0$ data.

distortion effect (Walsh et al., 1989; Sutherland et al., 2018), as the swell propagates during the measurements at a phase speed of 22 m s$^{-1}$, while the aircraft moves at 120 m s$^{-1}$.

The shorter waves measured by the Trèfle buoy (Fig. 8) occupy a wide range of directions from a narrow wind-sea peak from the south at 0.16 Hz ($L = 60$ m) to a broad directional distribution at 0.22 Hz ($L = 30$ m), with a mean direction of 130° and a half-width (spread) of 45°, hence covering directions from 85 to 175°. These shorter components are present in the data from flight tracks 5 and 6 in the form of very narrow stripes with orientations shown by short dashed lines in Fig. 9b (see also Fig. 9f–g for a zoom-in on track 6). The "long-crested" appearance of the short waves in (d) and (e) is an artifact due to the wavefront-matching observation geometry (Jackson et al., 1985), with all other directions averaged out by the large azimuth width of the radar beam. If

purely geophysical, the phase relationship between the DV and NRCS modulations is expected to give the wave propagation direction. For flight track 6 in (f) and (g), the long swell propagates towards the radar, and the brighter slopes
(white) correspond to eastward velocities toward the radar (blue). This will be discussed in further detail below. Finally, (h) and (i) exhibit chevron patterns with crests facing both northeast and northwest. Whereas the waves from the southwest are expected to be much longer than those from the
southeast, this is not apparent in the KuROS data.

## 4.2 Ku-band NRCS

In this section we discuss the dependence of the Ku-band NRCS as a function of azimuth and incidence angle for the 22 and 24 November cases. The fixed-antenna and rotating-
15 antenna data are presented. In order to reduce the dispersion introduced by the short-scale modulating processes discussed above, the data were averaged per $1°$ incidence angle and azimuth bins. As mentioned before, full tracks are straight and relatively long ($12\,\mathrm{km}$), and they view a mainly
homogeneous ocean region. For the fixed-antenna observations, azimuthal diversity is obtained by performing tracks in different flight directions, forming a "star" pattern.

The variations of the Ku-band $\sigma^0$ are shown for 22 November in Fig. 10. These measurements show the ex-
25 pected modulation of 0.8 to $0.9\,\mathrm{dB}$ with azimuth, with a downwind–crosswind contrast that increases with the incidence angle. This contrast is larger for the higher winds on 22 November. The upwind–downwind asymmetry is expected from the behavior of the surface slope probability
density function (Chapron et al., 2002; Walsh et al., 2008; Munk, 2008). The exception are the $\sigma^0$ values for the flight tracks with a fixed antenna around the azimuths 90 and 270 (Fig. 10a), which have anomalous normalized values between 1 and 1.3 instead of expected values much closer to 1.
We have no explanation for this anomaly, which is genuine. No such anomaly was found for the rotating-antenna data collected later on the same day (Fig. 10b).

Discarding these azimuth ranges (shaded in grey in Fig. 10c), the data could be well fitted with a functional form
$a_0 + a_1 \cos(\varphi - \varphi_{\sigma,1}) + a_2 \cos[2(\varphi - \varphi_{\sigma,2})]$. As explained in Sect. 2.1, measuring this azimuthal variation is critical for the interpretation of the mean Doppler velocity due to the spurious azimuth gradient contribution. As expected, the fitted directions $\varphi_{\sigma,1}$ and $\varphi_{\sigma,2}$ are very close to the wind direction,
except for the lowest incidence angles for which the contrast is less than $0.05\,\mathrm{dB}$.

On 24 November, the $\sigma^0$ azimuthal contrast was much weaker (Fig. 11) due to the much lower wind speed and was actually not aligned with the wind direction when the mea-
50 surements were performed.

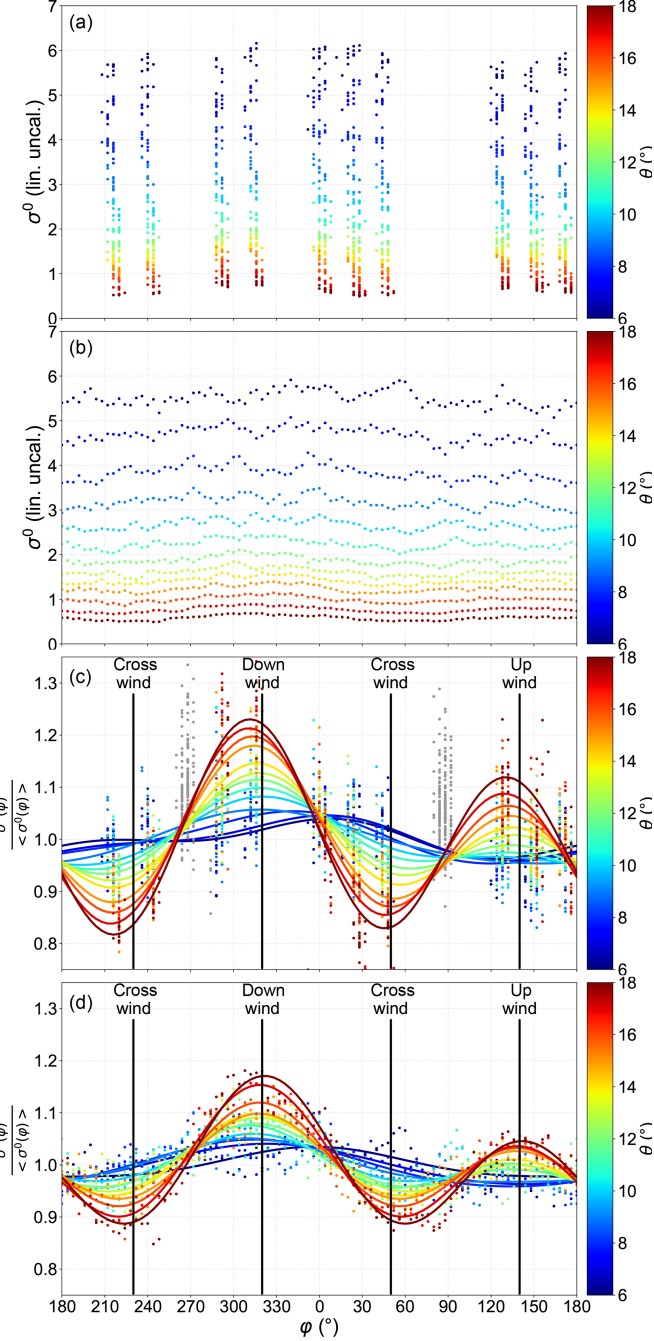

**Figure 10. (a, b)** Variations as a function of azimuth $\varphi$ of $\sigma^0$ for an incidence angle $\theta$ of 6 to $18°$ on 22 November for the port-looking (12:13–13:38 UTC TS15) and rotating-antenna (13:41–13:58 UTC TS16) flights, respectively. **(c, d)** Variations of $\sigma^0$ normalized by its azimuthal average for the fixed- and rotating-antenna data, respectively.

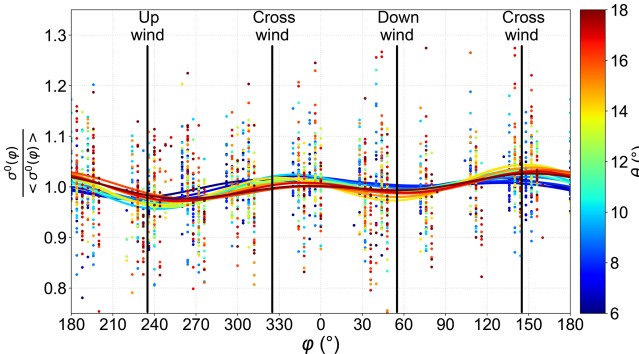

**Figure 11.** Same as Fig. 10c but using port-looking antenna data collected on 24 November at 11:22–13:03 TS17 (UTC).

## 4.3 Mean Doppler velocity from KaRADOC

We now quantitatively discuss the measured Doppler velocity signal in order to assess the agreement of our theory of the wave-induced contribution $U_{\text{WD}}$ with the measurements. This section is focused on the KaRADOC data, which are easier to interpret than the KuROS data due to the narrower radar beam of the instrument.

We present in Fig. 6 the low-pass-filtered $U_{\text{GD}}$ estimates retrieved from the 12° incidence angle KaRADOC data collected on 22 November between 12:13 CE9 and 12:59 (TU).

This representation is misleading, as much of the observed variability is in fact due to the effect of the flight track orientation. For instance, the largest contrast can be observed between the northeastward- and southwestward-directed flight tracks, even though to a first approximation a mere change in observation direction has occurred.

Another representation of the same data is proposed in Fig. 12. In this figure the $U_{\text{GD}}$ data are represented as blue lines shifted to the right of the plane ground track (in black) by an amount proportional to the instantaneous low-pass-filtered $U_{\text{GD}}$ value. This representation removes the trivial effect of observation direction changes and allows subtler effects to be better appreciated. For instance, noise-free observations of a constant vector $U_{\text{GD}}$ would appear as straight lines parallel to the flight tracks, all crossing at the tip of the vector. Deviations from this behavior, such as can be observed in Fig. 12, are indicative of measurement noise, geophysical variability or geophysical phenomena not accounted for by our theory.

For 22 November, 16 flight tracks are available collected from 12:13 to 12:59 (TU), and for 24 November 17 tracks were collected from 11:27 to 13:13 (TU).

Overall, the assumption of a constant vector is good to within $0.3\,\text{m}\,\text{s}^{-1}$. It is particularly striking that the three horizontal lines in Fig. 12a are almost perfectly aligned, corresponding to two flight tracks looking into azimuth 0° and one flight into azimuth 180°. On 22 November, the largest dispersion is for the 315 and 135° azimuths for which a total

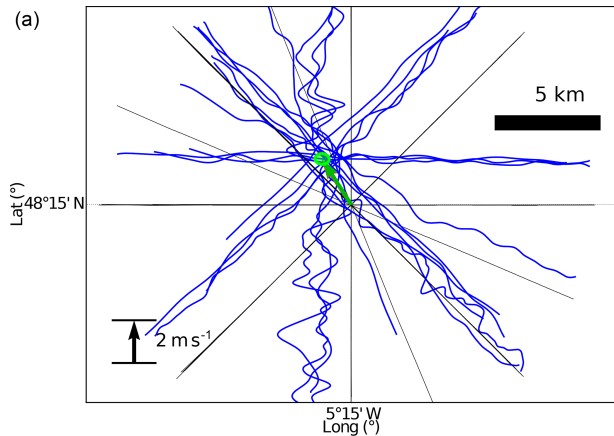

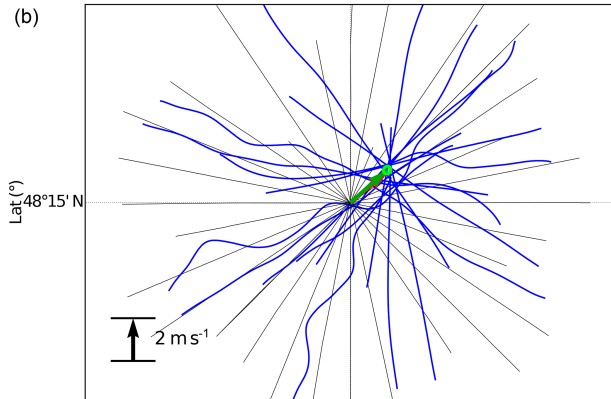

**Figure 12.** Plots of the Ka-band Doppler velocity signal on **(a)** 22 and **(b)** 24 November 2018. The flight tracks are marked as thin black lines. For each flight track, a thick blue line shifted to the right of the flight path by an amount proportional to the instantaneous low-pass-filtered Doppler velocity represents the projection of the $U_{\text{GD}}$ vector along the instrument line of sight. At the beginning of each track data were discarded until the plane stabilized. The green arrow represents the maximum likelihood estimate of the $U_{\text{GD}}$ vector using the whole data set. The (almost indistinguishable) red arrow shows the result of the least-squares sinusoidal fits shown in Fig. 13a and b. The 1 standard deviation error ellipse on the maximum likelihood estimate is represented in green.

of four tracks are available with very different values that are, however, consistent along each track.

Using the average values from the different tracks, we compare the measured Doppler velocity to the forward model given by Eq. (1), with $U_{\text{WD}}$ estimated from the in situ wave buoy data using the tools discussed in Sect. 2.2. The method combines the buoy spectrum up to 0.35 Hz and adds a high-frequency tail based on the Elfouhaily (1997) spectrum, then computes numerically the integrals of Eqs. (8) and (9) to obtain the DFS estimate. The TSCV contribution, $U_{\text{CD}}$, is taken to be the drift velocity of the nearest CARTHE drifter, which is uniform to within $3\,\text{cm}\,\text{s}^{-1}$ in the offshore area (interactive animations of all deployments and trajectories can be found at https://odl.bzh/eVRHv1TE, last access: TS18).

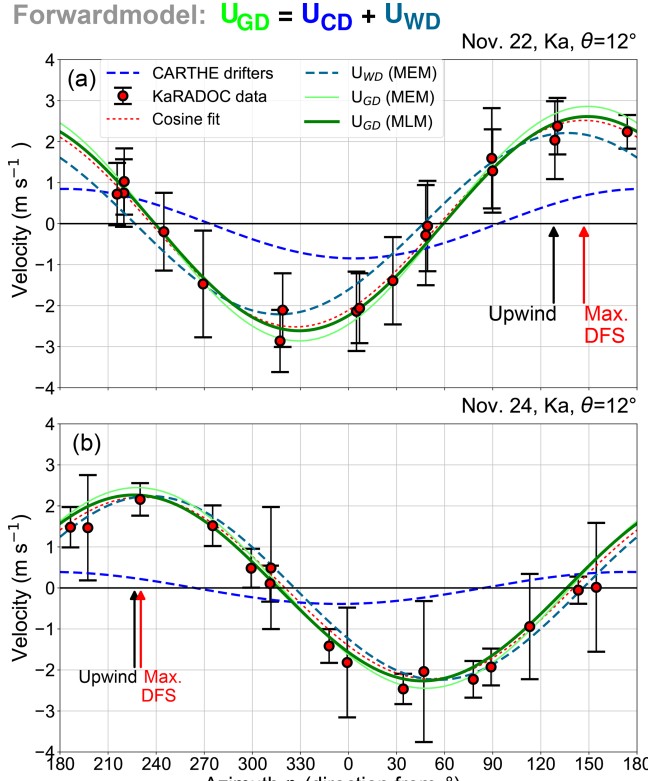

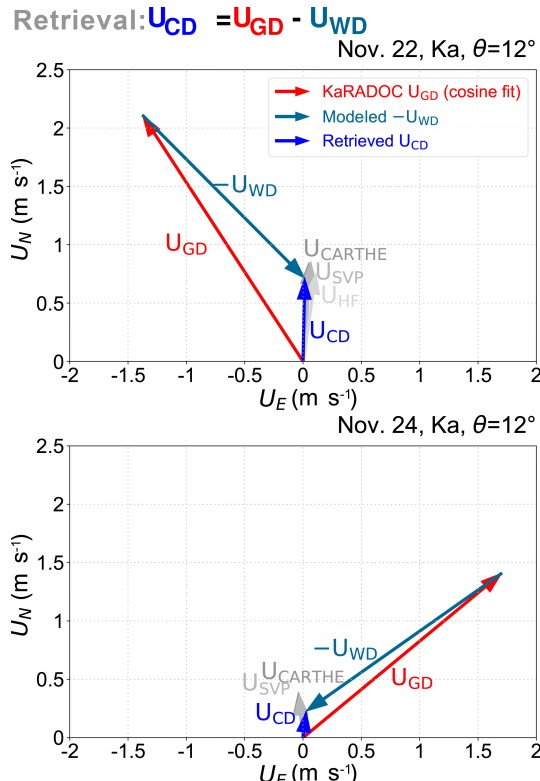

**Figure 13.** KaRADOC Doppler velocity (red circles) for the star-pattern flight on **(a)** 22 and **(b)** 24 November. Cosine function fits to the data (red lines). Modeled geophysical Doppler velocity $U_{GD}$ using the MEM (MLM) estimate of the directional wave spectrum (green and darker green). The modeled $U_{GD}$ is the sum of the CARTHE drifter velocity $U_{CD}$ (blue) and the wave Doppler velocity estimated from the measured spectra, $U_{WD}$ (midnight blue dashes).

**Figure 14.** TSCV retrieval $U_{CD}$ (in blue) obtained by subtracting the $U_{WD}$ from the MLM-processed Trèfle buoy data (in midnight blue) from the $U_{GD}$ vector determined from the KaRADOC measurements (in red), compared to field measurements by HF radar, CARTHE and SVP drifters (shades of grey).

Figure 13 shows the measured mean Doppler velocity and standard deviation for each track (the standard deviation is representative of the order of magnitude of the short-scale modulations due to waves, not of the error bar for the mean ⁵ DV). On 22 November (Fig. 13a), the current vector accounts for less than half of the observed magnitude of $U_{GD}$, and it is interesting that the maximum Doppler velocity is from azimuth 147°, between the wind direction (128°) and the up-current direction (183°). The directions of the modeled and ¹⁰ measured $U_{GD}$ are within 5° of each other.

Compared to the relatively high wind condition on 22 November, it is interesting to discuss the results for 24 November (Fig. 13b), with a wind speed of $5.5\,\mathrm{m\,s^{-1}}$ instead of $11\,\mathrm{m\,s^{-1}}$. The amplitude of the Doppler velocity ¹⁵ is not much reduced, in spite of more than halved current and Stokes drift. This is consistent with the expected near-constant value $C_0$ of the wave Doppler velocity magnitude, and this is the main result of the present paper.

The process leading to the estimation of the constant $C_0$ is, however, dependent on a number of assumptions: the di-²⁰ rectional wave spectrum must be evaluated from the buoy data, then matched to a parametric spectral shape before the necessary numerical integrations can be performed. The (Elfouhaily et al., 1997) spectral shape we have used depends on the wind speed and direction, but also on a wave age pa-²⁵ rameter, $\Omega$, equal to 0.84 for equilibrium seas.

Table 4 summarizes a subset of the extensive tests we have performed to check the sensitivities of this process. It is clear from this table that drastically changing the wind speed, as occurred between the two days, affects the magnitude of the ³⁰ computed $U_{WD}$ more at the Ku band than the Ka band, but not in a catastrophic way, and that the wave age parameter $\Omega$ can also be varied over its meaningful range quite freely. We have also checked that the transition frequency at which the spectral tail is matched to the observational data is not a very ³⁵ sensitive parameter, provided it is taken low enough for the buoy data to be of good quality where they are kept.

Extracting directional wave spectra from buoy data, however, is a quite an intricate and subjective step. Several methods have been developed over the years to this end, each ⁴⁰ with pros and cons (see Benoit et al., 1997, for a review).

**Table 4.** Modeled wave Doppler velocity amplitude $M_{WD}$ and direction $\varphi_{WD}$ at the Ku and Ka band with directional wave spectra produced using the maximum entropy or maximum likelihood methods and varying the wave age parameter $\Omega$ of the Elfouhaily (1997) high-frequency spectral tail. In all cases the transition frequency between the wave data and the high-frequency spectrum is $f_t = 0.35$ Hz. All values are estimated for $\theta = 12°$.

|  | Ka band | Ku band |
| --- | --- | --- |
| 11/22, 12:00 UTC | $M_{WD}$ (m s$^{-1}$)/$\varphi_{WD}$ (°) | |
| MEM, $\Omega = 0.84$ | 2.21/136.7° | 2.83/136.9° |
| MEM, $\Omega = 1.3$ | 2.21/136.8° | 2.83/137.0° |
| MEM, $\Omega = 2.5$ | 2.20/136.8° | 2.79/137.1° |
| MLM, $\Omega = 0.84$ | 1.97/135.1° | 2.54/135.3° |
| MLM, $\Omega = 1.3$ | 1.97/135.2° | 2.53/135.4° |
| MLM, $\Omega = 2.5$ | 1.96/135.2° | 2.51/135.6° |
| 11/24, 12:00 UTC | $M_{WD}$ (m s$^{-1}$)/$\varphi_{WD}$ (°) | |
| MEM, $\Omega = 0.84$ | 2.25/235.7° | 2.50/235.7° |
| MEM, $\Omega = 1.3$ | 2.24/236.3° | 2.49/236.4° |
| MEM, $\Omega = 2.5$ | 2.17/237.7° | 2.40/237.9° |
| MLM, $\Omega = 0.84$ | 2.07/234.3° | 2.29/234.3° |
| MLM, $\Omega = 1.3$ | 2.05/234.8° | 2.28/234.9° |
| MLM, $\Omega = 2.5$ | 1.99/236.2° | 2.19/236.2° |

Two of the best-established methods are the maximum entropy method (MEM) and the maximum likelihood method (MLM). The MEM is a parametric method which assumes a specific form of the directional spreading function. In each frequency band, the parameters of the spreading function are chosen such that the first moments of the azimuthal Fourier spectrum match the buoy-derived ones. The MLM is a nonparametric method akin to the Capon beamformer. In terms of directional moments measured by buoys, the MEM estimates provide spectra that exactly fit the measured moments, while the MLM produces spectra that have directional spreads larger than those obtained directly from the measured moments. However, it is not clear how they compare on other properties of the spectrum that may be relevant to the mean slope velocity. Comparing results obtained with these two methods was thus a convenient way to test the sensitivity of $U_{WD}$ to the sea state directional spread.

As Table 4 shows, using one technique or the other to estimate the resolved part of the wave directional spectrum does induce significant differences in the simulated $U_{WD}$ values, showing that the azimuthal width of the spectrum, which is currently not very well constrained observationally, is a sensitive factor. The values obtained using the broader MLM spectrum are consistently smaller than those obtained using the MEM spectrum. A broader azimuthal distribution only redistributes the weight between the different **Mss** components but reduces the contributions composing the msv$_0$ vector.

The results obtained using both methods are shown in Fig. 13 as light green and dark green lines. It appears that the MLM processing of the buoy data gives the best fit to the radar $U_{GD}$, showing that the directional spread of the sea state should not be taken too low. The possibility that the directional distribution of the Elfouhaily (1997) spectrum could be slightly too narrow for intermediate wavelengths of 2–10 m was, for instance, discussed in specific cases by Peureux et al. (2018). It is, however, not yet clear if it is specific to the very young wind seas they observed, although it could also explain some properties of L-band backscatter (Yueh et al., 2013). The MLM was used to process the Trèfle data in the rest of this study.

Conversely, Fig. 14 illustrates the use of the DV data for the retrieval of the surface current vector by subtraction of $U_{WD}$ from the fitted $U_{GD}$. The norm of the difference between the in situ measured and remotely sensed $U_{CD}$ vectors is less than 20 cm s$^{-1}$ on both days, which is significant but quite satisfying at such an early stage of the technique, especially taking into account the fact that geophysical variability due to time variations of wind and tidal current occurred over the several hours of the flight.

## 4.4 Mean Doppler velocity from KuROS

Due to the much broader radiation diagram of the KuROS antenna, analyzing the Ku-band data requires significantly more effort, as the $U_{AGD}$ spurious velocity contribution due to the azimuthal variation of $\sigma^0$ across the FOV discussed in Sect. 2.1 must be compensated for. The DV measurements, corrected for the $U_{NG}$ platform motion contribution but not for the $U_{AGD}$ contribution, are represented in Fig. 15 as red dots, while the green line represents the projection along the line-of-sight azimuth of the sum of the TSCV and the MLM-derived $U_{WD}$ vectors. The difference is clearly very large, reaching 2 m s$^{-1}$ in places on 22 November, and smaller on 24 November as the azimuthal modulation of the radar NRCS is much weaker.

Introducing the fits to the Ku-band NRCS data discussed in Sect. 4.2 in Eq. (A24) allows one to produce the corrected data represented by the magenta dots, which are in much better agreement with the green line (though a constant offset is apparent in the 24 November data, which is rejected by the cosine-fit procedure). Figure 16 summarizes the $U_{CD}$ retrieval operation in vector form (the magenta arrow represents only the first azimuthal harmonic component of the $U_{AGD}$ correction). The norm of the difference between the in situ (in grey) and remotely sensed (in blue) estimates of the TSCV is of the order of 0.5 m s$^{-1}$ on 22 November and of the order of 0.2 m s$^{-1}$ on 24 November. Again, these numbers, though admittedly not small, can be considered encouraging given the number of very large corrections applied to the data and the fact that the instrument had definitely not been designed for this purpose.

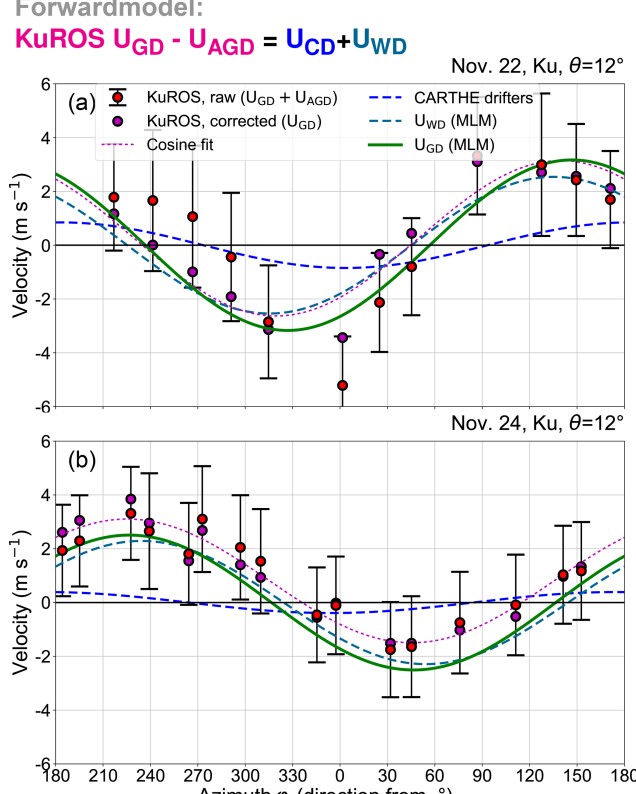

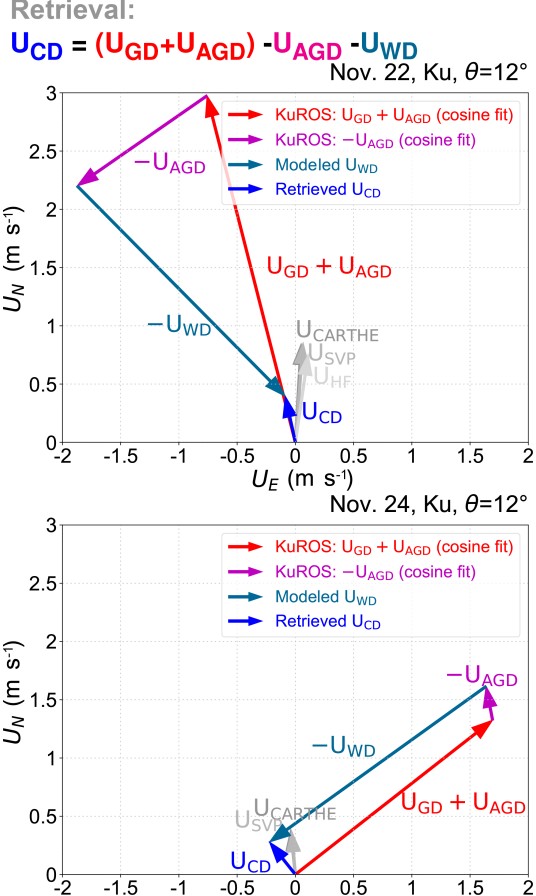

**Figure 15.** Ku-band Doppler measurements performed on **(a)** 22 November and **(b)** 24 November with the KuROS radar in the port-looking antenna configuration at a $\theta = 12°$ incidence angle. The graphical conventions are identical for the two plots. The red dots and error bars represent the average and $\pm 1$ standard deviation interval of the platform-motion-corrected DV measurements along the different tracks. The magenta dots mark the mean values after correction of the $U_{AGD}$ contribution. The magenta dotted line is the cosine fit to the corrected data. The blue and midnight blue lines respectively represent the projection along the line-of-sight azimuth of the CARTHE current measurements and the $U_{WD}$ vector computed from the MLM-processed Trèfle buoy data. The green line represents the sum of these two contributions and should agree with the magenta dotted line.

## 4.5 Observed Doppler velocity modulations

The range-resolution scheme implemented in KuROS makes it a very interesting instrument for the analysis of DFS and NRCS modulations. In particular, Caudal et al. (2014), with a different antenna (slightly narrower beam), have attempted to use the cross-spectrum of the DFS and NRCS to resolve the 180° ambiguity in the wave propagation direction. In the SKIM context, analyzing the contribution of the resolved scales to the correlation between $\sigma^0$ and the DFS could permit the development of empirical methods to estimate the unresolved part and provide estimates of $U_{WD}$.

In practice, with the antenna used for the DRIFT4SKIM flights, another contribution to the DFS modulations is also

**Figure 16.** Comparison of KuROS-derived Doppler velocity, corrected for the $U_{AGD}$ and $U_{WD}$ wave contributions, with in situ (CARTHE, SVP drifters and HF radar) current measurements.

caused by the gradients of $\sigma^0$ and the speed of the aircraft, just like the mean spurious $U_{AGD}$ velocity. Brighter areas in the field of view tend to strongly influence the DFS signal towards positive values if they are located to the front of the aircraft and negative values if they are located aft of the aircraft.

As a test of this, simulations were performed with the Radar Sensing Satellite Simulator (Nouguier, 2019), which are illustrated in Fig. 17. The amplitude of the spurious modulations is enhanced by 70 % when the antenna diagram is made 50 % wider in azimuth. With typical variations of $\sigma^0$ up to 1 dB over scales of the order of 1 km (e.g., Fig. 9), the variation of $\sigma^0$ with azimuth $\varphi$ is roughly proportional to $1/\sin\theta$, giving a $U_{AGD}$ that does not vary much with $\theta$, of the order of $1.5 \, \mathrm{m \, s^{-1}}$. This spurious velocity is larger than the $0.5 \, \mathrm{m \, s^{-1}}$ significant orbital velocity of the swell. As a result the phase relation between DFS and $\sigma^0$ can change sign as a function of azimuth due to the combination of two imaging mechanisms with comparable magnitudes and possibly opposite signs.

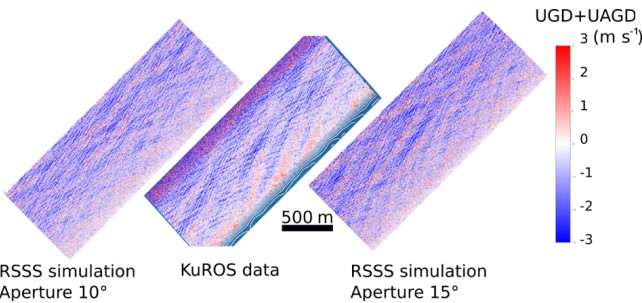

**Figure 17.** Qualitative validation of the R3S simulations of the radar imaging mechanism (Nouguier, 2019). Both the real data and simulation contain the geophysical modulation of velocities associated with surface velocities and slopes in the look direction (part of $U_{GD}$) as well as aircraft velocities and slopes in the flight direction (part of $U_{AGD}$). Note that the wave phases in the R3S simulation are random and cannot be expected to match those in the data or between the two simulations.

This effect will be weaker for shorter (wind-sea) components as soon as the wavelength and crest length become much shorter than the KuROS footprint $L_y$, as given by Eq. (A36): for a given $\sigma^0$ contrast, the gradient increases linearly as the scale $L$ is reduced, but the $U_{AGD}$ for a given gradient is reduced exponentially in $-L_y/L$.

## 5   Implications for SKIM

The use of two Doppler radars, in the Ka and Ku band with the same pulse-pair technique but antennas with very different radiation diagrams, has provided important insight for the preparation of the SKIM mission.

Regarding radar measurements, the DRIFT4SKIM campaign clearly demonstrated the feasibility of the TSCV retrieval approach proposed for SKIM (Ardhuin et al., 2018; ESA, 2019) based on the use of the SKIM wave spectrum measurements (here replaced by in situ buoy measurements) to estimate the wave Doppler velocity contribution $U_{WD}$ associated with the wave intrinsic phase speed. Measuring the first directional moments (on which the buoy estimates are based) is sufficient to estimate $U_{WD}$ and resolving wavelengths of 15 m (a frequency of 0.32 Hz) is sufficient to estimate the full spectral contribution, appending a parametric spectral shape for the unresolved shorter waves. In fact, it is most important to resolve the peak of the wind sea, and a resolved wavelength of 30 m is typically enough for wind speeds higher than $7 \, \mathrm{m \, s^{-1}}$. As this article has shown, however, the angular distribution of the directional spectrum is a sensitive element in both the resolved and parameterized wavelength ranges. Work is still needed to improve the spectral parameterization and to determine whether the accuracy of the sea state restitution algorithms intended for SKIM will be sufficient to solve this issue.

This experiment has also increased confidence in the use of forward models based on the Kirchhoff approximation, such as the R3S of Nouguier (2019), for the study of higher-order effects on the measured DFS. A subject of particular interest is, for instance, the effect of shear in the surface layer on the SKIM DFS, a key to the determination of the effective SKIM measurement depth.

The campaign also stressed the necessity of very good knowledge of the measurement geometry, including the antenna radiation diagram, and the spatial and azimuthal variation of the radar cross section. In this respect, the main characteristics of the instruments used for the present campaign and for the planned SKIM satellite mission are recalled in Table 5, together with the value of the prefactor of the $\sin(\varphi - \varphi_t) \partial_\varphi \log(\sigma^0)$ term in Eq. (6) of $U_{AGD}$ (as can be seen in Fig. 10, $\partial_\varphi \log(\sigma^0)$ is typically 0.1 rad$^{-1}$ at a 12° incidence angle). As the apparent mispointing due to $\sigma^0$ gradients in azimuth or space is proportional to the beamwidth squared, the non-geophysical velocities caused by this effect for SKIM, though non negligible, are actually much smaller than for KuROS, even at a 6° incidence angle.

As discussed in Sect. 2.3, due to the much reduced platform velocity, the pointing requirements for airborne systems are much easier to reach than for satellite systems for which a pointing accuracy of a few microradians cannot be achieved by attitude measurements alone (gyroscopes and star trackers) but must use a separation of the geophysical and non-geophysical patterns in the data (ESA, 2019). This data-driven approach is also used in airborne systems for correcting phase biases in the antenna diagram (Rodríguez et al., 2018).

Finally, as discussed in Sect. 2.3, we recall that the incidence angle is estimated from the range measurements in the cases of KuROS and SKIM and estimated directly from the platform attitude for the pencil-beam case of KaRADOC. In the spaceborne context, the local slope of the ocean has to be taken into account, as it can induce a mispointing of the nadir beam of up to 300 µrad (Sandwell and Smith, 2014) and induce a correction in the elevation angle at the observation point.

Other radar system constraints or optimizations for satellite systems are discussed by Rodriguez (2018) and the ESA (2019, chap. 5), with sampling issues further analyzed by Chelton et al. (2019).

## 6   Conclusions and perspectives

The DRIFT4SKIM campaign clearly demonstrated that surface geophysical velocities can be measured by microwave Doppler radars implementing the pulse-pair method at the Ka band at a 12° incidence angle. The Ku-band measurements, though less easy to interpret due to the large antenna beamwidth of the instrument, also supported this view. The campaign data are consistent with a geophysical model func-

**Table 5.** Main differences between the KaRADOC and KuROS airborne radars used in the present article and the SKIM system as presented by the ESA (2019). The factor $\sigma_\varphi^2 V_p/2$ is the prefactor of $\sin(\varphi - \varphi_t)\partial_\varphi \log(\sigma^0)$ in the expression of $U_{\mathrm{AGD}}$.

|  | KuROS 12° | KaRADOC 12° | SKIM 12° | SKIM 6° |
|---|---|---|---|---|
| Altitude (km) | 3 | 3 | 832 | 832 |
| Platform velocity $V_p$ (m s$^{-1}$) | 120 | 120 | 7000 | 7000 |
| Beamwidth ($\alpha_{-3\,\mathrm{dB}}$) (°) | 15.0 | 1.85 | 0.65 | 0.58 |
| Gaussian fit parameter $\sigma_\varphi$ (°) | 30.6 | 3.8 | 1.32 | 2.36 |
| $\sigma_\varphi^2 V_p/2$ (m s$^{-1}$ rad) | 17 | 0.26 | 1.9 | 5.9 |

tion (GMF) that expresses the geophysical DFS as the sum of the range component of the total surface current velocity and a wave DFS that is a weakly varying function of the sea state of the order of 2.0 m s$^{-1}$ at the Ka band and 2.4 m s$^{-1}$ at the Ku band. This wave DFS integrates contributions of all wavenumbers and directions, weighted by the surface slope spectrum. It can be well estimated from the sea surface elevation directional spectrum using the Kirchhoff approximation framework.

The campaign highlighted the importance of very good knowledge of the platform motion and orientation as well as the radar line-of-sight direction vector. The Ku-band NRCS–DFS imagery, though not very successful in that respect, observed a large number of interesting modulation phenomena, which will be analyzed in more detail in forthcoming contributions.

In general, the robustness of the theoretical GMF and its possible empirical adaptation will require the acquisition of more data in a wider range of wind and wave conditions. An in-depth investigation of the angular width of the sea state directional spectrum in the short gravity wave regimes seems of particularly high interest in this respect. Also, obtaining a description of the scale-resolved statistics of sea surface slope skewness would open the path to a Kirchhoff approximation study of the upwind–downwind asymmetry of the radar NRCS and DFS, which is currently lacking.

Finally, the test of near-nadir satellite measurements is limited by the very different viewing geometry due to the difference in altitude. Airborne measurement footprints are at most 500 m or so and thus cannot reproduce the averaging properties of the much wider footprint of a satellite instrument. Still, this medium-sized footprint is comparable to the unfocused SAR resolution that will be obtained with SKIM and provides some practical application with a similar azimuthal averaging that has a limited directional resolution for swell spectrum measurement.

Future airborne systems may ideally combine higher incidence angles, such as that used on DopplerScatt (Rodríguez et al., 2018), OSCAR and Wavemill (Martin et al., 2018), with near-nadir angles that allow for unambiguous wave measurements. In that case, the large azimuthal footprint of KuROS is probably not necessary, and a narrower beam like KaRADOC can be used, greatly simplifying the analysis.

## Appendix A: Doppler scatterometry theory

This Appendix proposes an extension of the theory of pencil-beam Doppler scatterometry exposed in Rodriguez (2018) and Rodríguez et al. (2018) to the case of near-nadir fan-beam instruments such as SKaR and KuROS. It compiles a number of processing steps or concepts that had to be developed for the analysis of the DRIFT4SKIM KuROS data. In each section the differences from and similarities to the spaceborne SKIM context are highlighted.

### A1 Pulse-pair theory

#### A1.1 Radar pulse-pair measurements

A radar instrument works by sending microwave pulses into the environment and recording the echo from its field of view. Usual scatterometers consider only the intensity of the return signal. Coherent instruments, such as SARs, measure both the amplitude of the return signal and its phase with respect to the transmitted carrier as a function of range. Over the ocean, the phase of the return signal for a single pulse is random and uniformly distributed over the unit circle. The radar returns of successive pulses transmitted at short intervals are, however, correlated, and the time history of the phase can be used to measure the relative motion of the radar and the scatterers. SARs make use of this property to refine the along-track resolution of backscattering cross-sectional measurements. SKIM and the other proposed Doppler missions aim to use it to obtain direct surface current measurements.

As explained by Rodriguez (2018, Appendix A), the complex amplitude of the return signal of a pulse transmitted at time $t_i$ can be expressed as

$$
E_i\left(t_i, r'\right) = n\left(t_i, r'\right) + \frac{A(r')}{r'^2}
$$
$$
\int G(t_i, \boldsymbol{x}) \chi\left(r' - r(t_i, \boldsymbol{x})\right)
$$
$$
\exp[-2ikr(t_i, \boldsymbol{x})] s(t_i, \boldsymbol{x}) \, \mathrm{d}S, \qquad (A1)
$$

where the integral is performed over the sea surface, $A(r')$ is a time-independent weakly dependent function of range, unimportant for our purposes here (corresponding in particular to the effects of transmitted signal amplitude, receiver and processing gain, and attenuation losses), $G(\boldsymbol{x})$ is the one-way antenna diagram, $\chi(r)$ is the range-point-target response of the instrument, $r'$ is the nominal pixel range in the time sampled signal, $k = 2\pi/\lambda$ is the radar wavenumber, $r(t_i, \boldsymbol{x})$ is the range from the radar to the observation point $\boldsymbol{x}$ at time $t_i$, $n(t_i, r')$ is the thermal noise contribution, and $s(t_i, \boldsymbol{x})$ is the complex reflection coefficient of the sea surface at instant $t_i$ and location $\boldsymbol{x}$.

As mentioned by Rodríguez et al. (2018), the thermal noise contribution, though it plays a major role in the quality of the measurements, is conceptually simple and can be safely considered $\delta$-correlated in time and characterized by a single quantity, its average power $N$. The reflection coefficient $s(t_i, \boldsymbol{x})$, on the other hand, emerges from the interaction of the electromagnetic waves with the ocean surface and has much richer physics. It is affected by electromagnetic phenomena as well as the geometry and kinematics of the sea surface itself, and its statistics are further complicated by the so-called "speckle" phenomenon. As stated by Rodríguez et al. (2018), the correlation function of this coefficient as a function of time and space separation, averaged over speckle realizations, can be modeled as

$$
\left\langle s(t, \boldsymbol{x}) s^*\left(t', \boldsymbol{x}'\right) \right\rangle_S = \delta\left(\boldsymbol{x} - \boldsymbol{x}'\right) \sigma^0(t, \boldsymbol{x}) \gamma_{\mathrm{TS}}\left(|t - t'|\right), \quad (A2)
$$

with $\sigma^0(t, \boldsymbol{x})$ the normalized radar backscattering cross section (NRCS) in the appropriate polarization and $\gamma_{\mathrm{TS}}(|\tau|)$ a function describing its time decorrelation at a fixed location due to the life history of individual scattering patches.

The so-called pulse-pair technique of Zrnic (1977) relies on the properties of the product of the return signals from consecutive radar pulses. Combining Eqs. (A1) and (A2) to compute the speckle-averaged product of the return signals for two radar pulses sent at $t_1$ and $t_2 = t_1 + \Delta t$, with $\Delta t$ the pulse repetition interval (PRI), one obtains

$$
\mathrm{PP}_{\Delta t}\left(t_1, r'\right) = \left\langle E_2\left(t_2 = t_1 + \Delta t, r'\right) E_1\left(t_1, r'\right)^* \right\rangle_S \qquad (A3)
$$

as

$$
\mathrm{PP}_{\Delta t}\left(t_1, r'\right) = \frac{A^2(r')}{r'^4} \gamma_{\mathrm{TS}}(|\Delta t|) \int \chi^2\left(r' - r(t_1, \boldsymbol{x})\right)
$$
$$
G^2\left(t_1, \boldsymbol{x}\right) \sigma^0\left(t_1, \boldsymbol{x}\right)
$$
$$
\exp[-2ik\left[r\left(t_1 + \Delta t, \boldsymbol{x}\right)\right.
$$
$$
\left. - r\left(t_1, \boldsymbol{x}\right)\right]] \, \mathrm{d}S. \qquad (A4)
$$

As can be seen in this equation, the phase of the pulse-pair signal contains a weighted average of the time rate of change of the distance separating the radar from the scattering elements in its instantaneous footprint. This rate of change can be interpreted as a velocity.

#### A1.2 Measurement geometry

Figure 1a and b summarize the acquisition geometry in the airborne and spaceborne settings. The antenna radiation diagram $G^2(t_1, \boldsymbol{x})$ is represented as grey shading of the sea surface, while the range-point response function $\chi^2(t_1, r' - r(t_1, \boldsymbol{x}))$ is represented as white grating. In Eq. (A4), we have made the assumptions that $G(t_1, \boldsymbol{x}) = G(t_2, \boldsymbol{x})$ and $\chi(r' - r(t_1, \boldsymbol{x})) = \chi(r' - r(t_2, \boldsymbol{x}))$, neglecting the effect of the spatial translation of the beam illumination pattern and range-resolution weighting distribution on the sea surface.

This is a very good approximation for airborne pulse-pair radar observations and a quite good one for spaceborne observations. For airborne instruments, the PRI is usually chosen such that the line-of-sight projection of the platform movement over a PRI is smaller than one-half the carrier

wavelength to avoid phase ambiguity. For spaceborne instruments, avoiding ambiguity is not practical due to the much larger platform velocity, but the PRI is constrained by other considerations, and the platform displacement over a PRI is much smaller than the characteristic scales of the antenna radiation diagram and of the range-point response.

### A1.3 Pulse-pair signal approximation

Returning to Eq. (A4), we see that over the time interval separating the two radar pulses, the radar has moved from its original position $x_R(t_1)$ to $x_R(t_1) + V_P \Delta t$, and the scatterers originally located at $x$ have moved to $x + v_s \Delta t$ (specifying the reference frame is not yet necessary since only relative separations are important at this stage). The radar-to-scatterer vector has thus changed by $[v_s(x) - V_P] \Delta t$. The distance change can be approximated by

$$r(t_1 + \Delta t, x) - r(t_1, x) = \Delta t \frac{x - x_R(t_1)}{||x - x_R(t_1)||} \cdot (v_s - V_R), \quad \text{(A5)}$$

where the neglected terms are of the order of $\Delta t^2 ||v_s - v_R||^2 / ||x - x_R(t_1)||^2$. Introducing

$$e(x) = \frac{x - x_R(t_1)}{|x - x_R(t_1)|}, \quad \text{(A6)}$$

the unit vector pointing from the radar location at $t_1$ to the observation point (choosing either time instant is equivalent, as the difference is of the same order of magnitude as the neglected terms), the pulse-pair signal can be expressed as

$$PP_{\Delta t}(t_1, r') = \frac{A^2(r')}{r'^4} \gamma_{TS}(\Delta t) \int G^2(t_1, x)$$
$$\chi^2(r' - r(t_1, x)) \sigma^0(t_1, x)$$
$$\exp[2ik \Delta t \, e(x) \cdot (V_R - v_s(x))] dS. \quad \text{(A7)}$$

This equation is not very practical, as the relative motion of the scatterers with respect to the radar enters as the argument of an exponential integrand. Obtaining an equivalent representation as the exponential of a sum of weighted integrals would be desirable. Introducing the effective illuminated surface,

$$S(t_1, r') = \int G^2(t_1, x) \, \chi^2(r' - r(t_1, x)) dS, \quad \text{(A8)}$$

the normalized weighting function,

$$W(t_1, r', x) = \frac{G^2(t_1, x) \, \chi^2(r' - r(t_1, x))}{S(t_1, r')}, \quad \text{(A9)}$$

the average and fluctuating parts of the NRCS

$$\overline{\sigma^0}(t_1, r') = \int W(t_1, r', x) \, \sigma^0(t_1, x) dS, \quad \text{(A10)}$$

$$\widetilde{\sigma^0}(t_1, r', x) = \frac{\sigma^0(t_1, x)}{\overline{\sigma^0}(t_1, r')}, \quad \text{(A11)}$$

and borrowing the algebraic technique of "cumulant expansion" from probability theory, it is possible to express $PP_{\Delta t}$ as

$$PP_{\Delta t}(t_1, r') = \frac{A^2(r')}{r'^4} \gamma_{TS}(\Delta t) \overline{\sigma^0}(t_1, r') \, S(t_1, r')$$
$$\exp\left[\sum_{n=1}^{\infty} \frac{(i2k \Delta t)^n}{n!} \kappa_n\right], \quad \text{(A12)}$$

with $\kappa_n$ the successive cumulants of $\underline{e}(x) \cdot (V_R - v_s(x))$ with respect to the density distribution $\sigma^0(t_1, r', x) \, W(t_1, r', x)$. As all the $\kappa_n$ are real, we see that odd-$n$ terms contribute to the argument of the pulse-pair signal, while even-$n$ terms contribute to its magnitude. Keeping only the first two terms in the sum, one obtains

$$PP_{\Delta t}(t_1, r') = \frac{A^2(r')}{r'^4} \gamma_{TS}(\Delta t) \overline{\sigma^0}(t_1, r') \, S(t_1, r')$$
$$\exp[i2k \Delta t \kappa_1] \exp\left[-2(k \Delta t)^2 \kappa_2\right]. \quad \text{(A13)}$$

As expected, the expression of $\kappa_1$,

$$\kappa_1(t_1, r') = \int W(t_1, r', x) \, \widetilde{\sigma^0}$$
$$(t_1, r', x) \, e(x) \cdot (V_R - v_s(x)) dS, \quad \text{(A14)}$$

shows that to first order the argument of the pulse-pair signal gives access to the integral over the footprint of the relative velocity of the scatterers with respect to the radar. The expression of $\kappa_2$,

$$\kappa_2(t_1, r') = \int W(t_1, r', x) \, \widetilde{\sigma^0}(t_1, r', x)$$
$$[e(x) \cdot (V_R - v_s(x)) - \kappa_1]^2 dS, \quad \text{(A15)}$$

is a description of the impact of the variability of $e(x)$, $\widetilde{\sigma^0}$ and $v_s$ inside the footprint on the pulse-pair signal magnitude.

### A1.4 Pulse-pair signal phase approximation

Working now in the Earth-fixed reference frame at the observation point, we define

$$V_{GD} = -\int W(t_1, r', x) \, \widetilde{\sigma^0}(t_1, r', x) \, e(x) \cdot v_s(x) dS, \quad \text{(A16)}$$

the (geophysically relevant) weighted projection of the scatterer velocity in that frame on the radar line of sight, and

$$V_{NG}(t_1, r') = V_R \cdot \int W(t_1, r', x) \, \widetilde{\sigma^0}(t_1, r', x) \, e(x) dS, \quad \text{(A17)}$$

the (non-geophysical) projection of the radar velocity (our conventions are such that $V_{GD}$ is positive when the scatterers move towards the radar and that $V_{NG}$ is positive when the radar moves towards the footprint, in keeping with everyday intuition).

With these conventions, one sees that

$$V_{GD}\left(t_1, r'\right) = \kappa_1\left(t_1, r'\right) - V_{NG}\left(t_1, r'\right). \tag{A18}$$

Using Eq. (A13), one can obtain $\kappa_1$ approximately as $1/(2k\,\Delta t)$ times the argument of the complex pulse-pair signal. At this stage, one must, however, consider a bit carefully the ambiguity that is inherent in phase measurements. As the phase of a complex number is only known up to a multiple of $2\pi$, $\kappa_1$ is only obtained up to a multiple of $\frac{\lambda}{2\Delta t}$. This effect can be neglected as long as both $V_{NG}$ and $\kappa_1$ remain within the unambiguous interval $\left[-\frac{\lambda}{4\Delta t}; \frac{\lambda}{4\Delta t}\right]$. For larger platform velocities, care must be taken to add the right multiple of $\frac{\lambda}{2\Delta t}$ to $\kappa_1$ before subtracting $V_{NG}$. For airborne instruments it is usually feasible to select a small enough PRI to avoid ambiguity altogether. For satellite instruments, one approach is to select a solid Earth-fixed reference frame, in which $\boldsymbol{v}_s$ is small, and to work on the phase-migrated pulse-pair signal

$$\widetilde{PP}_{\Delta t}\left(t_1, r'\right) = \exp\left[-i2k\,\Delta t\, V_{NG}\right]\, PP_{\Delta t}\left(t_1, r'\right). \tag{A19}$$

It is easy to see that $V_{GD}$ can be retrieved as

$$V_{GD}\left(t_1, r'\right) = \frac{1}{2k\,\Delta t}\arg\left(\widetilde{PP}_{\Delta t}\left(t_1, r'\right)\right). \tag{A20}$$

At this stage, even a coarse approximation of $V_{NG}$ can be used, as long as it is sufficient to resolve the phase ambiguity. This is important in particular for the onboard processors of satellite instruments, which have to rely on limited quality in terms of position, velocity and pointing information and typically cannot use the $\sigma^0$ distribution information that ground segment processors can retrieve from the signal. The correction applied by the onboard processor must, however, be accounted for in later processing stages.

## A2 Non-geophysical contribution $V_{NG}$

The non-geophysical contribution $V_{NG}$ must be estimated from the platform velocity and radar beam pointing. For pulse-limited instruments such as KuROS and SKaR, the incidence angle is determined in each range bin as a function of the altitude. The accuracies of the range-resolution and altitude determination processes are then critical. Last, asymmetric azimuthal variation of the sea surface NRCS within a given range bin tends to bias the effective observation azimuth towards the brighter part of the instrument FOV. This section discusses these different aspects.

### A2.1 Beam pointing accuracy

From now on, we work in the simplified setting of the flat-Earth approximation, in which the elevation and incidence angles $\gamma$ and $\theta$ are equal. We use a platform-fixed reference frame, the origin of which is located at the antenna phase center of the instrument, with the TS19 $\boldsymbol{x}$ vector pointing to the geometric front of the platform, the $\boldsymbol{y}$ vector pointing to starboard, the $\boldsymbol{z}$ vector pointing to the floor and a local geographic north–east–down reference frame, the origin of which is fixed to the solid Earth and located at a suitable point of the campaign area.

The orientation of the platform-fixed reference frame with respect to the local geographic frame is provided by the platform IMU as (roll, pitch, heading) Euler angles, from which one can construct the direction cosine matrix TS20

$$\mathbf{DCM} = \begin{bmatrix} c_p c_h & s_r s_p c_h - c_r s_h & c_r s_p c_h + s_r s_h \\ c_p s_h & s_r s_p s_h + c_r c_h & c_r s_p s_h - s_r c_h \\ -s_p & s_r c_p & c_r c_p \end{bmatrix}, \tag{A21}$$

allowing one to express the components of a vector in the $(N, E, D)$ frame from its $(x, y, z)$ components in the platform-fixed frame. The two reference frames are consistent in the sense that the frame vectors coincide when the platform is in constant-altitude flight towards the north. In the above expression we have used the transparent notation $c_p \rightarrow \cos(\text{pitch})$, $s_r \rightarrow \sin(\text{roll})$ and $c_h \rightarrow \cos(\text{heading})$. Other quantities worth introducing are the course $c$ and glide angle $g$ such that the plane velocity vector in the NED CE10 frame is

$$\boldsymbol{V_R} = V_R\left[\cos(g)\cos(c)\boldsymbol{N} + \cos(g)\sin(c)\boldsymbol{E} + \sin(g)\boldsymbol{D}\right]. \tag{A22}$$

In the NED frame, the pointing vector $\boldsymbol{e}$ can be expressed as

$$\boldsymbol{e} = \sin(\theta)\left[\cos(\varphi)\boldsymbol{N} + \sin(\varphi)\boldsymbol{E}\right] + \cos(\theta)\boldsymbol{D}. \tag{A23}$$

Its components in the platform-fixed frame can be determined using the fact that $\mathbf{DCM}^{-1} = \mathbf{DCM}^T$. The corresponding antenna azimuth and elevation angles $\varphi$ and $\gamma$, in terms of which the radiation diagram is specified, can then be expressed using the platform-fixed to antenna-fixed reference frame transformation matrix.

With these notations and using Eq. (A17), one can express $V_{NG}$ as

$$V_{NG}\left(t_1, r'\right) = V_R \int W\left(t_1, r', \boldsymbol{x}\right)\, \widetilde{\sigma^0}\left(t_1, r', \boldsymbol{x}\right)$$
$$\left[\cos(g)\sin(\theta)\cos(\varphi - c) + \sin(g)\cos(\theta)\right] dS. \tag{A24}$$

Constant-altitude flight corresponds to $g \simeq 0$. We thus concentrate on the impact of errors in the first term on the right-hand side of this equation. Quite clearly, the impact of errors in $\sin(\theta)$ is largest when the instrument views the area where $\cos(\varphi - c)$ is large, i.e., in the up-track and down-track directions, while the impact of errors in the azimuthal direction is largest when the instrument looks cross-track (i.e., where the derivative of $\cos(\varphi - c)$ is close to 1).

Leaving aside for the moment the effects of uncertainties on $W(t_1, r', \boldsymbol{x})$ and $\widetilde{\sigma^0}(t_1, r', \boldsymbol{x})$, one sees that at a 12° incidence angle and for a platform velocity of $7000\,\mathrm{m\,s^{-1}}$ (spaceborne instrument), the SKIM $40\,\mathrm{cm\,s^{-1}}$ error budget for horizontal velocity measurements translates to pointing accuracies of 4.5 and 21 µrad in incidence angle and azimuth, respectively (see the discussion in Sect. 2.3). In the airborne case at $120\,\mathrm{m\,s^{-1}}$ platform velocity and 3000 m of altitude, the corresponding numbers are 0.26 and 1.25 mrad for incidence angle and azimuth pointing accuracy for KuROS, respectively. In the cross-track viewing geometry of KaRADOC, only the comparatively mild (but still quite demanding) 1.25 mrad azimuth pointing accuracy requirement applies.

Figure A1a shows the measurement geometry seen from above. One can see that uncertainties on the viewing azimuth and incidence angle have different origins.

– The uncertainty in azimuth can be due to imperfect knowledge of the weighting corresponding to the $W(t_1, r', \boldsymbol{x})\,\sigma^0(t_1, r', \boldsymbol{x})$ term in Eq. (A24). This can of course come from imperfect platform attitude or antenna orientation information but also from an imperfect characterization of the antenna radiation diagram or the distribution of $\sigma^0$ on the sea surface.

– The uncertainty in incidence angle is due to imperfect knowledge of the radial position of the range-resolution bins (yellow striping of the footprint in Fig. A1a). This can be due to imperfect timing accuracy or to imperfect knowledge of the vertical separation between the instrument and sea surface.

## A2.2    Timing and altitude accuracy

For this brief discussion of the effects of timing and altitude accuracy on incidence angle estimation, we consider a single range bin whose "true" range from the radar is $r$, whose altitude with respect to the radar is $H$ and the incidence angle is $\theta$. In this case $\theta = \arccos(H/r)$. If the radar now suffers from a timing error $\delta r$, the instrument will detect a false altitude $H - \delta r$ but will ascribe to range bin $r - \delta r$ the signal coming from $r$. In the meantime, we consider the surface-tracking algorithm to suffer from an error $\delta h$ and detect the surface at range $H - \delta r - \delta h$. The data from this range bin will thus be processed using an angle of incidence

$$\theta + \delta\theta = \arccos\left(\frac{H - \delta r - \delta h}{r - \delta r}\right), \qquad (A25)$$

different from the correct value by

$$\delta\theta \simeq \frac{1}{H\tan(\theta)}\left[\delta h + \delta r\left[1 - \cos(\theta)\right]\right]. \qquad (A26)$$

Considering $\delta h$ and $\delta r$ to be independent, we see that at 12° the incidence angle knowledge requirements expressed

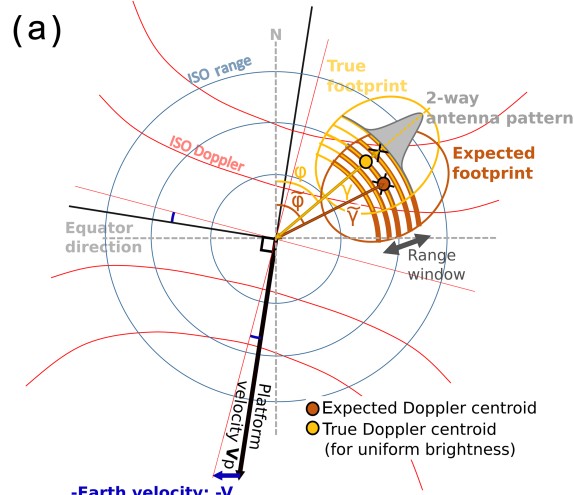

(a)

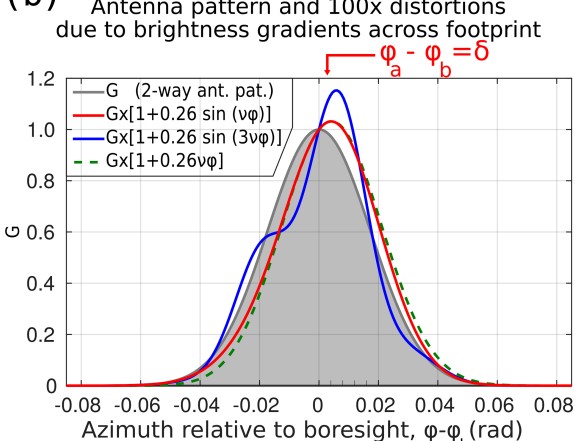

(b)

**Figure A1. (a)** True pointing. The attitude drift changes the antenna footprint direction and shifts the DFS centroid. Here $(\gamma)$ and $(\varphi)$ are the expected coordinates of the antenna gain ground projection, while $(\tilde{\gamma})$ and $(\tilde{\varphi})$ are the shifted versions of these coordinates by the attitude misknowledge (adapted from Delouis et al. TS21). **(b)** Apparent pointing $\varphi_a$ for the SKIM geometry. Examples of two-way antenna gain $G$ as a function of azimuth and distortions (exaggerated 100 times) induced by $\sigma^0$ gradients on the power integrated by the radar in the azimuth direction across the antenna diagram (grey curve). Three examples of asymmetric distortions are given: a sine function with $\nu = \sin\theta/\sigma_\alpha$, a sine function varying 3 times faster and a linear trend. Such distortions induce an apparent mispointing of the beam $\delta\varphi$ and a correction to the geometrical line-of-sight relative velocity estimate.

above for SKIM and KuROS respectively translate to timing accuracy requirements of 36.8 and 7.7 m and to surface-tracking accuracy requirements of 80.4 and 16.9 cm.

The timing accuracy requirements are easily met in the spaceborne context but can be challenging in the cost-constrained context of an airborne instrument.

The surface-tracking algorithm, however, does not benefit from the error compensation that exists for the timing error. The requirement for SKIM is easily met by the nadir altimeter payload of SKIM. The 80.4 cm altitude tracking requirement is out of reach of the KuROS airborne instrument. Our analysis of its DFS data will thus be restricted to the side-looking configurations for which, as per Eq. (A24), the pointing requirements are much milder.

## A2.3 Effective pointing and azimuth gradient DFS

As expressed in Eq. (A24), for each range-resolution cell $V_{NG}$ results from an integral over azimuth with a weight that depends on the product of the antenna radiation diagram and the sea surface NRCS, which varies as a function of the horizontal position $(x, y)$ due to the presence of waves, varying winds, currents, surfactants, sea ice and all the physical properties of the sea surface.

Even with perfect knowledge of the platform attitude and velocity, NRCS variations can thus make the effective pointing of the measurements deviate from the pure geometric estimates. Valuable insight into this effect can be gained by considering the saddle-point approximation of Eq. (A24) in the limit of a very narrow antenna diagram (which is clearly applicable for SKIM and KaRADOC, less so for KuROS).

Considering first the case of an antenna pointing towards azimuth $\varphi_b$ with an infinitely narrow radiation diagram, we see that the product $W(t_1, r', \varphi)\widetilde{\sigma^0}(t_1, r', \varphi)$ is well approximated by the Dirac distribution $\delta(\varphi - \varphi_b)$. In this limit,

$$V_{NG}(t_1, r') = V_R \left[ \cos(g)\sin(\theta)\cos(\varphi_b - c) + \sin(g)\cos(\theta) \right]. \tag{A27}$$

We recognize in this expression $V_{geo}$, the estimate of $V_{NG}$ one would have derived using direct geometric arguments.

The essence of the argument is that the sharpest factor in the integral is the beam radiation diagram. If it is now not infinitely sharp, we see that the effect of a gradient of $\sigma^0$ is to shift the peak of the distribution by an angle

$$\delta\varphi = -\frac{\partial_\varphi \log\left(\widetilde{\sigma^0}\right)\big|_{\varphi_b}}{\partial_{\varphi\varphi} \log(W)\big|_{\varphi_b}}. \tag{A28}$$

Assuming for $W(t_1, t', \varphi)$ a Gaussian approximation,

$$W(t_1, r', \varphi) = \frac{1}{\sqrt{\pi}\sigma_\varphi(r')} \exp\left[-\frac{(\varphi - \varphi_b)^2}{\sigma_\varphi^2(r')}\right], \tag{A29}$$

in which $\sigma_\varphi(r')$ is a parameter describing the width of the antenna diagram at the working incidence angle, one obtains

$$V_{NG}(t_1, r') = V_R \left[ \cos(g)\sin(\theta)\cos(\varphi_b - c + \delta\varphi) + \sin(g)\cos(\theta) \right] \tag{A30}$$

with

$$\delta\varphi = \frac{\sigma_\varphi^2(r')}{2} \partial_\varphi \log(\widetilde{\sigma^0}). \tag{A31}$$

Alternatively, one can choose to express $V_{NG}$ as the sum of $V_{geo}$, the geometric approximation, plus an azimuth gradient Doppler velocity contribution,

$$V_{NG}(t_1, r') = V_{geo}(t_1, r') + V_{AGD}(t_1, r'), \tag{A32}$$

with

$$V_{geo}(t_1, r') = V_R \left[ \cos(g)\sin(\theta)\cos(\varphi_b - c) + \sin(g)\cos(\theta) \right] \tag{A33}$$

and

$$V_{AGD}(t_1, r') = - V_R \cos(g)\sin(\theta)\sin(\varphi_b - c) \frac{\sigma_\varphi^2(r')}{2} \partial_\varphi \log\left(\widetilde{\sigma^0}\right). \tag{A34}$$

One can see from these expressions that for a given azimuthal variation of the NRCS the order of magnitude of $V_{AGD}$ is set by the width of the antenna radiation diagram: instruments with a thin diagram, such as SKIM and KaRADOC, are less affected than instruments with a broader diagram, such as KuROS. Also, one sees that $V_{AGD}$ is largest when the instrument looks in the cross-track direction and is zero in the up-track and down-track viewing directions. Finally, one sees that $V_{AGD}$ is equivalent to the line-of-sight projection of a spurious horizontal velocity $U_{AGD}$, which varies with incidence angle only through the variations of $\sigma^0$ and $\sigma_\varphi$:

$$U_{AGD}(t_1, r') = -V_R \sin(\varphi_b - c) \frac{\sigma_\varphi^2(r')}{2} \partial_\varphi \log\left(\widetilde{\sigma^0}\right). \tag{A35}$$

At small scales, spatial gradients add to the azimuthal gradient and also induce a spurious velocity with the same expression as a function of $\sigma^0$. Using the simple case of a single Fourier component $\sigma^0 = \varepsilon \sin[\nu(\varphi - \varphi_b)]$ allows one to evaluate the importance of different scales. The azimuthal shift can be obtained as

$$\delta\varphi = \varepsilon \exp\left(-\frac{(\nu^2 + 1)\sigma_\varphi^2}{4}\right) \sinh\left(\frac{\nu\sigma_\varphi^2}{2}\right). \tag{A36}$$

In the slow variation limit ($\nu, \sigma_\varphi \to 0$) and Eq. (A36) this expression coincides with Eq. (A31). For faster variations, one sees that the largest disturbance is obtained when $\nu \sim \sqrt{2}/\sigma_\varphi$. This azimuthal wavenumber is such that the footprint can host a bright and a dark patch, one on either side of the look direction. This configuration creates the largest disturbance for a given value of the brightness contrast $\varepsilon$. $\delta\varphi$ in this case is given by

$$\delta\varphi_{max} = \varepsilon\sigma_\varphi e^{-1/2}/\sqrt{2}. \tag{A37}$$

## Appendix B: KuROS antenna diagram determination

A precise determination of the antenna diagram is necessary for any Doppler application, given the possibly large contribution of pointing errors $\varphi_b - \widetilde{\varphi}$ in the estimation of the non-geophysical DFS and the effect of the antenna beamwidth in the spurious azimuth gradient velocity $U_{AGD}$. A comprehensive strategy has thus been developed for estimating the one-way antenna diagram in amplitude and phase by combining anechoic chamber measurements and verification using the campaign data with a final adjustment of systematic phase shifts in the data. In this section $\alpha$ and $\beta$ are respectively the latitude and longitude of a set of spherical coordinates centered on the antenna such that the main lobe extends in a longitudinal sector on the Equator $\alpha = 0°$, and the rotation axis of the antenna turntable points towards $\alpha = 0°$ and $\beta = 0°$. With this choice of coordinates the antenna diagram has separable Gaussian dependencies on $\alpha$ and $\beta$. In constant-altitude flight, when the antenna points towards $\varphi_b$, $\sin(\alpha) = \sin(\theta)\sin(\varphi - \varphi_b)$ and $\tan(\beta) = \tan(\theta)\cos(\varphi - \varphi_b)$.

## B1 Fixed-antenna NRCS correction

The anechoic chamber measurements are very accurate for the antenna alone. However, once integrated into the plane, the antenna diagram is perturbed. This is, for instance, particularly noticeable in the NRCS measurements in rotating mode, wherein a spurious azimuthal pattern could clearly be seen, and for fixed-antenna DFS observations, wherein a "striping" pattern as a function of incidence angle is obvious.

We have thus developed a complementary method that relies on the variations of the plane attitude during maneuvers. Using the plane IMU, we identify the angular coordinates $\alpha$ and $\beta$ of the nadir and use the measured power to map the antenna diagram (using as a reference point the constant-altitude return power values for each data segment to account for geophysical nadir NRCS variations). The combination of all the flights during the campaign gives the distribution of measured power as a function of $\alpha$ and $\beta$, which is shown in Figs. B1 and B2.

The measured distribution is well approximated by a Gaussian shape

$$G(\alpha, \beta) = \exp\left[-\frac{\alpha^2}{2\sigma_\alpha^2} - \frac{(\beta - \beta_0)^2}{2\sigma_\beta^2}\right]. \tag{B1}$$

Another expression for $G(\alpha, \beta)$, more suitable for use with the half-power beamwidths $\alpha_{-3\,dB}$ and $\beta_{-3\,dB}$ obtained from anechoic chamber measurements, is

$$G(\alpha, \beta) = 2^{\left[-\frac{4\alpha^2}{\alpha_{-3\,dB}^2} - \frac{4(\beta - \beta_0)^2}{\beta_{-3\,dB}^2}\right]}. \tag{B2}$$

The width parameters in these equations are linked by

$$\sigma_\alpha = \alpha_{-3\,dB}/\sqrt{8\log(2)}, \quad \sigma_\beta = \beta_{-3\,dB}/\sqrt{8\log(2)}. \tag{B3}$$

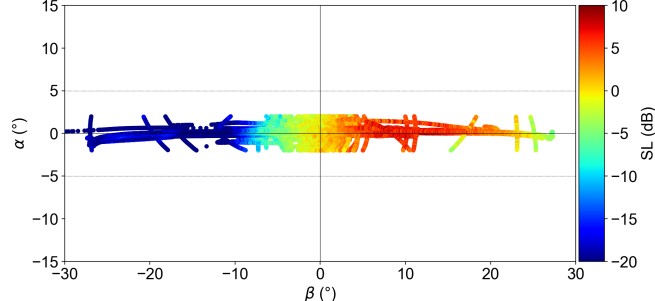

**Figure B1.** Reconstructed $\alpha$ and $\beta$ dependence of the two-way KuROS antenna diagram. For each 30 s data segment, the constant-altitude values, for which the nadir is at $\alpha = 0$ and $\beta = 0$, have been used as a reference level to account for geophysical variations in nadir NRCS.

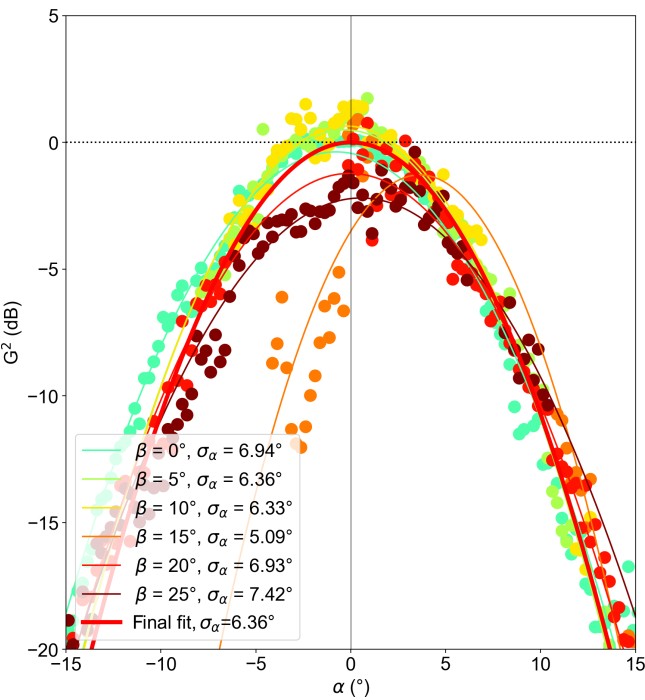

**Figure B2.** Reconstructed azimuth dependence of nadir return power for different incidence angles. For each incidence angle, the $\alpha = 0$ value has been used as a reference. The thick line shows the final Gaussian fit used in the data analysis. The $\beta = 15°$ data were excluded from the fit.

The parameter values used in this study are collected in Table 1.

One cautionary remark is that the illuminated patch at nadir is not infinitely sharp. The measured distribution is thus the convolution of the true antenna diagram by the power distribution at the nadir patch (which depends on the altitude-tracking error and the sea state; Chelton et al., 1989). Assuming Gaussian shapes, the squares of the width parameters

add, leading to

$$\sigma_{\text{observed}} \simeq \sigma_{\text{true}} \left[ 1 + \frac{\sigma_{\text{patch}}^2}{2\sigma_{\text{true}}^2} \right].$$
(B4)

The broadening of the diagram due to finite nadir patch size is thus a small correction provided the scale of the nadir patch remains smaller than the antenna diagram scales. For reasonable orders of magnitude of the altitude-tracking error and significant wave height, the patch $-3\,\text{dB}$ width is of the order of $3°$ when viewed from $3000\,\text{m}$ of height. This corresponds to a $3\,\%$ correction on the value of $\sigma_\alpha$. We have chosen to neglect this correction. The values summarized in Table 1 are the parameters of the Gaussian fits to the observed distributions.

## B2 Rotating-antenna NRCS correction

Using these parameters as a starting point, we have then constructed corrections for the rotating-antenna measurements of NRCS by allowing the boresight elevation $\beta_0$ to vary as a function of antenna orientation within the plane. The variation law was determined by minimizing the dependence of the rotating-antenna NRCS measurements as a function of flight direction over the offshore area for each day.

## B3 Fixed-antenna DFS correction

In a similar way, we have observed that the KuROS antenna diagram is slightly "wrinkled" in that the beam boresight azimuth changes as a function of elevation. This azimuthal mispointing transposes immediately into a striping modulation of the $U_{\text{GD}}$ estimates. A correction was introduced by allowing the boresight azimuth $\alpha_0$ to vary as a function of $\beta$. The variation law of $\alpha_0$ was determined by minimizing the average $U_{\text{GD}}$ over all flights for each value of $\beta$. As the variation of this quantity with respect to $\alpha_0(\beta)$ is not trivial, this required constructing, regularizing and inverting the observation matrix.

https://doi.org/10.5194/os-16-1-2020

**Appendix C:  KaRADOC system**

KaRADOC is built around an Agilent PNA-X network ana-
lyzer, complemented by a TX power amplifier, a T/R switch,
an RX low-noise amplifier and a high-gain purpose-built
slotted waveguide antenna (shown in Fig. C1).

The beam can be steered in elevation by changing the in-
strument working frequency (see Fig. C2a), and the antenna
is usually mounted on a pitch–roll stabilization platform.
For the DRIFT4SKIM experiment, however, the antenna was
10 rigidly mounted in a port-looking configuration centered on a
10° incidence angle with a 2° backward-looking tilt to com-
pensate for the aircraft pitch in constant-altitude flight. Plane
attitude variations were accounted for in the data process-
ing. Observations were collected at 33.7 GHz, correspond-
15 ing to a 12° nominal incidence angle. Other angles were also
scanned, but RF leakage from the TX to the RX subsystems
was too strong at the corresponding frequencies, making the
signal harder to analyze.

The antenna radiation diagram is very narrow, with a
20 beamwidth less than 1.5° in elevation and less than 2° in az-
imuth (see Fig. C2b). Figure C3a represents sections across
the KaRADOC main lobe in the azimuth and elevation direc-
tion at 33.7 GHz.

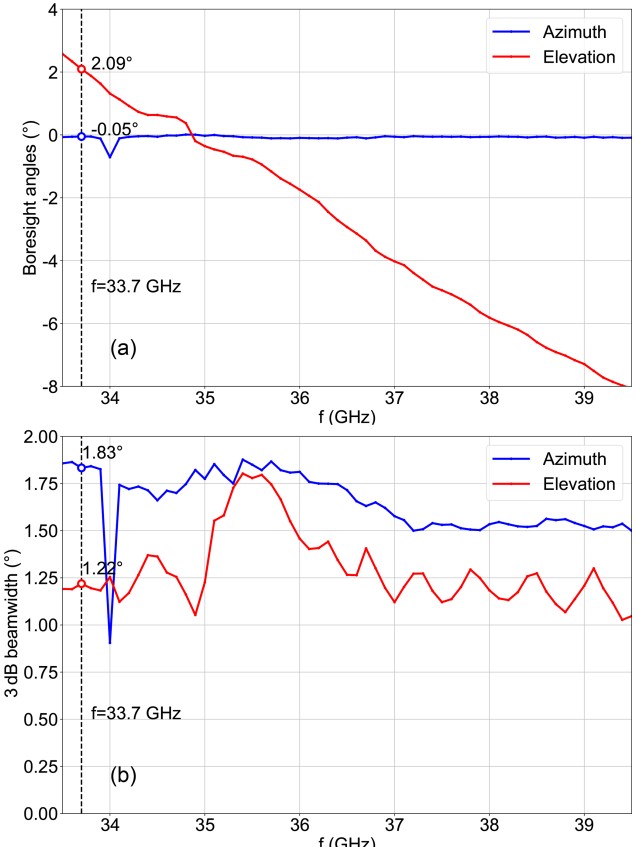

**Figure C2.** Frequency dependence of the KaRADOC main lobe
azimuth and elevation boresight angles **(a)** as well as half-power
beamwidths **(b)**.

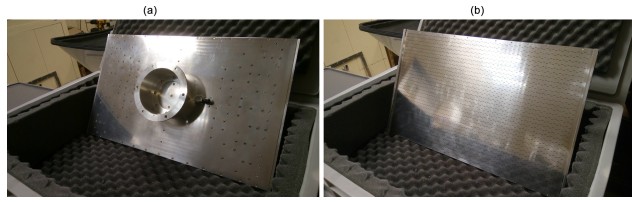

**Figure C1.** The back **(a)** and front **(b)** of the antenna.

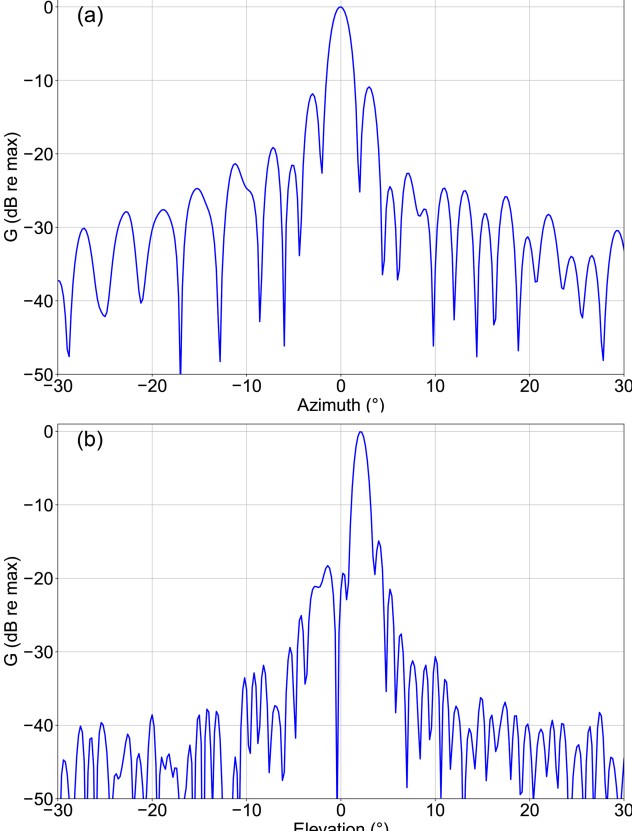

**Figure C3.** KaRADOC radiation diagram at 33.7 GHz as a function of (**a**) azimuth at 2.09° elevation and (**b**) elevation at −0.05° azimuth.

*Code and data availability.* Data and numerical model results presented in this article are available via FTP at the following address: http://tinyurl.com/SKIMftp TS22; they will become more easily accessible through the upcoming website of the ESA-funded IASCO project.

*Author contributions.* All authors have contributed to the writing of the paper. TS23

*Competing interests.* The authors declare that they have no conflict of interest.

*Disclaimer.* The views and opinions expressed in this publication can in no way be taken to reflect the official opinion of the European Space Agency.

*Acknowledgements.* This study was supported by the KuROS4SKIM, DRIFT4SKIM and IASCO contracts from the European Space Agency, made possible by the unflagging determination of Erik de Witte. The in situ measurements owe much to the dedication of the R/V *Thalia* crew. Airborne data were obtained using the aircraft managed by SAFIRE, the French facility for airborne research, part of the infrastructure of the French National Center for Scientific Research (CNRS), Météo-France and the French National Center for Space Studies (CNES). Many people at LOPS and OceanDataLab also contributed to the preparation, deployment and recovery of the instruments, including Mickael Accensi, Sylvain Herledan, Gilles Guitton, Lucile Gaultier, Michel Hamon, Olivier Péden, Stéphane Leizour and Pierre Branellec. Many people at IETR are involved in the KaRADOC developments: Cécile Leconte, Mohamed Himdi, Paul Leroy, Eric Pottier, and especially Guy Grunfelder and Mor Diama Lo, who made the measurement campaign possible during November 2018. Operation of KuROS during the experiment would not have been possible without the dedication of Christophe Le Gac, Nicolas Pauwels and Christophe Dufour from CNRS/LATMOS. We finally thank Roland Romeiser for his contribution to the online discussion of this article, as well as Ernesto Rodriguez and a second (anonymous) referee for their comments and suggestions, which led to major improvements of this article.

*Financial support.* This research has been supported by the European Space Agency (grant no. DRIFT4SKIM). TS24

*Review statement.* This paper was edited by John M. Huthnance and reviewed by Ernesto Rodriguez and one anonymous referee.

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

## Remarks from the language copy-editor

## Remarks from the typesetter

TS32   Please provide pages.
TS33   Please provide pages.
TS34   Please provide pages.