# Peer review of "Measuring Ocean Total Surface Current Velocity with the KuROS and KaRADOC airborne near-nadir Doppler radars: a multi-scale analysis in preparation of the SKIM mission"

_Ocean Science, 2019_

## Short Comment (SC1) · 2 Jan 2020

This is an interesting and well-written paper on the analysis of data from two airborne near-nadir Doppler radar systems. The measurements and data analysis simulate to some extent what could be achieved with the proposed spaceborne radar system SKIM in terms of ocean wave and current measurement techniques and capabilities. A variety of data interpretation / data processing issues are discussed, and some findings regarding the agreement between experimental results and numerical model results, the accuracy of retrieved surface current vectors, and implications with respect to SKIM

are presented. I think this is valuable work that can become the basis of future technical documents on actual SKIM data products. The paper is a little long, but this detailed discussion with a number of equations and diagrams is adequate for a comprehensive evaluation of the SKIM concept and valuable for the identification of issues for further improvement.

I didn't notice any obvious technical mistakes or controversial discussion points, so most of the following comments are of a purely cosmetic nature. For my taste, this paper is almost ready to be published in a final version.

– A few language issues: I noticed that "incidence" is used as a standalone word at some places where "incidence angle" would be more appropriate. Similarly, "the Doppler" should usually be "the Doppler velocity", or maybe "the Doppler frequency" for some occurrences. And does this journal accept the use of "data" as a singular word?

– SKaR, which occurs twice on page 38, is not defined.

– The formatting of some equations is strange. Most equations are formatted flush-left with an equation number at the right border, but two equations on page 5, two on page 37, and two on page 44 are centered with no numbers. Equations (A1), (A3), (A6), (A13)-(A14), and (A22) have strange breaks in them.

– In figures 2 and A1B, there is no axis text on the vertical axes. In figure 10, it is a little difficult to understand the meaning of the four vertical axes, and it is not clear why A and C have numbers on the horizontal axis and B and D don't.

– Finally, here is one technical comment: In section 5, "Implications for SKIM", it is said that wave spectral information from a buoy is generally sufficient for estimating wave contributions to the Doppler velocity. Yes, but shouldn't SKIM be able to estimate wave spectral parameters from its own data? My assumption so far has been that with the amount of information contained in SKIM raw data, it should be possible to estimate

wave spectral parameters and surface current vectors without a need for additional (external) input data. It should be clarified in the text whether this is indeed the ultimate goal for SKIM or not.

---

## Referee Comment (RC1) · Ernesto Rodriguez (Referee) · 26 Feb 2020

**Evaluation of "Measuring ocean surface velocities with the KuROS and KaRADOC airborne near-nadir Doppler radars: a multi-scale analysis in preparation of the SKIM mission" by Marié et al.**

Ernesto Rodriguez

February 26, 2020

**1 Summary Evaluation**

The SKIM mission is based on the concept of measuring total surface velocity using near-nadir Doppler scatterometry. One of the critical factors in the feasibility of this concept is demonstrating the ability to remove the velocity signature of gravity waves, which, following previous work by Nouguier et al. (2018), can be 20 to 30 times the value of the Stokes drift. This can result in wave induced signatures on the order of 2 m/s to 3 m/s, which are more than an order of magnitude greater than the desired current accuracy.

The main purpose of this paper is to demonstrate that this is feasible using the current model. To show this, the team has deployed two Doppler scatterometers (at Ku and Ka-bands) together with significant *in situ* resources, including a buoy to obtain surface

wave spectra, HF-radar, and two kinds of drifters drogued at different depths. The final results of the paper show a good agreement between the theory of Nouguier et al. at Ka-band (although see detailed comments below), the band proposed for SKIM, but poor agreement at Ku-band and a different frequency dependence between Ka and Ku than predicted by the theory.

The experiment was carefully and thoughtfully designed and the team has made a significant effort to characterize the instruments, especially as regards the mean behavior of the signal. Some discussion has been devoted to the effects of antenna beamwidth at Ku-band leading to contamination of the Doppler signal due to the variation of the radar cross section within the radar footprint. However, given the qualitative discrepancy between theory and observations, additional effort should be devoted to quantifying the measurement errors to show that the Ku-band observations could be compatible with the theory, given feasible measurement uncertainties. Alternately, physical sources for the discrepancy should be identified for future avenues of study. A more detailed suggestion is given below.

Overall, the paper has a logical outline. However integration of the different sections into a consistent style and level of detail has not been as successful, leading to some repetition and confusion, at times. The paper would benefit by a final integration to sharpen the presentation into a more uniform manuscript.

In spite of these reservations, I think that the data collected are an important data set that should be in the open literature and recommend its publication, hopefully after some of the more detailed comments below have been addressed. I recommend that the authors consider putting the data in the public domain, so that it can serve to lay the groundwork for work that will strengthen the case for the SKIM mission.

**2 Error Quantification**

Although there is a numerical discussion of various error sources (especially biases due to the antenna pattern and azimuthal variations of backscatter cross section), there is no attempt at deriving an error budget for either of the instruments. This would not be important if the observed measurement scatter were small. However, it is far from small, as can be seen in Figures 12, 13 and 16, where measurement standard deviations varying from 1 m/s to 2 m/s can be observed. Figure 12 is very enlightening about the variation characteristics of the Ka-band measurements, and an equivalent version would have been very useful for Ku-band. For SKIM, it is important to show that not only the model predicting the mean behavior is understood, but also that the error performance is understood. Currently, this information is not contained in the paper, but all the data are available to produce this validation.

The error budget should contain, at least: 1) Expected measurement random velocity errors, which can be calculated in a straightforward fashion from the pulse pair correlation. 2) Contributions from pointing errors. For KuROS, the incidence angle is very well constrained by the high range resolution (although platform elevation couples in at shallow angles, as noted by the authors), but this is not the case for KaRADOC, where a single footprint is used. Typical aircraft roll (and, to a lesser extent, pitch) variations will lead to variations in the local incidence angle of up to a few degrees (leading to large errors, if uncorrected) , and it is not clear in the description of the processing how these effects are mitigated. 3) Error bounds on the possible Doppler effects due to uncertainties in the antenna pattern. 4) Error bounds on the expected effects of the sigma0 azimuth modulation errors as a function of azimuth, which can be obtained using the wavelength of the resolved waves, shown in Figure 9. 5) Modeling assumptions (see below).

Both radar systems have high PRF to properly sampling the Doppler. Is the contamination due to range ambiguities significant? Has it been considered as a source of

[Figure]

error?

Examination of Figure 12 shows passes in the east-west direction have lower levels of variations than those going north-south. In addition, the frequency of variation is higher on the 22nd than on the 24th, but the amplitude of variability is larger on the 24th. What is the reason for this? It does not seem to align with wind or wave directions. In any case, the characteristics of the variations seem to be long-wavelength, leading one to suspect either attitude errors or errors due to the changes in the surface field characteristics. Examining the equivalent noise characteristics of the Ku-band data would potentially help in understanding the differences between the two frequencies.

One observation is that, comparing the variations in Figure 16 and 13, the level of within track variability is **smaller** for Ku band than for Ka-band. Thus the lack of agreement with the model is not due to higher random noise (as could be expected from wave sigma0 contamination), but through some systematic azimuth dependent effect. One potentially useful exercise is to assume that the azimuth brightness gradient contains additional harmonics to the ones estimated in going from Fig. 16a to 16b. Is it possible to account for the divergence from the model with these higher harmonics? If so, are these excluded by the sigma0 observations? Can they be ascribed to systematic coupling that might happen between the antenna pointing and the attitude? If these explanations are not feasible, does this indicate that additional physics needs to be incorporated into the model (at least at Ku-band)?

**3  Modeling and retrieval issues**

There seems to be some mixed messages regarding the modeling assumptions. In Nouguier et al. (2018), a Gaussian assumption is made throughout. On the other hand, the authors quote the asymmetry and skewness of the slope distribution (with references to Munk (2008) and Chapron et al. (2202)) in order to explain the

upwind/downwind asymmetry in the Ku-band backscatter cross-section (Figure 10), which is not insignificant. In equation 16, the isotropic backscatter curves of Nouguier et al. (2016) are used, but they are multiplied by an azimuthal modulation factor $F(\varphi)$, which is not in the original paper and which does not seem to show up again in the analysis. Was such a factor used? If so, is it related to the azimuthal modulation factor quoted in the azimuth modulation fits quoted (but whose values are never given) in the second paragraph in page 21? If not, where is it coming from? Backscatter data are collected at Ku-band and presented in Figure 10A. Do these backscatter data fit the model in equation 16? If so, are the azimuthal modulations derived from these data for both Ku and Ka? If not, is there a justification for using equation 16 when it does not match the data?

In the Nouguier et al. (2018) paper, there are two models presented: one for range resolved or not range resolved Dopplers. Since KaRADOC is not resolving the waves, I assume that the second model is used. This model contains two parts (equation 15, Nouguier et al. (2018)), one which dominates along the wave direction, and another one which has contributions at other azimuths. In this paper, only one terms seems to have been kept (i.e., equation 15, Nouguier et al. (2018)). What is the justification for neglecting the second contribution at other azimuths?

It is well known that non-Gaussian effects will lead to a correlation between the modulation of the slope rms and the location along the wave phase. This effect leads to the EM bias in altimetry, for example. Will the level of modulation consistent with EM bias results lead to a change in the predictions made by the model? Will it lead to an upwind-downwind asymmetry in the Doppler? Can it partially account for the 10-percent adjustment that had to be made to make the model predictions fit the data?

In the retrieval of the surface currents, it was assumed that the current in the scene remained constant. However, as shown in Table 2 and Figure 7, there was significant change in the currents due to tidal variations measured by the Trefle buoy. How was this accounted for during the fitting? The HF-radar imager linked to in the paper also

show some current gradients in the region: were the observable by the radars?

Table 2 also shows significant disagreement between the Trefle buoy velocities and those from the other *in situ* data. Could you comment on the source of discrepancy?

**4  Miscellaneous comments**

Figure 5 appears with insufficient attribution or description.  Part of it comes from Nouguier et al. (2016), but there are additional subpanels whose provenance should be clarified.

The term $\text{mss}_{shape}$ is introduced with just a reference to Nouguier et al 2016. To make things easier for the reader, it should be clarified that it is the apparent rms slope obtained by fitting the backscatter curves.

In page 32, there is a statement made about the equivalent depth of the measurements from near-nadir Doppler scatterometry. However, no such derivation is presented in the papers referenced. It would be useful to the community of this statement were backed with a calculation for the two wind speeds (perhaps as an appendix).

---

## Referee Comment (RC2) · Anonymous Referee #2 · 2 Mar 2020

Review on the paper "Measuring ocean surface velocities with the KuROS and KaRADOC airborne near-nadir Doppler radars: a multi-scale analysis in preparation of the SKIM mission"
by L. Marié et al.

This paper presents the technique and examples of current velocity measurements from an airborne platform carrying two radars. The data acquisition was performed at a site located off the western coast of France where the surface currents are continuously monitored by two HF radars. Surface drifters were deployed for validation of airborne velocity measurements. The paper provides a detailed discussion of the experimental setup, measurement technique, errors, and comparison of the velocity data from different sensors.

I find the paper very deep, well worth publishing, providing very valuable information for developers of radars for velocity measurements from space. All figures are of excellent quality. Congratulations to the others on a good paper, but an even more impressive field campaign.

However, the paper would benefit from the following changes (not only minor).

- The text needs a substantial reduction: 33 pages and 15 pages in Appendix, this is too much. Please make sure authors are happy with this? Many formulas and demonstrations come from the paper of Nouguier et al., 2018 and Rodriguez et al., 2018. The saving is worth it.

- The text needs a closer proof reading. There are some typos (altitude/attitude in page 39, 40; are/is following the word data within the whole paper, Appendixes/appendices in page 16, ...

- The main body of the paper requires a number of changes to the text where it appears confused while Appendixes are well written and very clear.

Specific points.

Abstract: what is the major finding in this study? Only an estimate of C0? The description of the experiment should be shortened giving the place to the main results.

P3 L15 something is wrong with the English of this sentence? The contribution ... of contributions

P4 L2 measurement equation. Maybe measurement is not necessary?

P9 Figure 4 caption: contribute to or contribution to. "to" is missed.

P10 Some problems with the English in many places. L1-2: the sentence seems not finished. L7: U is the current speed ... L8 wave slope variability? spectrum.  L16 While the incidence angle increases ... the backscatter becomes dominated

L27,30. eq. 14 contains phi or phi_s? it is confusing.

P11 L24: something gone wrong in this sentence.  ... work was focused in two boxes. Perhaps, work performed in locations matching by two boxes in Figure 6 ...

P13-P14. The text is very confusing and should be re-written.

P16 L8. Please check for frequency and remove band if only one frequency is used. L12 How to understand the ambiguity of 126 m/s ?

P17 L8 Consider: observations corresponding to Phi=12deg are reported.

P19 L1-4. Please remove repetition in this sentence: 30 seconds

P21 L12: Consider: Due to the narrower radar beam, the data from Karadoc are easier to interpret than the data from Kuros. L14 and P22 L1-2: something is wrong with the English in these lines.

P23 Figure 13 caption: remove one "blue" and complete the sentence.

P26 L1 Consider ... spectra estimated from measurements on November 2 ...

L4 energy is much lower than

P31 P7 Perhaps: Regarding the radar measurements, ...

P33 L19-21. This conclusion is confusing and should be re-written.

---

## Author Comment (AC1) · 7 May 2020

We thank Dr Romeiser for his careful reading of our manuscript, and for taking the time to contribute this positive appraisal of our work to the interactive discussion of the article. We are currently drafting a revised version to address a number of issues raised by him and the other reviewers. In the following we detail what modifications we have performed, or intend to perform, in response to his comments.

I didn't notice any obvious technical mistakes or controversial discussion points, so

most of the following comments are of a purely cosmetic nature. For my taste, this paper is almost ready to be published in a final version.

Again, we thank Dr Romeiser for his positive appraisal of our work.

– A few language issues: I noticed that "incidence" is used as a standalone word at some places where "incidence angle" would be more appropriate. Similarly, "the Doppler" should usually be "the Doppler velocity", or maybe "the Doppler frequency" for some occurrences. And does this journal accept the use of "data" as a singular word?

We thank Dr Romeiser for pointing out these issues, which we have corrected in our revised manuscript.

– SKaR, which occurs twice on page 38, is not defined.

The definition of the "SKIM Ka-band Radar" has been added in the introduction section, which has been quite extensively rewritten in response to comments by the referees.

– The formatting of some equations is strange. Most equations are formatted flush-left with an equation number at the right border, but two equations on page 5, two on page 37, and two on page 44 are centered with no numbers. Equations (A1), (A3), (A6),(A13)-(A14), and (A22) have strange breaks in them.

We have formatted the equations of pages 5, 37 and 44 flush-left and numbered them. The "strange breaks" in the mentioned equations of Appendix A were introduced in anticipation of the two-column formatting necessary for publication in Ocean Science. We have corrected them in the revised manuscript (and paid attention to the appearance of the equations in two-column format).

– In figures 2 and A1B, there is no axis text on the vertical axes. In figure 10, it is a little difficult to understand the meaning of the four vertical axes, and it is not clear why A and C have numbers on the horizontal axis and B and D don't.

We have corrected figures 2, 10, 11 and A1B.

– Finally, here is one technical comment: In section 5, "Implications for SKIM", it is said that wave spectral information from a buoy is generally sufficient for estimating wave contributions to the Doppler velocity. Yes, but shouldn't SKIM be able to estimate wave spectral parameters from its own data? My assumption so far has been that with the amount of information contained in SKIM raw data, it should be possible to estimate wave spectral parameters and surface current vectors without a need for additional (external) input data. It should be clarified in the text whether this is indeed the ultimate goal for SKIM or not.

We have clarified this point in section 5. Clearly, the aim in the SKIM context is to estimate the UWD contribution from the wave spectral information retrieved from the SKaR instrument. This was not performed for the present work in order to avoid compounding uncertainties associated with the sea state retrieval with those associated to the Doppler processing itself. The full end-to-end processing will be performed over the forthcoming years in the framework of the ESA-funded IASCO project and is expected to be the subject of future publications.

---

## Author Comment (AC2) · 10 May 2020

The comments of the reviewer are copied in bold and our reples are in normal font.

**This paper presents the technique and examples of current velocity measurements from an airborne platform carrying two radars. The data acquisition was performed at a site located off the western coast of France where the surface currents are continuously monitored by two HF radars. Surface drifters were deployed for validation of airborne velocity measurements. The paper provides a**

**detailed discussion of the experimental setup, measurement technique, errors, and comparison of the velocity data from different sensors. I find the paper very deep, well worth publishing, providing very valuable information for developers of radars for velocity measurements from space. All figures are of excellent quality. Congratulations to the authors on a good paper, but an even more impressive field campaign.**

We thank the referee for his/her careful reading of our manuscript, and for taking the time to contribute this positive appraisal of our work to the interactive discussion of the article. We are currently drafting a revised version to address a number of issues raised by him and the other reviewer. In the following we detail what modifications we have performed, or intend to perform, in response to his comments.

**However, the paper would benefit from the following changes (not only minor):**

**– The text needs a substantial reduction: 33 pages and 15 pages in Appendix, this is too much. Please make sure authors are happy with this?**

We understand the concern expressed by the reviewer, and will do our best to strike a balance between the need to make this article shorter and easier to read and a conflicting request from reviewer 1 which would on the opposite require us to add significant new material. We will do our best to streamline the flow of the paper and avoid unnecessary duplication. As for the length, we prefer to have longer appendices and a shorter main text in order to keep all the material in the same document without requiring the reader to go back to reports and grey literature. In fact, the present manuscript is the first time that the science behind near-nadir Doppler scatterometry for surface current is explained and demonstrated on real data. There are indeed some common aspects with higher incidences treated in Rodriguez (2018) and Nouguier et al. (2018), but the difference in incidence angle makes some of our analysis very specific. Also, the very broad radiation diagram of the KuROS instrument was the occasion for a detailed analysis of the averaging process by which the line-of-sight

Doppler shift emerges from an extended field-of-view. This research, summarized in Appendix A, is original, and may be relevant for future satellite instruments, for which the radiation diagram will be better controlled, but for which the very large platform velocity will make the requirements much more stringent.

**Many formulas and demonstrations come from the paper of Nouguier et al., 2018 and Rodriguez et al., 2018. The saving is worth it.**

As mentioned above, though the cited articles have been an important source of inspiration, we had to adapt significantly the concepts they developed to the near-nadir observations performed during Drift4SKIM. We will do our best to remove any duplication of these articles and to shorten our text wherever possible in our revised version.

**–The text needs a closer proof reading. There are some typos (altitude/attitude in page 39, 40; are/is following the word data within the whole paper, Appendixes/appendices in page 16, ...**

A thorough search of the text for such issues is ongoing.

**–The main body of the paper requires a number of changes to the text where it appears confused while Appendixes are well written and very clear.**

We are in the process of streamlining and clarifying the main text, which may have suffered from the large time allocation devoted to the Appendixes for the original submission. . ..

**Specific points. Abstract: what is the major finding in this study? Only an estimate of C0? The description of the experiment should be shortened giving the place to the main results.**

We agree the abstract is currently not suitable, and will rephrase it in our revised version, taking into account the comments from the referee as well the evolution of the text.
**P3 L15 something is wrong with the English of this sentence? The contribution ... of contributions**

This sentence has been corrected.

**P4 L2 measurement equation. Maybe measurement is not necessary?**

This sentence has been corrected.

**P9 Figure 4 caption: contribute to or contribution to. "to" is missed.**

This sentence has been corrected.

**P10 Some problems with the English in many places. L1-2: the sentence seems not finished. L7: U is the current speed ... L8 wave slope variability? spectrum. L16 While the incidence angle increases ... the backscatter becomes dominated L27,30. eq. 14 contains $\phi$ or $\phi_s$? it is confusing.**

On the basis of the comments from both referees, we intend to thoroughly rearrange the text of section 2. We will be very careful to make that important section clear, easy to read, and syntactically correct.

**P11 L24: something gone wrong in this sentence. ... work was focused in two boxes. Perhaps, work performed in locations matching by two boxes in Figure 6**
. . .
We have removed references to "boxes", and used the word "area" instead in the text.

**P13-P14. The text is very confusing and should be re-written.**

We have done our best to clarify the text of sections 3.1 and 3.2.

**P16 L8. Please check for frequency and remove band if only one frequency is used.**

This sentence has been corrected.

**L12 How to understand the ambiguity of 126 m/s ?**

We have been more explicit in our discussion of ambiguity in section A1.4. 126 m/s is equal to the upper bound of the unambiguous velocity interval at the KuROS wavelength and PRI.

**P17 L8 Consider: observations corresponding to Phi=12deg are reported.**

This sentence has been corrected.

**P19 L1-4. Please remove repetition in this sentence: 30 seconds**

This sentence has been corrected.

**P21 L12: Consider: Due to the narrower radar beam, the data from Karadoc are easier to interpret than the data from Kuros.**

We have implemented the referee's suggestion.

**P21 L14 and P22 L1-2: something is wrong with the English in these lines.**

This paragraph has been rephrased.

**P23 Figure 13 caption: remove one "blue" and complete the sentence.**

The caption of Figure 13 has been corrected.

**P26 L1 Consider ... spectra estimated from measurements on November 2 ...**

This sentence has been corrected.

**P26 L4 energy is much lower than**

This sentence has been corrected.

**P31 L7 Perhaps: Regarding the radar measurements,** . . .

This sentence has been corrected.

**P33 L19-21. This conclusion is confusing and should be re-written**

We agree the conclusion is currently not suitable, and will rephrase it in our revised version, taking into account the comments from the referee as well the evolution of the text.

---

## Author Comment (AC3) · 14 May 2020

We have copied the reviewer's comments in bold, with our replies following in normal font.

**1 Summary evaluation.**

**The SKIM mission is based on the concept of measuring total surface velocity using near-nadir Doppler scatterometry. One of the critical factors in the feasibility of this concept is demonstrating the ability to remove the velocity signature**

[Figure]

**of gravity waves,which, following previous work by Nouguier et al. (2018), can be 20 to 30 times the value of the Stokes drift. This can result in wave induced signatures on the order of 2 m/s to 3 m/s, which are more than an order of magnitude greater than the desired current accuracy.**

**The main purpose of this paper is to demonstrate that this is feasible using the current model. To show this, the team has deployed two Doppler scatterometers (at Ku and Ka-bands) together with significant in situ resources, including a buoy to obtain surface wave spectra, HF-radar, and two kinds of drifters drogued at different depths. The final results of the paper show a good agreement between the theory of Nouguier et al. at Ka-band (although see detailed comments below), the band proposed for SKIM, but poor agreement at Ku-band and a different frequency dependence between Ka and Ku than predicted by the theory.**

**The experiment was carefully and thoughtfully designed and the team has made a significant effort to characterize the instruments, especially as regards the mean behavior of the signal. Some discussion has been devoted to the effects of antenna beamwidth at Ku-band leading to contamination of the Doppler signal due to the variation of the radar cross section within the radar footprint. However, given the qualitative discrepancy between theory and observations, additional effort should be devoted to quantifying the measurement errors to show that the Ku-band observations could be compatible with the theory, given feasible measurement uncertainties. Alternatively, physical sources for the discrepancy should be identified for future avenues of study. A more detailed suggestion is given below.**

We thank Dr Rodríguez for his thorough reading of our manuscript, for his many insightful suggestions, which we will do our best to implement, and for waiving his anonymity.

We share Dr Rodríguez's opinion that the paper is not clear enough regarding the reasons for the large discrepancies observed between the Ku-band radar measurements

and the drifter-derived TSCV estimates. Though in our opinion these discrepancies are essentially explained by the very broad KuROS radiation diagram (recalling again that this instrument was not originally designed for this type of measurements), this is not stated explicitly enough in the originally submitted article.

We propose to attempt to reorganize some of the material of sections 2 and 5 in the form of an error budget restricted to its geometrical factors (other error contributions have been thoroughly addressed in Rodríguez et al., 2018) and to prove that our hypothesis is indeed correct. Should this prove impractical, or should this lead to an unreasonable increase in the manuscript length (which has already been mentioned by an Anonymous Referee as problematic), we would at least make an explicit statement of our hypothesis regarding the origin of the discrepancy.

**Overall, the paper has a logical outline. However integration of the different sections into a consistent style and level of detail has not been as successful, leading to some repetition and confusion, at times. The paper would benefit by a final integration to sharpen the presentation into a more uniform manuscript.**

We are in the process of streamlining the text and searching for typos. As requested by an Anonymous Referee, we will also do our best to remove repetitions to reduce the length of the manuscript.

**In spite of these reservations, I think that the data collected are an important data set that should be in the open literature and recommend its publication, hopefully after some of the more detailed comments below have been addressed. I recommend that the authors consider putting the data in the public domain, so that it can serve to lay the groundwork for work that will strengthen the case for the SKIM mission.**

We thank Dr Rodríguez for this appraisal of our work. Ensuring an open access to the Drift4SKIM dataset will be performed in the course of the IASCO project, funded by ESA.

**2 Error Quantification**

Although there is a numerical discussion of various error sources (especially biases due to the antenna pattern and azimuthal variations of backscatter cross section), there is no attempt at deriving an error budget for either of the instruments. This would not be important if the observed measurement scatter were small. However, it is far from small, as can be seen in Figures 12, 13 and 16, where measurement standard deviations varying from 1 m/s to 2 m/s can be observed. Figure 12 is very enlightening about the variation characteristics of the Ka-band measurements, and an equivalent version would have been very useful for Ku-band. For SKIM, it is important to show that not only the model predicting the mean behavior is understood, but also that the error performance is understood. Currently, this information is not contained in the paper, but all the data are available to produce this validation.

The error budget should contain, at least:

1) Expected measurement random velocity errors, which can be calculated in a straightforward fashion from the pulse pair correlation.

2) Contributions from pointing errors. For KuROS, the incidence angle is very well constrained by the high range resolution (although platform elevation couples in at shallow angles, as noted by the authors), but this is not the case for KaRADOC, where a single footprint is used. Typical aircraft roll (and, to a lesser extent, pitch) variations will lead to variations in the local incidence angle of up to a few degrees (leading to large errors, if uncorrected) , and it is not clear in the description of the processing how these effects are mitigated.

3) Error bounds on the possible Doppler effects due to uncertainties in the antenna pattern.

4) Error bounds on the expected effects of the sigma0 azimuth modulation errors
as a function of azimuth, which can be obtained us-ing the wavelength of the resolved waves, shown in Figure 9.

**5) Modeling assumptions(see below).**

As stated above, we understand Dr Rodríguez's position on this issue, and will either produce a formal error budget on the basis of his detailed suggestions, or if this makes the manuscript unpractically long, make the origin of the discrepancy explicit in the revised version.

**Both radar systems have high PRF to properly sampling the Doppler. Is the contamination due to range ambiguities significant? Has it been considered as a source of error?**

We have never seen any sign of this particular issue in the KuROS imagery, and have thus not considered it as a source of error. We expect the issue to be less significant at the near-nadir incidence angles discussed in our manuscript than at the quite large incidence angles typically used by DopplerScatt.

**Examination of Figure 12 shows passes in the east-west direction have lower levels of variations than those going north-south. In addition, the frequency of variation is higher on the 22nd than on the 24th, but the amplitude of variability is larger on the 24th. What is the reason for this? It does not seem to align with wind or wave directions. In any case, the characteristics of the variations seem to be long-wavelength, leading one to suspect either attitude errors or errors due to the changes in the surface field characteristics. Examining the equivalent noise characteristics of the Ku-band data would potentially help in understanding the differences between the two frequencies.**

The data shown in figure 12 have been low-pass filtered to remove the large fast vari-ations due to individual waves. This has been stated explicitly in the caption. We have checked the long-wavelength variations are not linked in a straightforward way to the

plane attitude, and our current position is that they are caused by changes in surface-field characteristics. As regards the Ku-band noise characteristics, as stated above, we currently favor the hypothesis that the difference in antenna radiation diagram is sufficient to explain the large discrepancy between the Ku-band and the Ka-band measurements. Should our analysis of the error budget show that this is not the case, we will investigate in more depth this suggestion.

One observation is that, comparing the variations in Figure 16 and 13, the level of within track variability is smaller for Ku band than for Ka-band. Thus the lack of agreement with the model is not due to higher random noise (as could be expected from wave sigma0 contamination), but through some systematic azimuth dependent effect. One potentially useful exercise is to assume that the azimuth brightness gradient contains additional harmonics to the ones estimated in going from Fig. 16a to 16b. Is it possible to account for the divergence from the model with these higher harmonics? If so, are these excluded by the sigma0 observations? Can they be ascribed to systematic coupling that might happen between the antenna pointing and the attitude? If these explanations are not feasible, does this indicate that additional physics needs to be incorporated into the model (at least at Ku-band)?

As stated above, we currently favor the hypothesis that the difference in antenna radiation diagram is sufficient to explain the large discrepancy between the Ku-band and the Ka-band measurements. Should our analysis of the error budget show that this is not the case, we will investigate in more depth these suggestions.

**3 Modeling and retrieval issues**

**There seems to be some mixed messages regarding the modeling assumptions. In Nouguier et al. (2018), a Gaussian assumption is made throughout. On the other hand, the authors quote the asymmetry and skewness of the slope distribution (with references to Munk (2008) and Chapron et al. (2002)) in order to explain the upwind/downwind asymmetry in the Ku-band backscatter cross-section**

**(Figure 10), which is not insignificant. In equation 16, the isotropic backscatter curves of Nouguier et al. (2016) are used, but they are multiplied by an azimuthal modulation factor F($\varphi$), which is not in the original paper and which does not seem to show up again in the analysis. Was such a factor used? If so, is it related to the azimuthal modulation factor quoted in the azimuth modulation fits quoted (but whose values are never given) in the second paragraph in page 21? If not, where is it coming from? Backscatter data are collected at Ku-band and presented in Figure 10A. Do these backscatter data fit the model in equation 16? If so, are the azimuthal modulations derived from these data for both Ku and Ka? If not, is there a justification for using equation 16 when it does not match the data?**

We will clarify these issues in the text. Clearly, the upwind/downwind asymmetry is not accounted for in our current model. Our rationale in using equation (16) even in this situation was that, though this asymmetry can be observed in the data, it is however strongly dominated by the Gaussian behaviour that the model is based on.

**In the Nouguier et al. (2018) paper, there are two models presented: one for range resolved or not range resolved Dopplers. Since KaRADOC is not resolving the waves, I assume that the second model is used. This model contains two parts (equation 15, Nouguier et al. (2018)), one which dominates along the wave direction, and another one which has contributions at other azimuths. In this paper, only one term seems to have been kept (i.e., equation 15, Nouguier et al. (2018)). What is the justification for neglecting the second contribution at other azimuths?**

We will clarify these issues in the text. Our justification for neglecting the second contribution was that it was practically difficult to estimate from the data, and theoretically subdominant.

**It is well known that non-Gaussian effects will lead to a correlation between the**

**modulation of the slope rms and the location along the wave phase. This effect leads to the EM bias in altimetry, for example. Will the level of modulation consistent with EM bias results lead to a change in the predictions made by the model? Will it lead to an upwind-downwind asymmetry in the Doppler? Can it partially account for the 10-percent adjustment that had to be made to make the model predictions fit the data?**

Though we share Dr Rodríguez's interest in these issues, we have not yet been able to analyze the Drift4SKIM dataset in sufficient depth to identify how we could contribute answers to all these questions. It is definitely in our plans for the forthcoming years to clarify these issues and assess the impact of non-Gaussian behaviour of the sea state on potential SKIM current retrievals, but this was not feasible in the scope of this necessarily limited first analysis of the dataset.

**In the retrieval of the surface currents, it was assumed that the current in the scene remained constant. However, as shown in Table 2 and Figure 7, there was significant change in the currents due to tidal variations measured by the Trefle buoy. How was this accounted for during the fitting? The HF-radar imager linked to in the paper also show some current gradients in the region: were they observable by the radars? Table 2 also shows significant disagreement between the Trefle buoy velocities and those from the other in situ data. Could you comment on the source of discrepancy?**

Once again, these effects, though interesting, were not sufficiently well resolved during the experiment to lend themselves to a thorough analysis. Our approach has thus been to compare time and space averages of the surface current estimates obtained using the different instruments. This unfortunately tends to degrade the agreement, by leaving as "unexplained discrepancies" effects which could be reduced into "resolved variability" by a more careful analysis. We felt this was however still out of the scope of this first account of the Drift4SKIM experiment.

Regarding the disagreement between the Trèfle and other in-situ velocities in Table 2, we suspect a misunderstanding: the data reported as "buoy (Us, Vs)" in the table are the Stokes drift components at the center of the "Offshore" area, estimated from the Trèfle buoy IMU data on November 22nd and from the closest Spotter buoy on November 24th. The figures are indeed markedly different from the drifter velocity data, but are in reasonable agreement with the Stokes drift estimates provided by the WAVEWATCH III model.

**4 Miscellaneous comments**

**Figure 5 appears with insufficient attribution or description. Part of it comes from Nouguier et al. (2016), but there are additional subpanels whose provenance should be clarified.**

Details for each panel have now been added to the caption: (a) The Stokes drift, wave height and wind speed are taken from buoy data at Ocean Station Papa from 2010 to 2017, with wave data is from WMO buoy 46246 maintained by the University of Washington (Thomson et al. 2013) (b) mssshape estimated from GPM satellite back-scatter using modeled co-locataed wind speed and wave height, reproduced from Nouguier et al. (2018). (c) and (d) MWD was computed for a wide range of modeled ocean wave spectra using the theoretical model of Nouguier et al. (2018), and plotted here as a function of the wind speed.

The term mssshape is introduced with just a reference to Nouguier et al 2016. To make things easier for the reader, it should be clarified that it is the apparent rms slope obtained by fitting the backscatter curves. Indeed, the mssshape is a parameter that is a function of the radar wavelength and is obtained from the variation of backscatter with azimuth. This is now introduced and defined with eq. (16) and clarified in the text:

mssshape is a diffraction-effective mean square slope that varies with the radar wavelength and that controls the variation of the backscatter power with the incidence angle (Nouguier et al. 2016).
**In page 32, there is a statement made about the equivalent depth of the measurements from near-nadir Doppler scatterometry. However, no such derivation is presented in the papers referenced. It would be useful to the community of this statement were backed with a calculation for the two wind speeds (perhaps as an appendix)**

We appreciate the importance of this comment. However, given the length and complexity of the present paper we have preferred to keep this discussion to a minimum (mentioning that the different phase speeds are weighted by their contribution to the mssshape), A detailed analysis of the possible influence of the vertical current shear will be given elsewhere. In short, each monochromatic wave train contributes to the back-scatter proportionally to its contribution to the mean square slope. Adapting the theory by Stewart and Joy (1974), we thus expect a measurement depth weighted by the slope spectrum. In practice, considering a realistic simulated wave spectra, this gives an average over the top 1 m of the ocean, compared to 2 m for the 12 MHz HF radar used here as a reference.

We are working on a short note giving the details of the theoretical and expected current measurements in the presence of a vertical shear (Nouguier et al., in prep).

---

## Author Response (AR1)

**Letter to the Editor of "Ocean Sciences" regarding manuscript os-2019-77**

Dear Professor Huthnance,
please find attached a revised version of our manuscript "os-2019-77", which we hope takes into account the remarks formulated by the referees in the previous round of review.

With respect to the previous version, we have made the following changes:

- The title has been slightly amended.

- Dr Peter Sutherland, who contributed during the field work and in the drafting of this second version, has been added as a coauthor.

- The body of the text has been corrected for typos and homogenized, as had been requested by the referees. In this process, most of the text has been amended, but the result is in our opinion much easier to follow.

- The abstract has been rewritten to reflect changes undergone by the rest of the text. We have striven to make our claims and main findings more clearly apparent.

- Section 1 (introduction) has been amended in minor ways.

- Section 2 (theory) has been homogenized and rewritten, with the aim to make it more focused on the particular challenges of near-nadir radar Doppler observations. A subsection pertaining to the overall error budget of the technique, as requested by Ernesto Rodriguez as a referee, has been added.

- Section 3 (description of the field experiment) has been homogenized and rewritten, but did not change substantially.

- Section 4 (results) has been thoroughly rewritten. The text has been clarified in many places. We have re-analysed our data, and managed to explain the major difference observed in the first version between the Ka-band and Ku-band radar measurements. The major instrumental bias present in the Ku-band data in the first version has been corrected in a much better way, and the Ku-band results can now be accounted for by our theory. One finding is also that the directional spread of the sea state seems to bear a stronger influence on the waves-induced contribution to the observed Doppler Frequency Shift than on previous quantities of remote sensing interest (normalized backscattering cross-section). Improving the results of this measurement technique may require developing a better understanding of higher-order statistics of the sea state than previously available and necessary.

- Section 5 (implication for SKIM) and 6 (conclusion and perspectives) have been rewritten accordingly.

One remark we have clearly failed to implement was the request from the Anonymous Referee that the text be substantially shortened. As the text is now more homogeneous in style, clearer and, we hope, much easier to follow, we hope the Referee will forgive this.

Yours sincerely, and on behalf of all the coauthors of manuscript "os-2019-77"

Dr Louis Marié, PhD.

**Point-by-point reply to the report from Reviewer 1**

**1 Summary evaluation.**

**The SKIM mission is based on the concept of measuring total surface velocity using near-nadir Doppler scatterometry. One of the critical factors in the feasibility of this concept is demonstrating the ability to remove the velocity signature of gravity waves,which, following previous work by Nouguier et al. (2018), can be 20 to 30 times the value of the Stokes drift. This can result in wave induced signatures on the order of 2 m/s to 3 m/s, which are more than an order of magnitude greater than the desired current accuracy.**

**The main purpose of this paper is to demonstrate that this is feasible using the current model. To show this, the team has deployed two Doppler scatterometers (at Ku and Ka-bands) together with significant in situ resources, including a buoy to obtain surface wave spectra, HF-radar, and two kinds of drifters drogued at different depths. The final results of the paper show a good agreement between the theory of Nouguier et al. at Ka-band (although see detailed comments below), the band proposed for SKIM, but poor agreement at Ku-band and a different frequency dependence between Ka and Ku than predicted by the theory.**

**The experiment was carefully and thoughtfully designed and the team has made a significant effort to characterize the instruments, especially as regards the mean behavior of the signal. Some discussion has been devoted to the effects of antenna beamwidth at Ku-band leading to contamination of the Doppler signal due to the variation of the radar cross section within the radar footprint. However, given the qualitative discrepancy between theory and observations, additional effort should be devoted to quantifying the measurement errors to show that the Ku-band observations could be compatible with the theory, given feasible measurement uncertainties. Alternatively, physical sources for the discrepancy should be identified for future avenues of study. A more detailed suggestion is given below.**

We thank Dr Rodríguez for his thorough reading of our manuscript, for his many insightful suggestions, which we have done our best to implement, and for waiving his anonymity.

We share Dr Rodríguez's opinion that the paper was not clear enough regarding the reasons for the large discrepancies observed between the Ku-band radar measurements and the drifter-derived TSCV estimates. We have tried to address this issue in two ways:

- a first step has been to do a fresh analysis of the Ku-band data, and to correct very carefully the Azimuth Gradient Doppler contribution to the observed Doppler Velocity. This has reduced very significantly the discrepancy between the drifters data and the Ku-band Total Surface Current Velocity vector retrievals, which can now be considered reasonable.
- A second step has been to implement Dr Rodríguez's suggestion of reorganizing some of the material into a formal error budget (restricted to its geometrical factors), which is now contained in the new section 2.3. This error budget clearly shows that the azimuth pointing accuracy required to keep the error on the TSCV retrieval to reasonable levels is far out of reach of KuROS without the special compensation procedure mentioned above (recalling again that this instrument was not originally designed for this type of measurements).

**Overall, the paper has a logical outline. However integration of the different sections into a consistent style and level of detail has not been as successful, leading to some repetition and confusion, at times. The paper would benefit by a final integration to sharpen the presentation into a more uniform manuscript.**

We have thoroughly rewritten the text in order to make it more uniform and easier to follow. As also requested by an Anonymous Referee, we have done our best to remove repetitions to reduce the length of the manuscript.

**In spite of these reservations, I think that the data collected are an important data set that should be in the open literature and recommend its publication, hopefully after some of the more detailed comments below have been addressed. I recommend that the authors consider putting the data in the public domain, so that it can serve to lay the groundwork for work that will strengthen the case for the SKIM mission.**

We thank Dr Rodríguez for this appraisal of our work. Ensuring an open access to the Drift4SKIM dataset will be performed in the course of the IASCO project, funded by ESA.

**2 Error Quantification**

**Although there is a numerical discussion of various error sources (especially biases due to the antenna pattern and azimuthal variations of backscatter cross section), there is no attempt at deriving an error budget for either of the instruments. This would not be important if the observed measurement scatter were small. However, it is far from small, as can be seen in Figures 12, 13 and 16, where measurement standard deviations varying from 1 m/s to 2 m/s can be observed. Figure 12 is very enlightening about the variation characteristics of the Ka-band measurements, and an equivalent version would have been very useful for Ku-band. For SKIM, it is important to show that not only the**

**model predicting the mean behavior is understood, but also that the error performance is understood. Currently, this information is not contained in the paper, but all the data are available to produce this validation.**

**The error budget should contain, at least:**
**1) Expected measurement random velocity errors, which can be calculated in a straightforward fashion from the pulse pair correlation.**

**2) Contributions from pointing errors. For KuROS, the incidence angle is very well constrained by the high range resolution (although platform elevation couples in at shallow angles, as noted by the authors), but this is not the case for KaRADOC, where a single footprint is used. Typical aircraft roll (and, to a lesser extent, pitch) variations will lead to variations in the local incidence angle of up to a few degrees (leading to large errors, if uncorrected) , and it is not clear in the description of the processing how these effects are mitigated.**

**3) Error bounds on the possible Doppler effects due to uncertainties in the antenna pattern.**

**4) Error bounds on the expected effects of the sigma0 azimuth modulation errors as a function of azimuth, which can be obtained using the wavelength of the resolved waves, shown in Figure 9.**

**5) Modeling assumptions(see below).**

As stated above, we have done our best to implement this suggestion. We have not provided such a fine analysis of the different contributions mentioned, but have at least delineated them, and given their geometrical weighting factors. Clearly, much work remains for further contributions.

**Both radar systems have high PRF to properly sampling the Doppler. Is the contamination due to range ambiguities significant? Has it been considered as a source of error?**

Due to the near-nadir viewing geometry and low flight altitude of KuROS with respect to DopplerScatt, each KuROS pulse can be received and processed before the following pulse is transmitted. Processing the KuROS data is thus comparably easier, and the instrument is not affected by the range ambiguity problem.

**Examination of Figure 12 shows passes in the east-west direction have lower levels of variations than those going north-south. In addition, the frequency of variation is higher on the 22nd than on the 24th, but the amplitude of variability is larger on the 24th. What is the reason for this? It does not seem to align with wind or wave directions. In any case, the characteristics of the variations seem**

to be long-wavelength, leading one to suspect either attitude errors or errors due to the changes in the surface field characteristics. Examining the equivalent noise characteristics of the Ku-band data would potentially help in understanding the differences between the two frequencies.

The data shown in figure 12 have been low-pass filtered to remove the large fast variations due to individual waves. This has now been stated explicitly in the caption. We have checked the long-wavelength variations are not linked in a straightforward way to the plane attitude, and our current opinion is that they are caused by changes in surface-field characteristics. Which can only be briefly mentioned in this already very long article, but will be the subject of further contributions by the Drift4SKIM team.

As regards the difference between the Ka-band and Ku-band Doppler Velocity data, as stated above, our conclusion is that it is caused by a systematic mispointing effect caused by the KuROS antenna radiation diagram.

**One observation is that, comparing the variations in Figure 16 and 13, the level of within track variability is smaller for Ku band than for Ka-band. Thus the lack of agreement with the model is not due to higher random noise (as could be expected from wave sigma0 contamination), but through some systematic azimuth dependent effect. One potentially useful exercise is to assume that the azimuth brightness gradient contains additional harmonics to the ones estimated in going from Fig. 16a to 16b. Is it possible to account for the divergence from the model with these higher harmonics? If so, are these excluded by the sigma0 observations? Can they be ascribed to systematic coupling that might happen between the antenna pointing and the attitude? If these explanations are not feasible, does this indicate that additional physics needs to be incorporated into the model (at least at Ku-band)?**

Indeed, we agree with Dr Rodríguez that the discrepancy between the Ku-band and Ka-band data was caused by a systematic azimuth-dependent effect. We hope Dr Rodríguez is satisfied with the explanation we propose in the revised version of the text.

**3 Modeling and retrieval issues**

**There seems to be some mixed messages regarding the modeling assumptions. In Nouguier et al. (2018), a Gaussian assumption is made throughout. On the other hand, the authors quote the asymmetry and skewness of the slope distribution (with references to Munk (2008) and Chapron et al. (2002)) in order to explain the upwind/downwind asymmetry in the Ku-band backscatter cross-section (Figure 10), which is not insignificant. In equation 16, the isotropic backscatter curves of Nouguier et al. (2016) are used, but they are multiplied by an azimuthal modulation factor $F(\varphi)$, which is**

**not in the original paper and which does not seem to show up again in the analysis. Was such a factor used? If so, is it related to the azimuthal modulation factor quoted in the azimuth modulation fits quoted (but whose values are never given) in the second paragraph in page 21? If not, where is it coming from? Backscatter data are collected at Ku-band and presented in Figure 10A. Do these backscatter data fit the model in equation 16? If so, are the azimuthal modulations derived from these data for both Ku and Ka? If not, is there a justification for using equation 16 when it does not match the data?**

We agree the original manuscript was definitely obscure on these subjects. We have made a significant effort to clarify all these issues in section 2.2.

**In the Nouguier et al. (2018) paper, there are two models presented: one for range resolved or not range resolved Dopplers. Since KaRADOC is not resolving the waves, I assume that the second model is used. This model contains two parts (equation 15, Nouguier et al. (2018)), one which dominates along the wave direction, and another one which has contributions at other azimuths. In this paper, only one term seems to have been kept (i.e., equation 15, Nouguier et al. (2018)). What is the justification for neglecting the second contribution at other azimuths?**

We have clarified all these issues in section 2.2. As it turned out, the analysis was in practice not based on the equations mentioned by Dr Rodríguez. We confirm we have not attempted to apply the range-resolved formalism of the Nouguier et al (2018) paper to the Drift4SKIM observations. Again, though probably desirable, this probably would have required a lengthy discussion, which would have made the text even more unreasonably long.

**It is well known that non-Gaussian effects will lead to a correlation between the modulation of the slope rms and the location along the wave phase. This effect leads to the EM bias in altimetry, for example. Will the level of modulation consistent with EM bias results lead to a change in the predictions made by the model? Will it lead to an upwind-downwind asymmetry in the Doppler? Can it partially account for the 10-percent adjustment that had to be made to make the model predictions fit the data?**

Though we share Dr Rodríguez's interest in these issues, we have not yet been able to analyze the Drift4SKIM dataset in sufficient depth to identify how we could contribute answers to all these questions.
It is definitely in our plans for the forthcoming years to clarify these issues and assess the impact of non-Gaussian behaviour of the sea state on potential SKIM current retrievals, but this was not feasible in the scope of this necessarily limited first analysis of the dataset.
At present, as stated in the text, our position on this point is that uncertainties on the

directional spread of the sea state are sufficient to explain the 10% discrepancy between the modeled Wave Doppler contribution and the observations.

**In the retrieval of the surface currents, it was assumed that the current in the scene remained constant. However, as shown in Table 2 and Figure 7, there was significant change in the currents due to tidal variations measured by the Trefle buoy. How was this accounted for during the fitting? The HF-radar imager linked to in the paper also show some current gradients in the region: were they observable by the radars? Table 2 also shows significant disagreement between the Trefle buoy velocities and those from the other in situ data. Could you comment on the source of discrepancy?**

Once again, these effects, though interesting, were not sufficiently well resolved during the experiment to lend themselves to a thorough analysis. Our approach has thus been to compare time and space averages of the surface current estimates obtained using the different instruments. This unfortunately tends to degrade the agreement, by leaving as "unexplained discrepancies" effects which could be reduced into "resolved variability" by a more careful analysis. We felt this was however still out of the scope of this first account of the Drift4SKIM experiment.

Regarding the disagreement between the Trèfle and other in-situ velocities in Table 2, we suspect a misunderstanding: the data reported as "buoy ($U_s$, $V_s$)" in the table are the Stokes drift components at the center of the "Offshore" area, estimated from the Trèfle buoy IMU data on November 22$^{nd}$ and from the closest Spotter buoy on November 24$^{th}$. The figures are indeed markedly different from the drifter velocity data, but are in reasonable agreement with the Stokes drift estimates provided by the WaveWatch3 model.

**4 Miscellaneous comments**
**Figure 5 appears with insufficient attribution or description. Part of it comes from Nouguier et al. (2016), but there are additional subpanels whose provenance should be clarified.**

Details for each panel have now been added to the caption:

**The term mssshape is introduced with just a reference to Nouguier et al 2016. To make things easier for the reader, it should be clarified that it is the apparent rms slope obtained by fitting the backscatter curves.**

Indeed, the $mss_{shape}$ is a parameter that is a function of the radar wavelength and is obtained from the variation of backscatter with azimuth. This is now clarified in the discussion of equations (16) and (17).

**In page 32, there is a statement made about the equivalent depth of the measurements from near-nadir Doppler scatterometry. However, no such derivation is presented in the papers referenced. It would be useful to the community of this statement were backed with a calculation for the two wind speeds (perhaps as an appendix)**

We appreciate the importance of this comment. However, given the length and complexity of the present paper we have removed the discussion of this point.
We are working on a short note giving the details of the theoretical and expected current measurements in the presence of a vertical shear (Nouguier et al., in prep).

**Point-by-point reply to the report from Reviewer 2**

**Review on the paper "Measuring ocean surface velocities with the KuROS and KaRADOC airborne near-nadir Doppler radars: a multi-scale analysis in preparation of the SKIM mission" by L. Marié et al.**

**This paper presents the technique and examples of current velocity measurements from an airborne platform carrying two radars. The data acquisition was performed at a site located off the western coast of France where the surface currents are continuously monitored by two HF radars. Surface drifters were deployed for validation of airborne velocity measurements. The paper provides a detailed discussion of the experimental setup, measurement technique, errors, and comparison of the velocity data from different sensors.**
**I find the paper very deep, well worth publishing, providing very valuable information for developers of radars for velocity measurements from space. All figures are of excellent quality. Congratulations to the authors on a good paper, but an even more impressive field campaign.**

We thank the referee for his/her careful reading of our manuscript, and for taking the time to contribute this positive appraisal of our work to the interactive discussion of the article. We have revised the article to address the issue he/she and the other reviewer had raised. In the following we detail what modifications we have performed in response to his comments.

**However, the paper would benefit from the following changes (not only minor):**

**– The text needs a substantial reduction: 33 pages and 15 pages in Appendix, this is too much. Please make sure authors are happy with this?**

We understand (and share) the concern expressed by the reviewer.
We have done our best to streamline the flow of the paper and tried to make it as easy to read as possible.
The field of Doppler radar oceanography is however fairly new. Presenting the results of the Drift4SKIM campaign in fact also requires presenting a number of concepts that had to be developed during the analysis. We have not been able to reach this dual goal of presenting the data with sufficient background in a pedagogical **and** terse way.
In fact, only section 2.2 can be considered as very strongly inspired by the previous work of Nouguier and collaborators, the rest being fully original. Including this section makes the article a self-contained introduction to the technique for workers from

other fields.

Also, we have kept long appendixes, which make the article a self-contained reference for future work. Some of this work, which would not deserve publication by itself, would otherwise probably be lost.

We hope the reviewer will forgive us for this.

**Many formulas and demonstrations come from the paper of Nouguier et al., 2018 and Rodríguez et al., 2018. The saving is worth it.**

As mentioned above, though the cited articles have been an important source of inspiration, we had to adapt significantly the concepts they developed to the near-nadir observations performed during Drift4SKIM. We have done our best to remove any duplication of these articles and to shorten our text, which admittedly remains long.

**–The text needs a closer proof reading. There are some typos (altitude/attitude in page 39, 40; are/is following the word data within the whole paper, Appendixes/appendices in page 16, ...**

We have thoroughly searched the text for such issues.

**–The main body of the paper requires a number of changes to the text where it appears confused while Appendixes are well written and very clear.**

The main text has been clarified.

**Specific points.**

**Abstract: what is the major finding in this study? Only an estimate of C0? The description of the experiment should be shortened giving the place to the main results.**

In our opinion, there are several findings in this study:

- we have developed a number of concepts necessary for the analysis of Doppler Velocity data collected from a fast-moving platform.

- we provide an experimental check of the fact that the Kirchhoff Approximation electromagnetic model provides good estimates of the wave-induced component of the Doppler Frequency Shift.

- this allows us to provide confirmation of the fact that the norm of this component is weakly variable with respect to environmental variables, and that the direction follows quite closely that of the wind.

- we demonstrate the feasibility of retrieving the Total Surface Current Velocity vector from radar Doppler observations of the sea surface.

**P3 L15 something is wrong with the English of this sentence? The contribution ... of contributions**

The main text has been thoroughly searched for such issues.

**P4 L2 measurement equation. Maybe measurement is not necessary?**

This sentence has been corrected.

**P9 Figure 4 caption: contribute to or contribution to. "to" is missed.**

This sentence has been corrected.

**P10 Some problems with the English in many places.**
**L1-2: the sentence seems not finished.**
**L7: U is the current speed ...**
**L8 wave slope variability? spectrum.**
**L16 While the incidence angle increases ... the backscatter becomes dominated**
**L27,30. eq. 14 contains phi or phi_s? it is confusing.**

On the basis of the comments from both referees, we have thoroughly rearranged the text of section 2. We have done our best to make that important section clear, easy to read, and syntactically correct.

**P11 L24: something gone wrong in this sentence. ... work was focused in two boxes. Perhaps, work performed in locations matching by two boxes in Figure 6 …**

We have removed references to "boxes", and used the word "area" instead in the text.

**P13-P14. The text is very confusing and should be re-written.**

We have done our best to clarify the text of sections 3.1 and 3.2.

**P16 L8. Please check for frequency and remove band if only one frequency is used.**

This sentence has been corrected.

**L12 How to understand the ambiguity of 126 m/s ?**

We have been more explicit in our discussion of ambiguity in section A1.4.

126 m/s is equal to the upper bound of the unambiguous velocity interval at the KuROS wavelength and PRI.

**P17 L8 Consider: observations corresponding to Phi=12deg are reported.**

This sentence has been corrected.

**P19 L1-4. Please remove repetition in this sentence: 30 seconds**

This sentence has been corrected.

**P21 L12: Consider: Due to the narrower radar beam, the data from Karadoc are easier to interpret than the data from Kuros.**

We have implemented the referee's suggestion.

**P21 L14 and P22 L1-2: something is wrong with the English in these lines.**

This paragraph has been rephrased.

**P23 Figure 13 caption: remove one "blue" and complete the sentence.**

The caption of Figure 13 has been corrected.

**P26 L1 Consider ... spectra estimated from measurements on November 2 ...**
This sentence has been corrected.

**P26 L4 energy is much lower than**

This sentence has been corrected.

**P31 L7 Perhaps: Regarding the radar measurements, …**
This sentence has been corrected.

**P33 L19-21. This conclusion is confusing and should be re-written**

We have rephrased the conclusion in our revised version, taking into account the comments from the referee as well as the evolution of the text.

**List of changes made to the manuscript**

As requested by both referees, the manuscript has been thoroughly rewritten to make it more uniform and easier to read. A point-by-point list of all the modifications would probably fail to convey the intended information. We thus only provide here a list of "macro-changes" performed in response to the referee comments. With respect to the previous version, we have made the following changes:

- The title has been slightly amended.

- Dr Peter Sutherland, who contributed during the field work and in the drafting of this second version, has been added as a coauthor.

- The body of the text has been corrected for typos and homogenized, as had been requested by the referees. In this process, most of the text has been amended, but the result is in our opinion much easier to follow.

- The abstract has been rewritten to reflect changes undergone by the rest of the text. We have striven to make our claims and main findings more clearly apparent.

- Section 1 (introduction) has been amended in minor ways.

- Section 2 (theory) has been homogenized and rewritten, with the aim to make it more focused on the particular challenges of near-nadir radar Doppler observations. A subsection pertaining to the overall error budget of the technique, as requested by Ernesto Rodriguez as a referee, has been added.

- Section 3 (description of the field experiment) has been homogenized and rewritten, but did not change substantially.

- Section 4 (results) has been thoroughly rewritten. The text has been clarified in many places. We have re-analyzed our data, and managed to explain the major difference observed in the first version between the Ka-band and Ku-band radar measurements. The major instrumental bias present in the Ku-band data in the first version has been corrected in a much better way, and the Ku-band results can now be accounted for by the theory. One finding is also that the directional spread of the sea state seems to bear a stronger influence on the waves-induced contribution to the observed Doppler Frequency Shift than on previous quantities of remote sensing interest (such as normalized backscattering cross-section). Improving the results of this measurement technique may require developing a better understanding of higher-order statistics of the sea state than previously available and necessary. The discussion of KuROS data collected at 18° incidence angle, which did not bring in new information, has been removed to shorten the text.

- Sections 5 (implication for SKIM) and 6 (conclusion and perspectives) have been rewritten accordingly.

[revised manuscript text omitted]

–  Validate the Radar Sensing Satellite Simulator (**?**) and its capability to  simulate airborne configurations.
* * *
**Figure 1.** (A) Schematic of ATR-42 and KuROS instrument and definition of viewing angles, azimuth $\varphi$ and incidence angle $\theta$, and (B) comparison with the SKIM viewing geometry.  The unit vector $\mathbf{e}_\varphi$ is the projection on the horizontal of the line of sight direction vector. The variation of surface backscatter across the footprint and as a function of azimuth $\varphi$, which causes the effective mispointing $\delta\varphi$, is represented  as a grey shading. In the KuROS data, each measurement is integrated in azimuth across the antenna lobe. In the case of SKIM, the use of unfocused SAR processing allows the separation of echoes in the azimuth direction with a resolution dDop$\simeq$ 300 m.

As highlighted in Figure 1, the viewing geometry of an airborne system is vastly different from that of a satellite system, with a much smaller footprint and  incidence angle variations at scales comparable to the wavelength of the dominant ocean waves.  Another obvious difference is the stability of the platform and its velocity, 7 km/s for low Earth orbit, and around 120 m/s for the ATR-42 aircraft used here. As a result, transposing the performance of  an airborne system to a satellite system

 requires a thorough analysis, supplemented by carefully designed and validated simulation tools. Performing this analysis is however worthwhile, as it leads one to develop valuable insight into the instrument imaging principle and design

25 trade-offs.

This article is intended to provide an overview of the Drift4SKIM campaign data and a first discussion of their implications for the emerging field of near-nadir Doppler radar observations of TSCV. It is structured as follows: the principle of the pulse-pair measurements and the  different contributions to the observed DFS are detailed in section  2 and Appendix ??. Section ?? gives a brief account of the

30 field work performed and conditions encountered during the campaign. The results of the airborne measurements are exposed in section  ??. Results and implications for SKIM are then discussed in section  ??. Conclusions and perspectives follow in section  ??.

**2   Near-nadir radar Doppler measurements of ocean velocities: theory**

of a target, in our case Ship-borne Doppler measurements of ocean currents are routinely performed

5 using so-called "Vessel-Mounted Acoustic Doppler Current Profilers" (VMADCPs, see for instance ?). Some of the  data processing concepts transpose directly to the space-borne context: the raw DFS signal contains a large non-geophysical contribution due to the platform motion, which must be estimated from ancillary sensors and compensated. The accuracy of the final geophysical product is practically set by the accuracy of the non-geophysical velocity estimation and correction procedure. In the VMADCP context, however, the backscattering elements responsible for the production of the acoustic return signal

10 (particulate suspended matter, zooplanktonic organisms) are passive and follow accurately the water mass. This does not carry over in the electromagnetic case: here, the return signal is produced by the interaction of the transmitted signal with the roughness elements of the sea surface, which move with respect to the water mass with an intrinsic phase velocity that is an order of magnitude larger than typical ocean currents. This effect is for instance well known in the  ground-based HF-radar currents measurement context (?), and must also be compensated.

15 In our case, the measurement geometry is represented in figure 1, and the line-of-sight Doppler velocity $V_{\mathrm{LOS}}$  looking towards incidence angle $\theta$ and azimuth $\varphi$  (in this paper, line-of-sight DV contributions are denoted by "V", and the corresponding horizontal velocity contributions are denoted by "U") is the sum of the projection of a horizontal current contribution $U_{\mathrm{CD}}(\varphi)$, a  wave-induced contribution $V_{\mathrm{WD}}(\theta,\varphi)$ and a non-geophysical ~~velocity $V_{\mathrm{NG}}$. The following measurement equation is given by projections of the target and sensor

20 velocity vectors onto the line of sight as shown in figure 1,~~ contribution $V_{\mathrm{NG}}(\theta,\varphi)$. The equation that permits the retrieval of the TSCV contribution $U_{\mathrm{CD}}(\varphi)$ from the raw measured $V_{\mathrm{LOS}}$ can be written as

$$U_{\mathrm{\underline{GD}CD}}(\varphi) = \underline{V_{\mathrm{LOS}}(\theta,\varphi) - V_{\mathrm{NG}}(\theta,\varphi)/\sin\theta} \frac{[V_{\mathrm{LOS}}(\theta,\varphi) - V_{\mathrm{NG}}(\theta,\varphi) - V_{\mathrm{WD}}(\theta,\varphi)]}{\sin\theta}. \tag{1}$$

The aim of this section is to provide a detailed analysis of the different terms of this expression. The non-geophysical contribution $V_{NG}$ is discussed in subsection **??** and Appendix **??**. The wave Doppler contribution is discussed in subsection **??**. A brief summary of the measurement error budget is finally provided in subsection **??**.

**2.1 Non-geophysical velocity $V_{NG}$**

~~In practice, $V_{NG}$ is the radar velocity projected onto the effective look direction, that includes an apparent azimuth mispointing $\delta$ due to the finite antenna beamwidth combined with the variations of NRCS in the radar footprint. This NRCS variability includes both spatial gradients and azimuthal gradients. As a result, the beamwidth is a very important parameter of the radar, and the values for KuROS and KaRADOC are given in Table 1. For KuROS they have been determined following the procedure detailed in Appendix B. For KaRADOC, they are the result of anechoic chamber measurements (Appendix **??**).~~

 As mentioned above, the accuracy of ship-borne acoustic Doppler current measurements is affected in a dominant way by the platform motion compensation process. In the space-borne context, the  platform velocity is almost three orders of magnitude larger (7000 m.s$^{-1}$ *vs.* 10 m.s$^{-1}$). The accuracy requirements are thus tremendously exacerbated, and attention must in particular be paid to the detailed effects of the antenna  radiation diagram and sea-surface Normalized Radar Cross Section (NRCS) variations with space and observation azimuth. A detailed discussion of these effects is given in Appendix **??**.

In summary, in the case of ~~level flight and for low incidence observations, one can approximate the antenna pattern as a Gaussian 1-way antenna pattern with a parameter $\sigma_\alpha$. Using usual radar conventions the beamwidth is given by a 1-way full antenna width $\alpha_{-3dB}$, that is the angle between the two directions for which the transmlitted power is reduced by 3 dB compared to the maximum radiated power in the boresight direction. With the the usual approximation $10\log(0.5)/\log(10) \simeq -3$, we have~~

[revised manuscript text omitted]

When $\sigma^0$ varies at scales comparable to the footprint, e.g. $\sigma^0 = a^0[1 + \epsilon\sin(\nu\varphi)]$, then

——

.

This mispointing is maximum for $\nu = \sin\theta/\sigma_\alpha$, and the smaller scales, those with higher values of $\nu$, average out. The larger scales only give a small variation across the antenna pattern. This will be further discussed in section 5 in the context of SKIM. For large scale variations, $\nu \to 0$ and $\epsilon\nu \to \partial\sigma^0/\partial\varphi/\sigma^0$, so that we recover eq. (6).

**2.2 Geophysical velocity $U_{\text{GD}}$: Waves and Current Doppler velocities**

The geophysical part of the Doppler shift DFS measured by a microwave radar over the ocean, using both Along-Track-Interferometry and Doppler centroid techniques is caused by the backscatter-weighted average of the surface velocities along the emerges from the average over the instrument field of view (FOV) of the backscatter-weighted line of sight projection of the surface velocity, as illustrated in figure ??.

For a perfect sine wave of period $T$ propagating over deep water, the phase speed of the wave is

$$C = \frac{gT}{2\pi} + U\cos(\varphi_w - \varphi_U)$$

where $U$ is the current speed, $\varphi_w$ is In the well-understood case of decametric electromagnetic waves interacting with the sea surface at grazing incidence, the interaction is dominated by the Bragg coherent backscattering mechanism (?), in which the

[Figure]

**(A) waves motion only**

radar

2 **measurements**:
- **velocity** in line of sight(length)
- **backscatter** (width)

facet velocity

normal to facet
elementary
rough facet

γ

$U_{LOS}$

wave propagation

direction

sea surface

radar

**(B) waves+current**
**= total motion**

sea surface

γ

**Figure 4.** Schematic of (A) wave and (B) wave and current contributions to Doppler velocities at the scale of elementary facets. These small-scale processes are averaged over the radar field of view, and a mean velocity signal emerges due to the correlation of surface brightness and velocities in the wave field.

270   backscattered field reflects the properties (amplitude, phase speed) of a very finely selected component of the sea state, namely that whose wavevector is precisely equal to the so-called Ewald vector, the wave propagation azimuth direction, and φ_U is the current direction. Measuring difference between the wavevectors of the scattered and incident electromagnetic waves. Exploiting the deviation of the phase speed deviation from the of this sea state component from its theoretical value is the principle of the coastal HF radars (??), for which the grazing angles coherent Bragg back-scattering mechanism selects very effectively the

275   sine wave components of the sea state which interact with the radio waves.

 HF radars operationally used to measure the ocean TSCV in coastal areas (**??**).

 In the case of the near-nadir interaction of microwaves with the sea surface, which is the configuration considered for SKIM and used by the AirSWOT, KuROS and KaRADOC airborne instruments, this mental picture must be adapted: the Bragg scattering mechanism is not dominant, and the main contribution comes from quasi-specular reflections on those facets  the

$$C \quad \to \quad U_{\text{GD}}$$

$$\frac{gT}{2\pi} \quad \to \quad U_{\text{WD}}$$

$$U \quad \to \quad U_{\text{CD}}$$

[revised manuscript text omitted]

——

——The required pointing knowledge is on the apparent pointing of the radar beam, which depends on the NRCS of the ocean surface $\sigma^0(x, y, \theta, \varphi)$. This NRCS is a property of the ocean surface that varies as a function of the horizontal position $(x, y)$ due to the presence of waves, varying winds, currents, surfactants, sea ice and all the physical properties of the sea surface. The NRCS also varies with the viewing geometry, in particular, within a radar range gate, the azimuth change $\varphi$ can be large enough to have a large impact on the Doppler. Both effects give apparent mispointings in elevation $\epsilon$ and azimuth $\delta$,

——

that can also be written as an additional "Azimuth Gradient Doppler" velocity component $U_{\mathrm{AGD}}$,

——

---

## Author Response (AR2)

**Letter to the Editor of "Ocean Sciences" regarding manuscript os-2019-77**

**Dear Professor Huthnance,**
**please find attached a revised version of our manuscript "os-2019-77", which implements the corrections suggested by Dr Ernesto Rodríguez and yourself.**

**With respect to the previous version, the following changes have been made:**

**- In response to Referee comments:**

**This is an excellent restructuring and clarification of the original paper and I recommend that it be published (almost) as is: the presentation is much cleaner and the main points are no longer obscured.**

We thank Dr Rodríguez for this appraisal of our work, and for his remarks and suggestions, which led to major improvements of the article.

**I would suggest only fixing typos, of which I found the following:**
**P10, L12: Directional → directional.**

This has been corrected.

**P5, L25: should be 100 m s-1, not 10 m s-1.**

In fact, "10 m s-1" refers to the ship-borne case, in which the figure is correct. We have clarified this in the text.

**P11, L25 (and subsequent): there may be some confusion in the x sign indication of the matrix product Mss-1 x msv0. The x could be taken as a vector cross-product. I suggest replacing with a ., indicating a contraction (or dot product) between an order 2 tensor and an order 1 tensor, leading to a vector as more conventional.**

This has been corrected.

**P22, L18: m s$^1$ should be m s$^{-1}$.**

This has been corrected.

**- In response to Editor comments:**

**P9, L9: "techniques, emerges" (add ",")**

This has been corrected.

**L30 and L31. Add "," after ">" and after Qz, OR remove "," after "elevation"**

This has been corrected.

**Equation 12. msv components should be between [ . . ] ?**

This has been corrected.

**Equation 15 and line after. Please standardise notation as $k_M$ or $\kappa_M$.**

This has been corrected. $k_M$ has been replaced by $\kappa_M$.

**Equation 17. I am puzzled by the disappearance of QH, especially as it appears on page 12 line 15.**

We have checked that Equation 17 is correct, explicitly introduced the expression of $\mathbf{Q_H}$ and $Q_z$ on P9, L30, and corrected the equation on P12, L15, which was indeed incorrect.

**P12, L12. "statistics, can be obtained" (add ",")**

This has been corrected.

**P14, L16. "others" -> "other"**

This has been corrected.

**P14, L17-18. "As . . . however" seem incompatible; omit "however"?**

This has been corrected.

**P17, L8. "Pierres Noires" (WMO #62069) . . buoy; please show in figure 6.**

This has been corrected.

**Page 29 line 3. Better "affects the magnitude of the computed UWD more at Ku-band than at Ka-band,"? [Word order for "more"]**

This has been corrected.

**Figure 8, left panel (22/11/2018). The swell appears to be from the NORTH-west.**

The figure has been corrected.

**Figure 12 caption. In your response to Reviewer 1 "The data shown in figure 12 have been low-pass filtered . . . . . . . This has now been stated explicitly in the caption." Not apparent to me in caption.**

We apologize for this mistake, which is now corrected.

**Page 48 line 4. Omit second ")"**

This has been corrected.

**Yours sincerely, and on behalf of all the coauthors of manuscript "os-2019-77"**

**Dr Louis Marié, PhD.**